# Plant autophagosomes mature into amphisomes prior to their delivery to the central vacuole

Jierui Zhao[1,2]*, Mai Thu Bui[1]*, Juncai Ma[3]*, Fabian Künzl[1], Lorenzo Picchianti[1,2], Juan Carlos De La Concepcion[1], Yixuan Chen[1], Sofia Petsangouraki[1], Azadeh Mohseni[1], Marta García-Leon[1], Marta Salas Gomez[1], Caterina Giannini[4], Dubois Gwennogan[5], Roksolana Kobylinska[1], Marion Clavel[1], Swen Schellmann[6], Yvon Jaillais[5], Jiri Friml[4], Byung-Ho Kang[3], and Yasin Dagdas[1]

**Autophagosomes are double-membraned vesicles that traffic harmful or unwanted cellular macromolecules to the vacuole for recycling. Although autophagosome biogenesis has been extensively studied, autophagosome maturation, i.e., delivery and fusion with the vacuole, remains largely unknown in plants. Here, we have identified an autophagy adaptor, CFS1, that directly interacts with the autophagosome marker ATG8 and localizes on both membranes of the autophagosome. Autophagosomes form normally in *Arabidopsis thaliana cfs1* mutants, but their delivery to the vacuole is disrupted. CFS1's function is evolutionarily conserved in plants, as it also localizes to the autophagosomes and plays a role in autophagic flux in the liverwort *Marchantia polymorpha*. CFS1 regulates autophagic flux by bridging autophagosomes with the multivesicular body-localized ESCRT-I component VPS23A, leading to the formation of amphisomes. Similar to CFS1-ATG8 interaction, disrupting the CFS1-VPS23A interaction blocks autophagic flux and renders plants sensitive to nitrogen starvation. Altogether, our results reveal a conserved vacuolar sorting hub that regulates autophagic flux in plants.**

## Introduction

Macroautophagy (hereafter autophagy) is a conserved vacuolar trafficking pathway that mediates three Rs in eukaryotic cells, including plants: (i) It *remodels* the cellular environment for developmental and temporary reprogramming events that underlie cellular differentiation and adaptation. In plants, for example, autophagy is essential for callus regeneration in *Arabidopsis thaliana*, wound-induced de-differentiation in *Physcomitrium patens*, sperm maturation in *Marchantia polymorpha*, and pollen formation in rice (Rodriguez et al., 2020; Norizuki et al., 2021 Preprint; Kurusu et al., 2014). Autophagy-mediated cellular adaptation is also crucial for stress tolerance including drought, infection, and high temperature stress. Studies involving *Arabidopsis*, maize, and rice have shown that autophagy mutants such as *atg2*, *atg5*, and *atg7* are highly susceptible to biotic and abiotic stress factors and undergo early senescence (Signorelli et al., 2019; McLoughlin et al., 2018; Wada et al., 2015). (ii) At the cellular level, autophagy *renovates* the cell by removing the organelles, protein complexes, and other dysfunctional macromolecules that would otherwise reduce cellular fitness (Dikic, 2017; Marshall and Vierstra, 2018). Finally, (iii) during nutrient limitation, autophagy *replenishes* cellular energy

pools and prolongs survival by recycling surplus cellular material (McLoughlin et al., 2020; Rabinowitz and White, 2010). Thus, autophagy is a major degradation and recycling pathway that keeps the cell in tune with the ever-changing environment and maintains cellular homeostasis.

The main vehicle of autophagy is a de novo formed, double-membrane vesicle termed the autophagosome. Autophagosomes capture their cargo and deliver them to the vacuole (or lysosomes in metazoans) for recycling. Autophagosome biogenesis involves the concerted action of highly conserved autophagy-related gene (ATG) proteins that coordinate the nucleation and growth of a cup-shaped phagophore around the autophagic cargo (Nakatogawa, 2020; Chang et al., 2021; Weidberg et al., 2011). The two opposing membranes are then sealed with endosomal sorting complex required for transport (ESCRT) proteins to form the autophagosome (Chang et al., 2021). Both autophagosome membranes are labeled with lipidated ATG8 family proteins that interact with (i) other ATG proteins to coordinate autophagosome formation, (ii) cargo receptors that selectively recruit cargo macromolecules and underlie selective autophagy, and (iii) adaptor proteins that mediate the

[1]Gregor Mendel Institute, Austrian Academy of Sciences, Vienna BioCenter, Vienna, Austria;   [2]Vienna BioCenter PhD Program, Doctoral School of the University at Vienna and Medical University of Vienna, Vienna, Austria;   [3]School of Life Sciences, Centre for Cell & Developmental Biology and State Key Laboratory of Agrobiotechnology, The Chinese University of Hong Kong, Shatin, New Territories, Hong Kong, China;   [4]Institute of Science and Technology Austria, Klosterneuburg, Austria;   [5]Laboratoire Reproduction et Développement des Plantes, Université de Lyon, École normale supérieure de Lyon, Centre national de la recherche scientifique (CNRS), Institut National de la Recherche Agronomique (INRAE), Lyon, France;   [6]Botanik III, Biocenter, University of Cologne, Cologne, Germany.

*J. Zhao, M.T. Bui, and J. Ma contributed equally to this paper.   Correspondence to Byung-Ho Kang: bkang@cuhk.edu.hk;   Yasin Dagdas: yasin.dagdas@gmi.oeaw.ac.at.

trafficking and vacuolar fusion of autophagosomes (Stolz et al., 2014). Most of these ATG8-interacting proteins contain highly conserved short linear motifs termed as the ATG8-interacting motif (AIM). The core AIM is denoted as [W/F/Y]xx[L/I/V], where x represents any amino acid. The AIM peptide is bound by the highly conserved W and L loops, collectively known as the AIM-docking site (ADS), on ATG8 (Birgisdottir et al., 2013). Despite recent advances in cargo receptor identification and characterization, no autophagy adaptors have been characterized in plants. As a result, we have only a limited understanding of how autophagosomes are delivered to the vacuole in plants.

Autophagosome maturation is logistically different in yeast, metazoans, and plants. In yeast, autophagosome biogenesis and maturation happen in the vicinity of the vacuole; the coordination of these processes is aided by spatial proximity (Zhao and Zhang, 2019). In metazoans, autophagosomes are formed at various sites around the cell and subsequently fuse with endosomes or lysosomes (Zhao et al., 2021). Autophagosome-endosome fusions create amphisomes, which mature into autolysosomes by acquiring lytic enzymes (Sanchez-Wandelmer and Reggiori, 2013). Despite these logistical differences, in both yeast and metazoans, the concerted action of dedicated soluble N-ethylmaleimide-sensitive factor attachment protein receptor (SNARE) proteins, tethering factors, and adaptors mediate the fusion of autophagosomes with the lytic compartments (Zhao et al., 2021). In plants, autophagosomes are formed at sites around the cell and are then delivered to the central vacuole, which can occupy as much as 80% of the cell volume (Marshall and Vierstra, 2018). The molecular details of autophagosome trafficking, fusion with the vacuole, and how these events are coordinated with other vacuolar trafficking pathways are currently unknown in plants. Whether or not plant autophagosomes converge into amphisomes before arriving to the central vacuole also remains unknown.

To address these questions, we focused on the identification of autophagy adaptors. We developed a differential centrifugation protocol to enrich for intact autophagosomes prior to affinity purification-mass spectrometry (AP-MS) with ATG8 as bait. This approach identified CFS1 (cell death related endosomal FYVE/SYLF protein 1), a highly conserved FYVE (Fab-1, YGL023, Vps27, and EEA1) and SYLF (SH3YL1, Ysc84p/Lsb4p, Lsb3p, and plant FYVE) domain-containing protein that was previously linked to autophagy (Sutipatanasomboon et al., 2017; Kim et al., 2022). Characterization of CFS1 revealed that it interacts with ATG8 in an AIM-dependent manner and specifically regulates autophagic flux in both *A. thaliana* and *M. polymorpha*. Genome-wide yeast two hybrid screening showed that CFS1 also interacts with the multivesicular body-localized ESCRT-I complex protein VPS23A. Live cell imaging and electron microscopy analyses demonstrate that CFS1 colocalizes with VPS23A at amphisomes. Further flux assays and phenotypic studies showed that the CFS1-VPS23A interaction is crucial for autophagic flux. Altogether, our data suggest that plants employ a prevacuolar sorting hub to coordinate vacuolar trafficking pathways in plants.

# Results

## Differential centrifugation coupled to affinity purification-mass spectrometry (AP-MS) revealed autophagosome-associated proteins in *A. thaliana*

Since autophagy adaptors play crucial roles in autophagic flux but are unknown in plants, we first set out to identify autophagy adaptors. We induced autophagy in GFP-ATG8A expressing *A. thaliana* seedlings with Torin1 treatment and performed differential centrifugation experiments to enrich for small membranous compartments, including intact autophagosomes, while removing larger, bulkier compartments such as organelles (Liu and Bassham, 2010; LaMontagne et al., 2016; Fig. 1 A). To test whether the membrane-associated fractions (P4) contain intact autophagosomes, we performed protease protection assays. Both NBR1, a well-characterized autophagy receptor that is localized within the autophagosomes, and ATG8, which localizes on both sides of the autophagosome, were protected in these assays (Svenning et al., 2011; Stolz et al., 2014; Fig. S1, A and B). After addition of Triton X-100, a detergent that destabilizes membranes, both proteins became sensitive to protease treatment (Fig. S1 B). These experiments suggested that we could enrich for intact autophagosomes (Borner et al., 2005). We then combined this approach with AP-MS of mCherry-ATG8E expressing lines and screened for protease-sensitive proteins (i.e., those localized on the outer autophagosome membrane) in the membrane-enriched fractions (Fig. 1 A). Mass spectrometry analysis revealed 48 proteins that are ATG8E-associated, P4-enriched, and proteinase K-sensitive, including SH3P2, FRA3, NUP93A, and NUP98A (Fig. 1, B and C; and Tables S1 and S2). One of these 48 proteins was the FYVE and SYLF domain-containing protein CFS1 (At3g43230; Fig. 1, B and C). Since CFS1 has previously been linked to autophagy and FYVE and SYLF domain-containing proteins are well-known players in vesicle trafficking (Sutipatanasomboon et al., 2017; Kim et al., 2022; Melia et al., 2020), we decided to characterize CFS1 in depth.

## CFS1 localizes to the autophagosomes

To map the cellular distribution of CFS1, we generated transgenic *Arabidopsis* lines that stably co-express mCherry-CFS1 with endomembrane compartment markers, including NAG1-EGFP (Golgi bodies), VHAa1-GFP (trans-Golgi network), GFP-ARA7 (late endosomes), and GFP-ATG8A (autophagosomes; Geldner et al., 2009; Bassham, 2015). Live-cell confocal imaging and quantification under control and two different autophagy inducing conditions (nitrogen starvation and salt stress) showed that mCherry-CFS1 puncta specifically colocalize with the autophagosomes (Thompson et al., 2005; Liu et al., 2009; Fig. S1, C and E; and Fig. 1 D). Mander's colocalization coefficients showed that both the M1 and M2 between mCherry-CFS1 and GFP-ATG8A are higher than 0.4, while the M1 and M2 between mCherry-CFS1 and other GFP markers are <0.2 (Fig. S1, D, F, and E). We also performed microscopy experiments with the amphiphilic styryl dye FM4-64, a stain that is widely used for tracing endocytic vesicles (Rigal et al., 2015). CFS1 did not colocalize with FM4-64 even under autophagy-inducing conditions (Fig. S1, G and H).

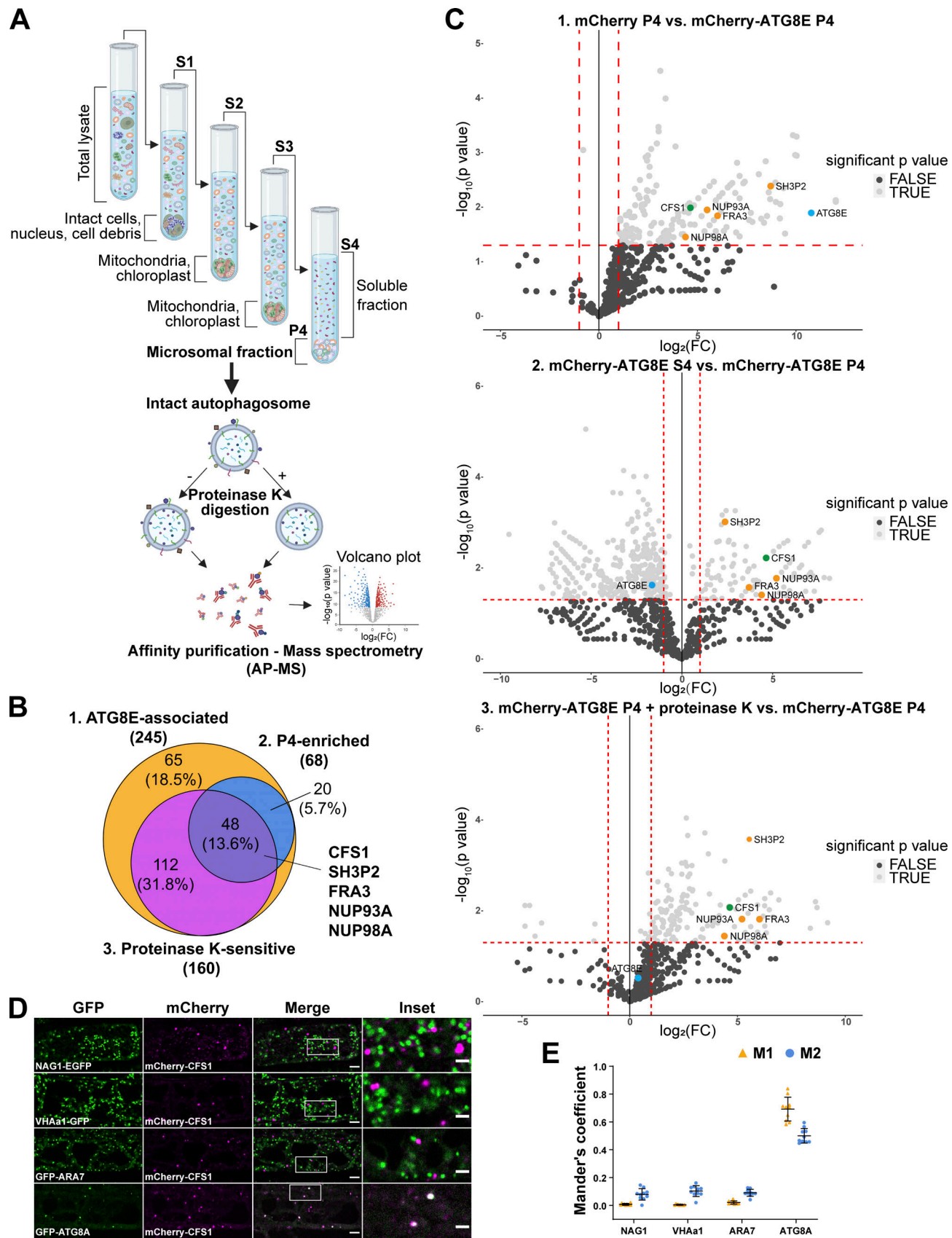

Figure 1. **Differential centrifugation coupled to affinity purification-mass spectrometry (AP-MS) revealed autophagosome-associated proteins in *A. thaliana*. (A)** Schematic diagram showing the differential centrifugation coupled to affinity purification-mass spectrometry (AP-MS) workflow. Total lysate of *Arabidopsis* seedlings underwent several differential centrifugation steps, where each time the supernatant (S) was transferred, and the pellet (P) was left

untouched. Samples were spun for 10 min at 1,000 $g$ (S1), to remove intact cells, nuclei, and cell debris; 10 min at 10,000 $g$, to remove bigger organelles like mitochondria and chloroplasts (S2); 10 min at 15,000 $g$, to further remove organelles (S3) and finally 60 min at 100,000 $g$ (S4 and P4; LaMontagne et al., 2016). The P4 fraction, containing small vesicles and microsomes (microsomal fraction), i.e., autophagosomes, was further subjected to 30 ng/µl proteinase K treatment (P4 + proteinase K). All S4, P4, and P4 + proteinase K samples were processed for AP-MS. **(B)** Venn diagram showing the overlap between ATG8E-associated, P4-enriched, and protease K-sensitive proteins that were identified by the AP-MS workflow described in A. 7-d-old *Arabidopsis* seedlings expressing pUBQ::mCherry or pUBQ::mCherry-ATG8E were treated with 3 µM Torin 1 for 90 min for autophagy induction before lysing. Diagram was generated using Venny 2.1.0 (Oliveros, 2016). **(C)** Volcano plots of mCherry-ATG8E AP-MS datasets identified CFS1 in all three pairwise comparisons. Upper panel, volcano plot of the pairwise comparison of "mCherry P4" and "mCherry-ATG8E P4" shows proteins enriched by the bait mCherry-ATG8E. X- and Y-axis displays $\log_2$ fold-change ($\log_2[FC]$) and $-\log_{10}$(P value), respectively. Dashed lines represent threshold for $\log_2(FC) > 1$ and P value <0.05. Only proteins passing both P value <0.05 and $\log_2(FC) < 0$ filter were considered in the first set "ATG8E-associated" in B. Middle panel, volcano plot of the pairwise comparison of "mCherry-ATG8E S4" and "mCherry-ATG8E P4," shows proteins enriched by mCherry-ATG8E in the pellet (P4) relative to the supernatant (S4). X-axis and Y-axis display $\log_2(FC)$ and $-\log_{10}$(P value), respectively. Dashed lines represent threshold for $\log_2(FC) > 1$ and P value <0.05. Only proteins passing both P value <0.05 and $\log_2(FC) < 0$ filter were considered in the second set "P4-enriched" in B. Lower panel, volcano plot of the pairwise comparison of "mCherry-ATG8E P4 + proteinase K" and "mCherry-ATG8E P4", shows proteins enriched by mCherry-ATG8E in the pellet (P4) before proteinase K treatment. X-axis and Y-axis display $\log_2(FC)$ and $-\log_{10}$(P value), respectively. Dashed lines represent threshold for $\log_2(FC) > 1$ and P value <0.05. Only proteins passing both P value <0.05 and $\log_2(FC) < 0$ filter were considered in the third set "Proteinase K-sensitive" in B. For all volcano plots, CFS1, interested proteins, and ATG8E are labeled by green, orange, or light blue dots, respectively. **(D)** Confocal microscopy images of *Arabidopsis* root epidermal cells co-expressing pUBQ::mCherry-CFS1 with either Golgi body marker p35S::NAG1-EGFP, trans-Golgi network marker pa1::VHAa1-GFP, MVB marker pRPS5a::GFP-ARA7 or autophagosome marker pUBQ::GFP-ATG8A under nitrogen starvation. 5-d-old *Arabidopsis* seedlings were incubated in nitrogen-deficient 1/2 MS media for 4 h for autophagy induction before imaging. Representative images of 10 replicates are shown. Area highlighted in the white-boxed region in the merge panel was further enlarged and presented in the inset panel. Scale bars, 5 µm. Inset scale bars, 2 µm. **(E)** Quantification of confocal experiments in D showing the Mander's colocalization coefficients between mCherry-CFS1 and the GFP-fused marker proteins NAG1, VHAa1, ARA7, or ATG8A. M1, fraction of GFP-fused marker signal that overlaps with mCherry-CFS1 signal. M2, fraction of mCherry-CFS1 signal that overlaps with GFP-fused marker signal. Bars indicate the mean ± SD of 10 replicates.

To further corroborate the autophagosome localization of CFS1, we visualized it with two other autophagosome-localized proteins: ATG11, a core autophagy protein that is crucial for recruitment of selective autophagy receptors and cargoes to the autophagosome, and NBR1 (Zientara-Rytter and Subramani, 2020; Svenning et al., 2011). mCherry-CFS1 colocalized with both GFP-ATG11 and NBR1-GFP upon induction of autophagy with salt stress (Fig. S1, I and J). Finally, we performed spinning disc time-lapse imaging of *Arabidopsis* lines that stably co-express mCherry-CFS1 with GFP-ATG8A or NBR1-GFP. Consistent with our confocal microscopy data, these time course experiments showed that CFS1 moves together with ATG8 and NBR1 puncta (Videos 1 and 2). Altogether, these results demonstrate that CFS1 specifically labels autophagic compartments.

A previous study analyzing FYVE domain-containing proteins in *Arabidopsis* identified another protein (CFS2, At1g29800) that also has FYVE and SYLF domains and shares 57.3% identity with CFS1 (Wywial and Singh, 2010; Fig. S2 A). Our phylogenetic analysis of CFS1 proteins across the plant kingdom detected no homology between CFS1 and CFS2 (Fig. S2 B). CFS1 and CFS2 formed separate, well-supported clades with different evolutionary histories. Nevertheless, we tested whether CFS2 plays a role in autophagy. We first carried out microscopy experiments similar to those described above. Although CFS2 was stably expressed, it did not form any puncta and had a diffuse localization pattern, even under autophagy inducing conditions (Fig. S2, C and D). Nitrogen starvation plate assays, which are typically used to evaluate autophagy defects in *Arabidopsis*, showed no difference between *cfs2* mutants and wild-type Col-0 plants (Phillips et al., 2008; Fig. S2 E). In contrast, *cfs1* mutants showed early senescence after 10 d of nitrogen starvation, similar to the autophagy-deficient *atg5* mutant (Thompson et al., 2005; Fig. S2 E). We then compared autophagic flux in *cfs1* and *cfs2* mutants. NBR1 is degraded upon induction of autophagy, and this is used as a proxy for

autophagic flux measurements (Bassham, 2015). Under both nitrogen starvation and salt-stress conditions, *cfs1* mutants had higher NBR1 levels compared to wild-type plants (Fig. S2, F–I). However, *cfs2* mutants showed no significant difference, and *cfs1cfs2* double mutants were comparable to *cfs1* (Fig. S2, F–I). Altogether, these results suggest that CFS2 is not involved in autophagy and prompted us to focus on CFS1 for further characterization.

### CFS1 interacts with ATG8 in an AIM-dependent manner

Autophagy adaptors interact with ATG8 directly through an AIM (Birgisdottir et al., 2013; Stolz et al., 2014; Zaffagnini and Martens, 2016). To investigate whether CFS1 interacts with ATG8, we first performed GST pull-down assays with *Arabidopsis* lysates expressing mCherry-CFS1 and *Escherichia coli* crude extracts expressing all nine GST-tagged *Arabidopsis* ATG8 isoforms or GST alone. mCherry-CFS1 interacted with GST-ATG8A to GST-ATG8F with similar affinities, but did not interact with GST-ATG8G, GST-ATG8H, and GST-ATG8I (Fig. S3 A). To further study whether CFS1 interacts with ATG8 in an AIM-dependent manner, we mixed crude extracts from *E. coli* expressing WT GST-ATG8A, an ADS mutant of ATG8 (GST-ATG8A[ADS]), or GST alone with *Arabidopsis* lysates expressing mCherry-CFS1 and performed GST pull-down assays. mCherry-CFS1 interacted with GST-ATG8A, but not with GST-ATG8A[ADS] or GST alone (Fig. 2 A). Consistently, the interaction between CFS1 and ATG8 could be outcompeted with a high-affinity AIM peptide, but not with an AIM mutant peptide (Stephani et al., 2020; Fig. 2 A). We then set out to identify the AIM in CFS1. Amino acid sequence alignments revealed a highly conserved candidate AIM between the FYVE and SYLF domains of CFS1 (Fig. S4). To test whether this motif is important for ATG8 interaction, we mutated the core AIM residues into alanine (WLNL-267-ALNA) and expressed the resulting mCherry-CFS1[AIM] in the *cfs1* mutant. This mutation did not affect CFS1

stability as both proteins accumulate to similar levels (Fig. 2 B). Pull-down experiments showed that, in contrast to mCherry-CFS1, mCherry-CFS1[AIM] interacted significantly less with GST-ATG8A (Fig. 2 B). We then performed in vivo co-immunoprecipitation (co-IP) experiments using plants stably co-expressing mCherry-CFS1 with GFP-ATG8A. We observed a strong association between CFS1 and ATG8A (Fig. 2 C). This association was further strengthened during nitrogen starvation, suggesting recruitment of CFS1 to the autophagosomes upon autophagy induction (Fig. 2 C). In contrast, mCherry-CFS1[AIM]-GFP-ATG8A association was substantially weaker under both control and nitrogen-starved conditions (Fig. 2 C).

We then performed nitrogen starvation plate assays to test the physiological relevance of the CFS1 AIM. Expression of wild-type mCherry-CFS1 rescued the nitrogen-sensitivity phenotype of cfs1 mutants. mCherry-CFS1[AIM] expressing plants remained sensitive to nitrogen starvation, similar to the cfs1 and atg5 mutants (Fig. 2 D). Finally, we compared the localization patterns of mCherry-CFS1 and mCherry-CFS1[AIM] relative to GFP-ATG8A. mCherry-CFS1 formed a significantly higher number of colocalizing puncta compared to mCherry-CFS1[AIM] under both control and autophagy-inducing conditions (Fig. 2, E and F). Collectively, these results suggest that CFS1 interacts with ATG8 in an AIM-dependent manner and that the CFS1–ATG8 interaction is essential for CFS1 function and autophagosome localization.

To further study the residual puncta formed by mCherry-CFS1[AIM], we crossed mCherry-ATG8E with the atg5 mutant, which does not form autophagosomes (Thompson et al., 2005). mCherry-CFS1 was still able to form a limited number of puncta in both basal and autophagy-inducing conditions, indicating that CFS1 forms puncta independently of autophagosomes (Fig. 2, G and H). To understand how CFS1 could still form puncta in the absence of autophagy, we looked at the other functional domains on CFS1. CFS1 has well-defined FYVE and SYLF domains that bind phosphatidylinositol-3-phosphate (Pi3P) and actin, respectively (Sutipatanasomboon et al., 2017; Fig. S2, A and B). We point-mutated the FYVE domain (RHHCR-195-AHACA) or the SYLF domain (K282A, R288A, K320A) of CFS1 to generate mCherry-CFS1[FYVE] and mCherry-CFS1[SYLF]. We also combined the mutations from mCherry-CFS1[AIM], mCherry-CFS1[FYVE], and mCherry-CFS1[SYLF] to generate a triple CFS1 mutant, mCherry-CFS1[tri]. Confocal microscopy results showed that mutating these domains, either individually or in combination, did not alter protein stability but did lead to diffuse localization patterns and disrupted ATG8A co-localization (Fig. S3, C and D), suggesting that CFS1 bridges Pi3P-rich endomembrane compartments with autophagosomes. In addition, in vivo co-IP experiments showed that compared to the wild-type mCherry-CFS1, mCherry-CFS1[FYVE], mCherry-CFS1[SYLF], and mCherry-CFS1[tri] associated less with GFP-ATG8A (Fig. S3 D). Altogether, these results suggest that in addition to the AIM, FYVE and SYLF domains are also important for CFS1–ATG8 interaction.

## CFS1 functions as an autophagy adaptor
We next explored the molecular function of CFS1 in autophagy. Three types of autophagy-related proteins contain AIMs: (i) core autophagy proteins involved in autophagosome biogenesis, (ii) autophagy receptors that mediate selective cargo recruitment and undergo autophagic degradation together with their respective cargoes, and (iii) autophagy adaptors, which interact with ATG8 on the outer autophagosome membrane and mediate autophagosome trafficking and maturation (Stolz et al., 2014). To determine whether CFS1 is involved in autophagosome biogenesis, we performed transmission electron microscopy (TEM) experiments in Arabidopsis root cells. Ultrastructural analysis of autophagosomes in cfs1 mutants showed fully formed autophagosomes, indistinguishable from those in wild-type cells, ruling out a role for CFS1 in autophagosome biogenesis (Fig. 3 A). To test whether CFS1 functions as an autophagy receptor or adaptor, we performed comparative vacuolar flux analysis, using NBR1 as a representative autophagy receptor. We stained the vacuolar lumen with BCECF-AM and blocked vacuolar degradation with the vATPase inhibitor concanamycin A (conA; Krebs et al., 2010; Bassham, 2015), which enabled us to quantify puncta within the vacuolar lumen. Upon induction of autophagy with salt stress, we found significantly less CFS1 puncta compared to ATG8 or NBR1 puncta (Fig. 3, B and C). Consistently, although both CFS1 and NBR1 colocalize with ATG8E at cytosolic autophagosomes, upon conA treatment there were fewer colocalizing CFS1-ATG8E puncta inside the vacuole compared to NBR1-ATG8E puncta (Fig. 3, D and E). These results suggest that CFS1 functions as an autophagy adaptor. Since autophagy adaptors should localize on the outer autophagosome membrane, we performed immunogold-labeling TEM experiments on Arabidopsis seedlings expressing mCherry-CFS1. We could readily detect gold particles on the outer autophagosome membranes (Fig. 3, F and G), consistent with CFS1 having an adaptor function. Of note, we also detected gold particles at the inner autophagosome membrane (Fig. 3 F), suggesting CFS1 is recruited to the autophagosomes during phagophore growth. In sum, the comparative vacuolar flux analysis and the ultrastructural localization experiments support the role of CFS1 as an autophagy adaptor.

## CFS1 is crucial for autophagic flux
Autophagy adaptors regulate the delivery or fusion of autophagosomes to the vacuole, known as autophagic flux (Stolz et al., 2014). We next examined the role of CFS1 in this process. First, we expressed mCherry-ATG8E in wild-type Col-0, cfs1, or atg5, and quantified the number of mCherry-ATG8E-labeled autophagosomes in root epidermal cells of these lines. Upon autophagy induction with salt stress, significantly more autophagosomes accumulated in the cytosol of cfs1 cells compared to Col-0 (Fig. 4, A and B). However, after treatment with conA, which stabilizes autophagic bodies in the vacuole, cfs1 mutants had significantly less mCherry-ATG8E puncta (Fig. 4, A and B). atg5 mutants had no autophagosomes in either condition (Fig. 4, A and B). These results suggest that cfs1 mutants have defects in the delivery of autophagosomes to the vacuole, which leads to the accumulation of autophagosomes in the cytosol. To support these data, we performed GFP-release assays under the same conditions. When GFP-ATG8 is delivered to the vacuole, a stable GFP fragment is released due to vacuolar protease activity.

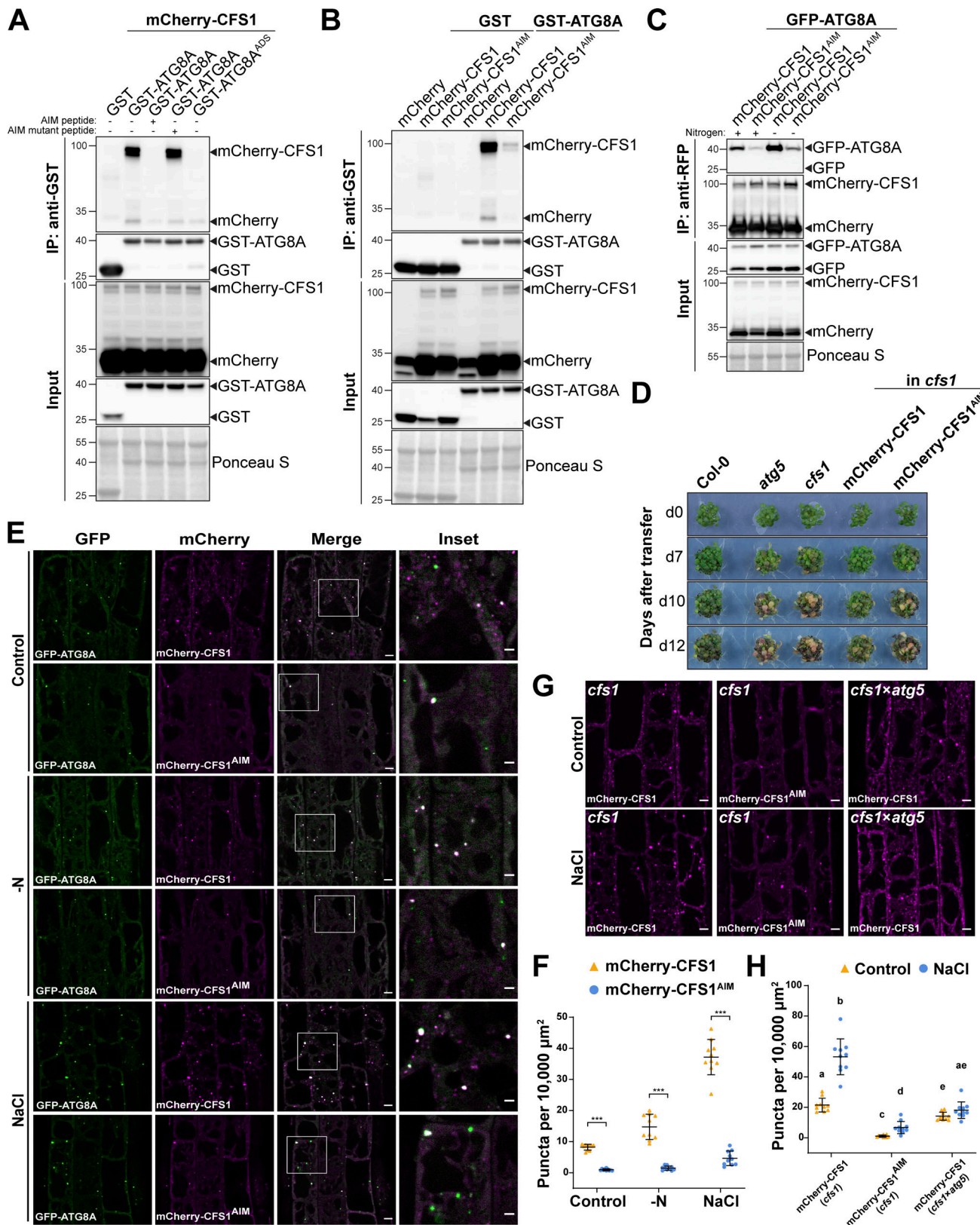

**Figure 2. CFS1 interacts with ATG8A in an AIM (ATG8 Interacting Motif)-dependent manner. (A)** GST pull-down coupled with peptide competition with *E. coli* lysates expressing either GST, GST-ATG8A, or GST-ATG8A^ADS and *A. thaliana* whole-seedling lysates expressing mCherry-CFS1. The peptides were added to a final concentration of 200 µM. Proteins were visualized by immunoblotting with anti-GST and anti-RFP antibodies. Representative images of three replicates are shown. Reference protein sizes are labeled as numbers at the left side of the blots (unit: kD). ADS, AIM docking site. **(B)** GST pull-down with *E. coli* lysates expressing either GST or GST-ATG8A and *A. thaliana* whole-seedling lysates expressing either mCherry, mCherry-CFS1, or mCherry-CFS1^AIM. Proteins

were visualized by immunoblotting with anti-GST and anti-RFP antibodies. Representative images of three replicates are shown. Reference protein sizes are labeled as numbers at the left side of the blots (unit: kD). **(C)** RFP-Trap pull-down of *Arabidopsis* seedlings co-expressing pUBQ::GFP-ATG8A with either pUBQ::mCherry-CFS1 or pUBQ::mCherry-CFS1^AIM. 7-d-old seedlings were incubated in either control (+) or nitrogen-deficient (−) 1/2 MS media for 12 h. Protein extracts were immunoblotted with anti-GFP and anti-RFP antibodies. Representative images of four replicates are shown. Reference protein sizes are labeled as numbers at the left side of the blots (unit: kD). **(D)** Phenotypic characterization of Col-0, *atg5*, *cfs1*, *cfs1* complemented with pUBQ::mCherry-CFS1 or *cfs1* complemented with pUBQ::mCherry-CFS1^AIM upon nitrogen starvation. 25 seeds per genotype were grown on 1/2 MS media plates (+1% plant agar) for 1-wk and 7-d-old seedlings were subsequently transferred to nitrogen-deficient 1/2 MS media plates (+0.8% plant agar) and grown for 2 wk. Plants were grown at 21°C under LEDs with 85 μM/m²/s and a 14 h light/10 h dark photoperiod. d0 depicts the day of transfer. Brightness of pictures was enhanced ≤19% with Adobe Photoshop (2020). Representative images of four replicates are shown. **(E)** Confocal microscopy images of *Arabidopsis* root epidermal cells co-expressing pUBQ::GFP-ATG8A with either pUBQ::mCherry-CFS1 or pUBQ::mCherry-CFS1^AIM. 5-d-old *Arabidopsis* seedlings were incubated in either control, nitrogen-deficient (−N) or 150 mM NaCl-containing 1/2 MS media before imaging. Representative images of 10 replicates are shown. Area highlighted in the white-boxed region in the merge panel was further enlarged and presented in the inset panel. Scale bars, 5 μm. Inset scale bars, 2 μm. **(F)** Quantification of confocal experiments in E showing the number of mCherry-CFS1 puncta per normalized area (10,000 μm²). Bars indicate the mean ± SD of 10 replicates. Two-tailed and paired Student *t* tests were performed to analyze the significance differences of the mCherry-CFS1 puncta number. ***, P value <0.001. **(G)** Confocal microscopy images of root epidermal cells of *cfs1* expressing pUBQ::mCherry-CFS1 or pUBQ::mCherry-CFS1^AIM, or *cfs1* × *atg5* expressing pUBQ::mCherry-CFS1. 5-d-old *Arabidopsis* seedlings were incubated in either control or 150 mM NaCl-containing 1/2 MS media for 2 h before imaging. Representative images of 10 replicates are shown. Scale bars, 5 μm. **(H)** Quantification of confocal experiments in G showing the number of mCherry-CFS1 puncta per normalized area (10,000 μm²). Bars indicate the mean ± SD of 10 replicates. Brown–Forsythe and Welch one-way ANOVA test were performed to analyze the differences of the mCherry-CFS1 puncta number between each group. Unpaired *t* tests with Welch's correction were used for multiple comparisons. Family-wise significance and confidence level, 0.05 (95% confidence interval), were used for analysis. Source data are available for this figure: SourceData F2.

The ratio of free GFP to GFP-ATG8 can thus be used to quantify autophagic flux (Bassham, 2015; Yoshii and Mizushima, 2017). Quantification of five independent experiments showed that *cfs1* had higher levels of full-length GFP-ATG8A and lower levels of free GFP under nitrogen starvation or salt-stress conditions compared to wild type (Fig. 4, C and D). As mentioned above, the rate of NBR1 degradation can be also used to measure autophagic flux (Bassham, 2015; Yoshii and Mizushima, 2017). Following autophagy induction with salt stress, NBR1 levels remained high in *cfs1* compared to wild type (Fig. S2, F–I; and Fig. 4, C and E). Collectively, these results show that CFS1 is crucial for autophagic flux in *A. thaliana*.

We then tested whether CFS1 is involved in autophagic flux during selective autophagy. Using our recently established uncoupler-induced mitophagy assays, we compared mitophagic flux in wild type and *cfs1* cells (Ma et al., 2021). Ultrastructural analysis of uncoupler-treated *cfs1* cells showed fully formed mitophagosomes, further confirming that CFS1 does not play a role in autophagosome biogenesis (Fig. 4 F). We then measured mitophagic flux using western blots. Uncoupler treatment led to a decrease in levels of the mitochondrial matrix protein isocitrate dehydrogenase (IDH). This decrease was restored upon conA treatment, confirming the induction of mitophagy. *cfs1* mutants had higher IDH levels upon uncoupler treatment, suggesting a defect in mitophagic flux (Fig. 4, G and H). When we examined mitochondrial ultrastructure in *cfs1* by electron microscopy, we saw accumulation of damaged mitochondria with distinctive electron dense precipitates, which were rare in wild type, but common in *atg5* mutant (Fig. 4 I). Altogether, these results suggest CFS1 is also crucial for selective autophagy flux in *A. thaliana*.

We next asked whether the function of CFS1 is conserved across plants by studying the *M. polymorpha* (Mp) CFS1 homolog. Stable *M. polymorpha* plants co-expressing mScarlet-MpCFS1 with GFP-MpATG8A or GFP-MpATG8B showed that MpCFS1 colocalizes with both MpATG8 isoforms (Fig. 5, A and B). Heterologous expression of MpCFS1 in *A. thaliana* also showed colocalization of MpCFS1 with GFP-ATG8A (Fig. 5, C and D). Finally, GFP-release

assays in *Marchantia* showed that *Mpcfs1* mutants have a defect in GFP-ATG8 degradation (Fig. 5, E and F). Together, these results suggest that CFS1 function is conserved across plants.

Since autophagic flux measurements assess vacuolar delivery, we decided to test if other vacuolar trafficking pathways are also affected in *cfs1* mutants. First, we measured the endocytic delivery of FM4-64 in Col-0, *cfs1*, *cfs2*, *cfs1cfs2*, and *cfs1* complementation lines. We quantified FM4-64 positive puncta upon conA treatment, which stabilizes endocytic vesicles. There was no difference between Col-0 and the mutant lines (Fig. 6, A and B). We then measured the uptake of a proteinaceous endocytic cargo, the auxin efflux carrier protein PIN2, in Col-0 and *cfs1* (Kleine-Vehn et al., 2008). Similar to FM4-64, upon induction of PIN2 endocytosis with dark treatment, we did not observe any difference between Col-0 and *cfs1* (Fig. 6, C and D). In addition, we compared the vacuolar morphology in Col-0, *cfs1*, *cfs2*, *cfs1cfs2*, and *cfs1* complementation lines, using BCECF-AM staining. We observed multiple small vacuoles in the epidermal cells of the root meristematic zone (Fig. 6 E) and intact central vacuoles at the transition zone in all lines (Fig. 6 F). Altogether, these results indicate that CFS1 specifically regulates autophagic flux without affecting other vacuolar pathways or vacuolar morphology.

## CFS1 interacts with VPS23A and mediates the formation of amphisomes

How then does CFS1 regulate autophagic flux? We hypothesized that it may interact with tethering factors, such as the CORVET or the HOPS complex, and thereby bridge the autophagosomes with the tonoplast (Takemoto et al., 2018). To test this hypothesis, we generated *Arabidopsis* lines that co-expressed mCherry-CFS1 with the CORVET complex component VPS3, the HOPS complex component VPS39, and the tonoplast localized SNARE protein VAMP711 (Takemoto et al., 2018; Geldner et al., 2009). Under both control and autophagy-inducing conditions, CFS1 did not colocalize with any of those proteins, negating out our hypothesis (Fig. S5, A–F).

This prompted us to step back and investigate the CFS1 interactome. We performed a genome-wide yeast two hybrid

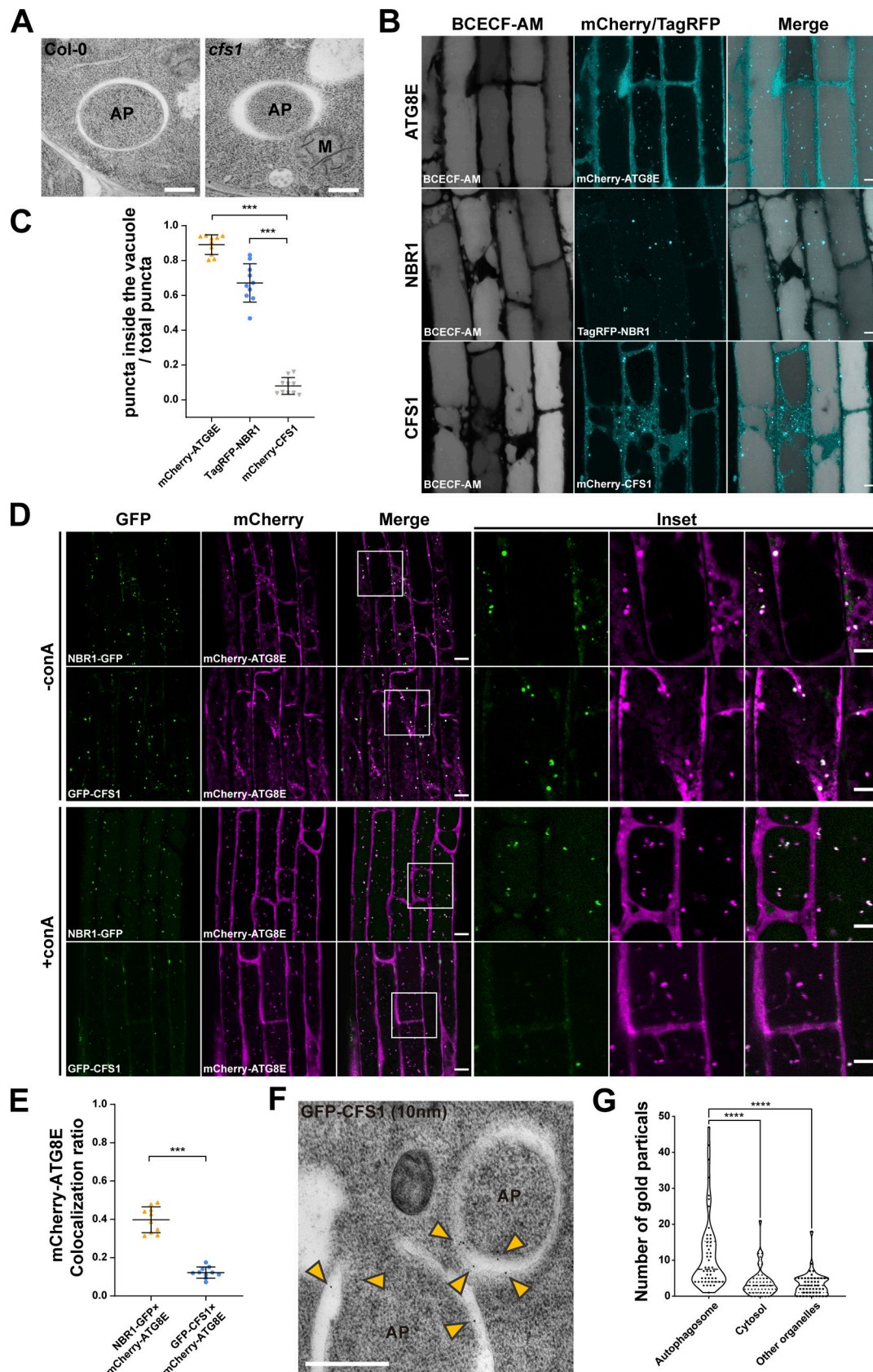

Figure 3. **CFS1 functions as an autophagy adaptor. (A)** Transmission electron microscopy (TEM) micrographs showing fully formed autophagosomes in the root cells of Col-0 and *cfs1*. 7-d-old *Arabidopsis* seedlings were incubated in 150 mM NaCl-containing 1/2 MS media for 1 h for autophagy induction before cryofixation. Scale bars, 500 nm. AP, autophagosome. M, mitochondrion. **(B)** Confocal microscopy images of *Arabidopsis* root epidermal cells expressing either

pUBQ::mCherry-ATG8E, pNBR1::TagRFP-NBR1 or pUBQ::mCherry-CFS1. 5-d-old seedlings were first incubated in 5 µM BCECF-AM-containing 1/2 MS media for 30 min for vacuole staining and were subsequently transferred to 1/2 MS media containing 90 mM NaCl and 1 µM concanamycin A (conA) for 2 h before imaging. Representative images of 10 replicates are shown. Scale bars, 5 µm. **(C)** Quantification of confocal experiments in B showing the ratio between the number of mCherry-ATG8E, TagRFP-NBR1, or mCherry-CFS1 puncta inside the vacuole, compared to the total number of mCherry-ATG8E, TagRFP-NBR1, or mCherry-CFS1 puncta. Bars indicate the mean ± SD of 10 replicates. Two-tailed and unpaired *t* tests with Welch's corrections were performed to analyze the differences of puncta numbers between mCherry-ATG8E and mCherry-CFS1 or between TagRFP-NBR1 and mCherry-CFS1. ***, P value <0.001. **(D)** Confocal microscopy images of *Arabidopsis* root epidermal cells co-expressing pUBQ::mCherry-ATG8E with either pUBQ::GFP-CFS1 or pNBR1::NBR1-GFP. 5-d-old *Arabidopsis* seedlings were incubated in 1/2 MS media containing either 150 mM NaCl (without conA; -conA), or 90 mM NaCl and 1 µM conA (+conA) for 2 h before imaging. Representative images of 10 replicates are shown. Area highlighted in the boxed region in the merge panel was further enlarged and presented in the inset panel. Scale bars, 10 µm. Inset scale bars, 5 µm. **(E)** Quantification of confocal experiments in D showing the mCherry-ATG8E colocalization ratio of NBR1-GFP and GFP-CFS1 to mCherry-ATG8E under +conA treatment conditions. The mCherry-ATG8E colocalization ratio is calculated as the ratio between the number of mCherry-ATG8E puncta that colocalize with NBR1-GFP or GFP-CFS1 puncta compared with the number of total mCherry-ATG8E puncta. Bars indicate the mean ± SD of 10 replicates. Two-tailed and unpaired *t* tests with Welch's corrections were performed to analyze the differences of mCherry-ATG8E colocalization ratio between GFP-CFS1 and NBR1-GFP. ***, P value <0.001. **(F)** TEM images showing immuno-gold labeled GFP-CFS1 at the autophagosomes in *Arabidopsis* root cells. 7-d-old seedlings were incubated in 150 mM NaCl-containing 1/2 MS media for 2 h for autophagy induction before cryofixation. Sections from pUBQ::GFP-CFS1 expressing samples were labeled with an anti-GFP primary antibody and a secondary antibody conjugated to 10 nm gold particles. Yellow arrowheads mark the gold particles associated with autophagosomes. Scale bars, 500 nm. AP, autophagosome. **(G)** Quantification of the localization of the GFP-specific gold particles imaged in the experiment shown in F. Approximately 900 gold particles in 50 TEM images captured from five independent samples were grouped into autophagosomes, cytosol or other organelles according to their locations. One-way ANOVA was performed to analyze the significant difference between different gold particle locations. ****, P value <0.0001.

screens, which revealed 51 confident interactors (Table S4). Notably, one of these interactors was VPS23A, an ESCRT-I complex component that is known to regulate endomembrane trafficking (Nagel et al., 2017; Shen et al., 2018; Gao et al., 2015). Confocal microscopy analyses showed that CFS1 colocalize with VPS23A under both control and salt-stressed conditions (Fig. 7, A and B). To further test the association between CFS1 and the ESCRT-I complex, we colocalized CFS1 with two other ESCRT-localized adaptor proteins, FREE1 and ALIX (Gao et al., 2015; Kalinowska et al., 2015). Similar to VPS23A, both proteins partially colocalized with CFS1 (Fig. S5, G and H). Interestingly, our airyscan and spinning disc microscopy analysis showed that two GFP-CFS1 puncta localized on distinct regions of VPS23A-TagRFP puncta (Fig. 7 C) and CFS1 move together with VPS23A (Video 3). These results suggest that CFS1 could bridge autophagosomes with VPS23A-labeled multivesicular bodies.

We then performed immunogold labeling TEM experiments to visualize the compartments where CFS1 and VPS23A colocalize. We used two differently sized gold particles conjugated to anti-GFP or anti-RFP antibodies, recognizing GFP-CFS1 and VPS23A-TagRFP, respectively. Electron micrographs obtained from cryo-fixed *Arabidopsis* root cells revealed that CFS1 and VPS23A colocalized at amphisomes, hybrid structures where autophagosomes fuse with intraluminal vesicle containing multivesicular bodies (Sanchez-Wandelmer and Reggiori, 2013; Fig. 7 D) This demonstrates that, similar to metazoans, plants have amphisomes where CFS1 colocalizes with VPS23A.

This prompted us to test the significance of amphisomes in autophagic flux. We first measured autophagic flux in *vps23* mutants. *Arabidopsis* has two VPS23 isoforms, VPS23A/VPS23.1 and VPS23B/VPS23.2 (Nagel et al., 2017). Since *Arabidopsis vps23avps23b* double mutants are lethal, we measured autophagic flux on single *vps23a* or *vps23b* mutants (Nagel et al., 2017). We expressed GFP-ATG8A in Col-0, *cfs1*, *vps23a*, and *vps23b* and quantified the GFP-ATG8A-marked autophagic bodies with and without salt stress and conA treatment. Neither *vps23a* nor *vps23b* mutants showed a significant difference compared to Col-0 (Fig. 7, E and F). Consistently, neither *vps23a* nor *vps23b* mutants

showed autophagic flux defects in NBR1 degradation assays (Fig. 7, G and H). Finally, *vps23a* and *vps23b* were indistinguishable from wild type in nitrogen starvation plate assays (Fig. 7 I). Altogether, these data indicate that VPS23A and VPS23B may act redundantly or do not play a role in autophagy.

To circumvent the potential redundancy between VPS23A and VPS23B, we mutated CFS1 residues likely to mediate the CFS1–VPS23A interaction. Previous studies have shown that VPS23 interacting proteins contain a "PSAPP" motif, which docks into the ubiquitin E2 variant domain on VPS23 (Pornillos et al., 2003; Sutipatanasomboon et al., 2017). Consistently, Alphafold prediction suggested CFS1 and VPS23A interacted in a PSAPP-dependent manner (Jumper and Hassabis, 2022; Fig. 8 A). We thus generated the mCherry-CFS1$^{PSAPP}$ mutant, where the PSAPP motif is mutated to alanines (PSAPP-145-AAAAA) and expressed it in the *cfs1* background. We also generated the mCherry-CFS1$^{AIM+PSAPP}$ double mutant, where both the AIM and the PSAPP motifs were mutated. The PSAPP mutation did not affect the stability of mCherry-CFS1 and its interaction with GFP-ATG8A (Fig. 8 B). We then tested whether the PSAPP motif is important for CFS1–VPS23A interaction. We stably co-expressed VPS23A-mTurquoise2 and GFP-ATG8A with mCherry-CFS1 or mCherry-CFS1$^{PSAPP}$. RFP pull-down experiments showed that mCherry-CFS1, mCherry-CFS1$^{PSAPP}$ had similar association levels with GFP-ATG8A. However, the PSAPP mutant associated significantly less with VPS23A (Fig. 8 C). Furthermore, confocal microscopy experiments on *Arabidopsis* lines co-expressing GFP-VPS23A with either mCherry-CFS1 or mCherry-CFS1$^{PSAPP}$ showed that mCherry-CFS1$^{PSAPP}$ did not colocalize with GFP-VPS23A anymore (Fig. 8, D and E). Of note, in several cases, mCherry-CFS1$^{PSAPP}$ puncta were in close proximity of the VPS23 puncta, but were unable to fully colocalize, consistent with the loss of association that we observed in the co-IP experiments (Fig. 8, C and D). Altogether, these findings suggest that CFS1 interacts with VPS23A in a PSAPP-dependent manner.

We next wanted to understand the physiological importance of the CFS1–VPS23A interaction in *Arabidopsis*. We measured

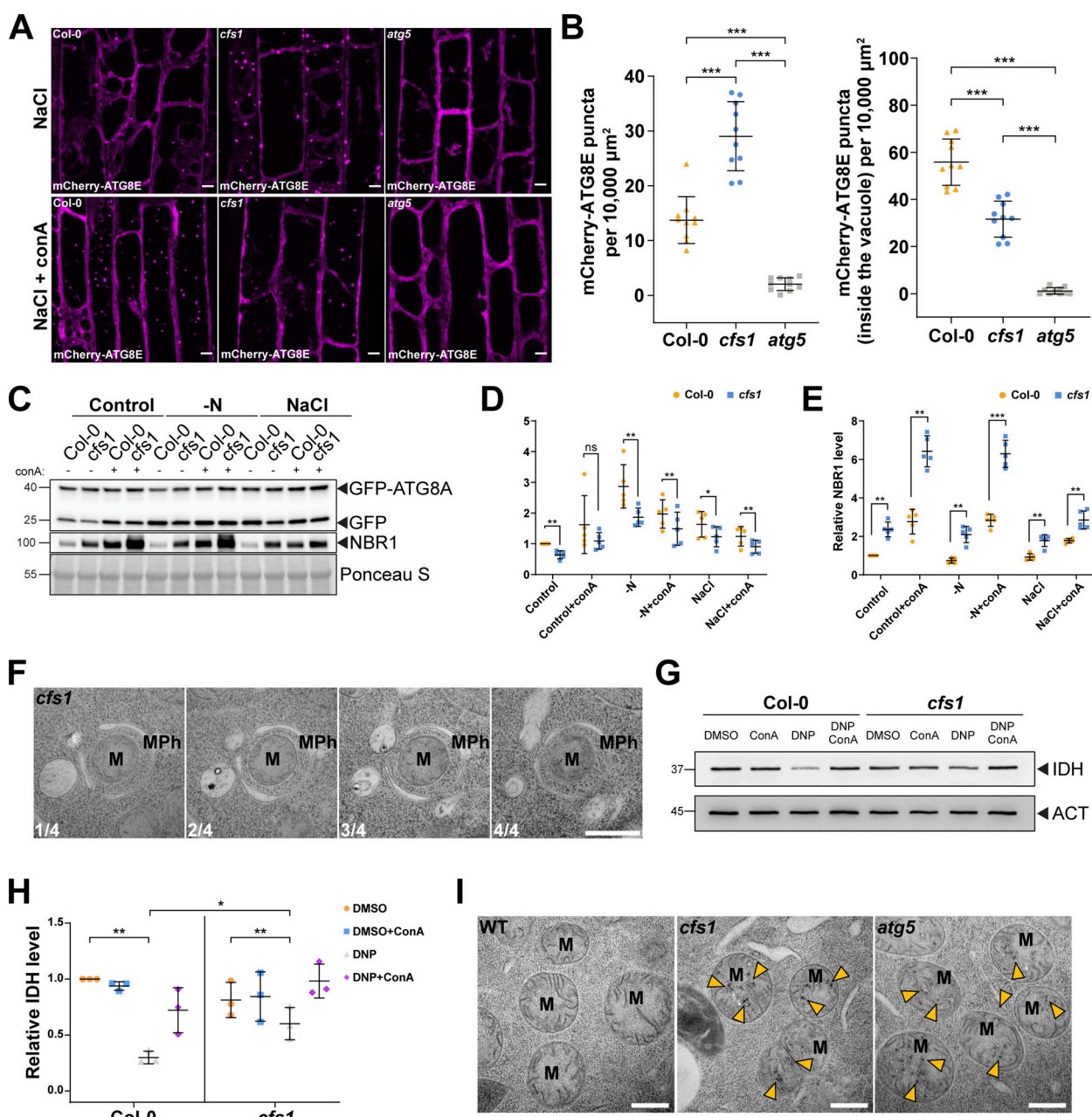

Figure 4. **CFS1 is crucial for autophagic flux in *A. thaliana*. (A)** Confocal microscopy images of root epidermal cells of Col-0, *cfs1*, or *atg5* expressing pUBQ::mCherry-ATG8E under NaCl or NaCl + conA treatment. 5-d-old *Arabidopsis* seedlings were incubated in 1/2 MS media containing either 150 mM NaCl (NaCl) or 90 mM NaCl and 1 μM conA (NaCl + conA) for 2 h before imaging. Representative images of 10 replicates are shown. Scale bars, 5 μm. **(B)** Left panel, quantification of the mCherry-ATG8E puncta per normalized area (10,000 μm²) of the NaCl-treated cells imaged in A. Right panel, quantification of the mCherry-ATG8E puncta inside the vacuole per normalized area (10,000 μm²) of the NaCl + conA-treated cells imaged in A. Bars indicate the mean ± SD of 10 replicates. Two-tailed and unpaired Student *t* tests with Welch's correction were performed to analyze the significance of mCherry-ATG8E puncta density differences between Col-0 and *cfs1*, Col-0 and *atg5*, or *cfs1* and *atg5*. ***, P value <0.001. **(C)** Western blots showing GFP-ATG8A cleavage level and endogenous NBR1 level in Col-0 or *cfs1* mutants under control, nitrogen-deficient (−N) or salt-stressed (NaCl) conditions. *Arabidopsis* seedlings were grown under continuous light in 1/2 MS media for 1-wk and 7-d-old seedlings were subsequently transferred to 1/2 MS media ±1 μM conA, nitrogen-deficient (−N) 1/2 MS media ±1 μM conA, or 1/2 MS media containing 150 mM NaCl ±1 μM conA for 12 h. 15 μg of total protein extract was loaded and immunoblotted with anti-GFP and anti-NBR1 antibodies. Representative images of five replicates are shown. Reference protein sizes are labeled as numbers at the left side of the blots (unit: kD). **(D)** Quantification of the relative autophagic flux in C. Values were calculated through protein band intensities of GFP divided by GFP-ATG8A and normalized to untreated (Control) Col-0. Results are shown as the mean ± SD of five replicates. One-tailed and paired Student *t* tests were performed to analyze the significance of the relative autophagic flux differences. ns, not significant. *, P < 0.05. **, P < 0.01. **(E)** Quantification of the relative NBR1 level in C compared to untreated (control) Col-0. Values were calculated through normalization of protein bands to Ponceau S and shown as the mean ± SD of five replicates. One-tailed and paired Student *t* tests were performed to analyze the significance of the relative NBR1 level differences. **, P < 0.01. ***, P < 0.001. **(F)** Serial sections of transmission electron microscopy micrographs showing a mitophagosome engulfing a mitochondrion. 7-d-old *Arabidopsis cfs1* seedlings were incubated in 1/2 MS media containing 50 μM DNP for 1 h before cryofixation. Scale bar, 500 nm. MPh, mitophagosome. M, mitochondrion. **(G)** Immunoblot

assay of uncoupler-induced mitochondrial protein degradation in *Arabidopsis* Col-0 and *cfs1* seedlings. 7-d-old seedlings were incubated in 1/2 MS media containing 50 µM DNP ±1 µM conA for 4 h before protein extraction. A mitochondrial matrix protein, isocitrate dehydrogenase (IDH), was immunoblotted by anti-IDH antibodies. Actin (ACT) was immunoblotted by anti-ACT antibodies and was used as a loading control. Representative images of three replicates are shown. Reference protein sizes are labeled as numbers at the left side of the blots (unit: kD). **(H)** Quantification of relative IDH intensities in G compared to untreated (control) Col-0. Values were calculated via normalization of protein bands to ACT and shown as the mean ± SD of three replicates. One-tailed and paired Student's *t* tests were performed to analyze the significance of the relative IDH level differences. ns, not significant. *, $P < 0.05$. **, $P < 0.01$. **(I)** TEM micrographs of mitochondria in *Arabidopsis* Col-0, *cfs1* and *atg5* root cells. 7-d-old seedlings were incubated in 150 mM NaCl-containing 1/2 MS media for 2 h for autophagy induction before cryofixation. Yellow arrowheads mark the distinctive electron dense precipitates of compromised mitochondria that appear after NaCl treatment. Scale bars, 500 nm. M, mitochondria. Source data are available for this figure: SourceData F4.

autophagic flux in mCherry-CFS1 and mCherry-CFS1[PSAPP] expressing *A. thaliana* lines, using three different assays. First, we quantified vacuolar GFP-ATG8A puncta under autophagy inducing salt stress conditions. mCherry-CFS1[PSAPP] lines had significantly fewer autophagic bodies in the vacuole (Fig. 8, G and H). Consistently, GFP-release assays showed that mCherry-CFS1[PSAPP] lines phenocopied the mCherry-CFS1[AIM] lines, with an autophagic flux defect under nitrogen starvation conditions (Fig. 8, I and J). NBR1 flux measurements also indicated a defect in autophagic flux (Fig. 8, I and K). Ultimately, to test the physiological importance of the CFS1–VPS23A interaction, we performed nitrogen starvation plate assays. Similar to the expression of mCherry-CFS1[AIM], the expression of mCherry-CFS1[PSAPP] failed to rescue the nitrogen starvation-sensitivity phenotype of *cfs1* (Fig. 8 L). Altogether, these findings demonstrate that the CFS1–VPS23A interaction is critical for autophagic flux.

## Discussion

Autophagosome maturation involves the trafficking and fusion of double-membraned autophagosomes with lytic compartments. Studies in yeast and metazoans have shown that this maturation step has several similarities with endocytic vesicle fusion and trafficking. Tethering complexes, RAB GTPases, SNARE proteins, and adaptors facilitate the trafficking and fusion of autophagosomes with the endolysosomal compartments (Zhao and Zhang, 2019; Zhao et al., 2021). While well studied in yeast and metazoans, a systematic analysis of autophagosome maturation is still missing in plants. In addition, plant genomes lack homologs of key maturation proteins, such as the autophagic SNARE, Syntaxin 17, suggesting plants have evolved different components for autophagosome maturation. Here, we show that CFS1 is an autophagy adaptor that bridges autophagosomes with multivesicular bodies and mediates the formation of amphisomes. We propose that CFS1 functions as a licensing factor that tethers autophagosomes to multivesicular bodies through the ESCRT-I complex. This is reminiscent of COPII tethering factor p115 that targets a subpopulation of COPII vesicles to cis-Golgi (Allan et al., 2000).

Previous studies in *Arabidopsis* showed that ESCRT-III subunits VPS2.1 and CHMP1A and B are involved in autophagy and regulate carbon starvation-induced bulk and selective autophagic degradation (Spitzer et al., 2015; Katsiarimpa et al., 2013). Similarly, in mammalians, ESCRT-III complex have been shown to mediate the closure of autophagosomes (Takahashi et al., 2018; Takahashi et al., 2019). Our findings now further extend

ESCRT-autophagy crosstalk and show that ESCRT-I subunits are also involved in autophagic degradation.

Our findings are consistent with a role of CFS1 in autophagosome maturation as opposed to autophagosome biogenesis, which was suggested recently (Kim et al., 2022): (i) our differential centrifugation-based autophagosome enrichment procedure selects for proteins that associate with closed, mature autophagosomes. AP-MS did not identify biogenesis components but did identify trafficking-related proteins such as Myosin 14 (Fig. 1 B and Table S3); (ii) we observe fully formed autophagosomes in micrographs obtained from *cfs1* mutants indistinguishable from and in similar numbers to wild type (Fig. 3 A and Fig. 4 F); and (iii) we observe a clear colocalization with the ESCRT-I protein VPS23A in both confocal and electron micrographs (Fig. 7). Although we see significant defects in bulk and selective autophagic flux in *cfs1*, CFS1[AIM], and CFS1[PSAPP] mutants, autophagic flux is not fully blocked in any of them. These findings suggest two, not mutually exclusive, explanations: (i) There could be other autophagy adaptors that mediate the maturation of a sub-population of autophagosomes. Consistently, in metazoans, there are several autophagy adaptors that are either involved in SNARE recruitment, Rab GTPase activation or autophagosome trafficking (Zhao et al., 2021). Performing autophagosome enrichment experiments in a GFP-ATG8A/*cfs1* line or using reporter lines of GFP-ATG8G, H or I that do not interact with CFS1 could reveal these adaptors. (ii) Some autophagosomes may be trafficked directly to the vacuole without an intermediary amphisome step. A systematic characterization of the autophagosome fusion machinery would shed light on both of these possibilities.

There are multiple trafficking routes to the vacuole: endocytic trafficking, post-Golgi trafficking, ER to vacuole transport, and autophagy (Aniento et al., 2022). Although we are starting to understand how these routes operate individually, how distinct vacuolar trafficking pathways are coordinated in response to changes in metabolic demands and external stimuli remains elusive. Here, based on our findings, we propose that vacuolar trafficking is organized as a "hub and spoke" type distribution system, where amphisomes serve as sorting hubs for multivesicular bodies and autophagosomes (Fig. 9). The "hub and spoke" model was developed by Delta Airlines in the 1950s and has since been successfully employed by various types of supply chain logistics and aviation industry. The model implies that, rather than transportation taking place from A to B as in the point-to-point system, all materials transit through centralized hubs. It thereby reduces logistical costs as fewer routes are necessary. It also permits economies of scale, since complicated

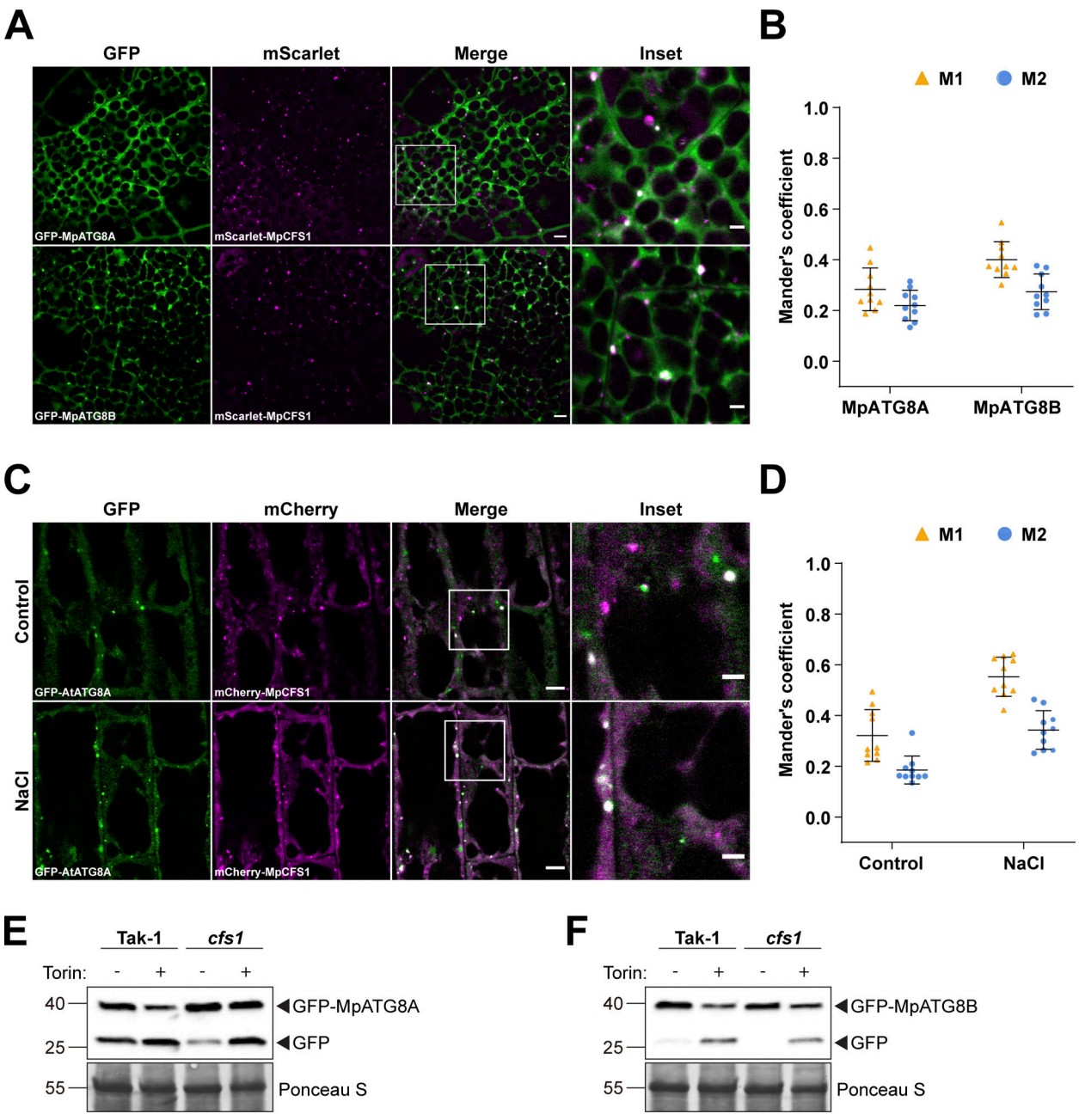

**Figure 5. CFS1 function is conserved in *M. polymorpha*. (A)** Confocal microscopy images of *M. polymorpha* thallus cells co-expressing pEF1::mScarlet-MpCFS1 with either pEF1::GFP-MpATG8A or pEF1::GFP-MpATG8B. 2-d-old thalli were incubated in 1/2 Gamborg B5 media before imaging. Representative images of 10 replicates are shown. Area highlighted in the white-boxed region in the merge panel was further enlarged and presented in the inset panel. Scale bars, 5 µm. Inset scale bars, 2 µm. **(B)** Quantification of confocal experiments in A showing the Mander's colocalization coefficients between mScarlet-MpCFS1 and GFP-fused MpATG8A or MpATG8B. M1, fraction of the GFP-fused MpATG8A or MpATG8B signals that overlaps with mScarlet-MpCFS1 signal. M2, fraction of mScarlet-MpCFS1 signal that overlaps with GFP-fused MpATG8A or MpATG8B signals. Bars indicate the mean ± SD of 10 replicates. **(C)** Confocal microscopy images of *Arabidopsis* root epidermal cells co-expressing pUBQ::mCherry-MpCFS1 and pUBQ::GFP-AtATG8A. 5-d-old seedlings were incubated in either control or 150 mM NaCl-containing 1/2 MS media for 1 h before imaging. Representative images of 10 replicates are shown. Scale bars, 5 µm. Inset scale bars, 2 µm. **(D)** Quantification of confocal experiments in (C) showing the Mander's colocalization coefficients between mCherry-MpCFS1 and GFP-AtATG8A under control or salt-stressed (NaCl) conditions. M1, fraction of the GFP-AtATG8A signal that overlaps with the mCherry-MpCFS1 signal. M2, fraction of the mCherry-MpCFS1 signal that overlaps with the GFP-AtATG8A signal. Bars indicate the mean ± SD of 10 replicates. **(E)** GFP cleavage assay of pEF1::GFP-MpATG8A in *M. polymorpha* wild type (Tak-1) or *cfs1* mutants. 10-d-old propagules were treated with 12 µM Torin for 5 h before protein extraction. 15 µg of total protein extract was loaded and immunoblotted with anti-GFP antibodies. Representative images of two replicates are shown. Reference protein sizes are labeled as numbers at the left side of the blots (unit: kD). **(F)** GFP cleavage assay of pEF1::GFP-MpATG8B in *M. polymorpha* wild type (Tak-1) or *cfs1* mutants. 10-d-old propagules were treated with 12 µM Torin for 5 h before protein extraction. 15 µg of total protein extract was loaded and was immunoblotted with anti-GFP antibodies. Representative images of two replicates are shown. Reference protein sizes are labeled as numbers at the left side of the blots (unit: kD). Source data are available for this figure: SourceData F5.

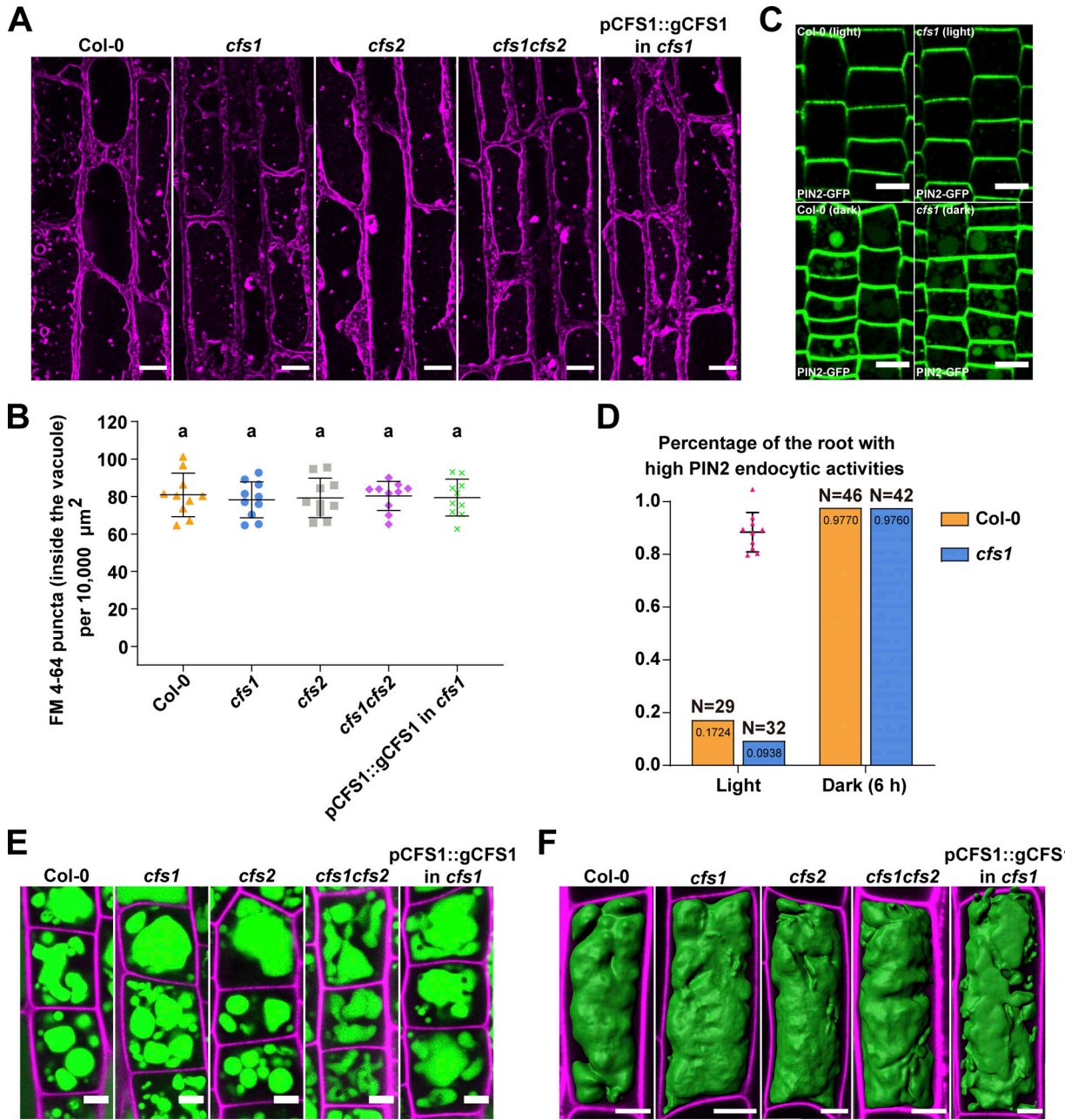

Figure 6. **Endocytic trafficking or vacuolar morphology is not affected in *cfs1* mutants. (A)** Confocal microscopy images of *Arabidopsis* root epidermal cells of Col-0, *cfs1*, *cfs2*, *cfs1cfs2*, and *cfs1* complemented with pCFS1::gCFS1(pCFS1::gCFS1 in *cfs1*). 5-d-old *Arabidopsis* seedlings were first incubated in 4 µM FM 4-64-containing 1/2 MS media for 30 min and then transferred to 1 µM concanamycin A-containing 1/2 MS media for 2 h before imaging. Representative images of 10 replicates are shown. Scale bars, 10 µm. **(B)** Quantification of the FM 4-64 stained puncta inside the vacuole per normalized area (10,000 µm²) of the cells imaged in A. Bars indicate the mean ± SD of 10 replicates. One-way ANOVA tests were performed to analyze the differences of the number of FM 4-64 stained puncta between each group. Tukey's multiple comparison tests were used for multiple comparisons. Family-wise significance and confidence level, 0.05 (95% confidence interval). **(C)** Representative microscopy images showing PIN2 endocytosis in the epidermal cells in the root tip meristem region of Col-0 and *cfs1* under light or 6 h dark conditions. 5-d-old *Arabidopsis* seedlings expressing pPIN2::PIN2-GFP were grown on 1/2 MS media plates (+1% plant agar) under light or 6 h dark conditions before imaging. Scale bars, 10 µm. **(D)** Quantification of PIN2 endocytic activities in Col-0 and *cfs1* shown in C. The *Arabidopsis* seedlings with at least five root epidermal cells that contained visible PIN2-GFP in the vacuole were considered as high PIN2 endocytic activities. The percentage of Col-0 and *cfs1* with high PIN2 endocytic activities under light or 6-h dark conditions are shown in the graph. Numbers inside the bars represent the exact value (4 decimals) of each bar. N represents the total number of the Col-0 or *cfs1* seedlings used for imaging and quantification in three independent experiments. **(E)** Confocal microscopy images showing the BCECF-AM-stained root epidermal cells in the meristem region of Col-0, *cfs1*, *cfs2*, *cfs1cfs2*, or pCFS1::gCFS1 in *cfs1*. 5-d-old *Arabidopsis* seedlings were incubated in 1/2 MS media containing 5 µM BCECF-AM for 30 min before imaging. Samples were mounted on slides with 0.002 mg/ml propidium iodide. Representative images of three replicates are shown. Green signals indicate the BCECF-AM-stained vacuole. Magenta signals indicate the propidium iodide-stained cell wall. Scale bars, 5 µm. **(F)** Three-dimensional images showing the vacuolar structure of the BCECF-AM-stained root epidermal cells in the transition region of Col-0, *cfs1*, *cfs2*, *cfs1cfs2*, or pCFS1::gCFS1 in *cfs1*. 5-d-old *Arabidopsis* seedlings were incubated in 1/2 MS media containing 5 µM BCECF-AM for 30 min before imaging. Samples were mounted on slides with 0.002 mg/ml propidium iodide. Representative images of three replicates are shown. Green signals indicate the BCECF-AM-stained vacuole. Magenta signals indicate the propidium iodide-stained cell wall. Scale bars, 10 µm.

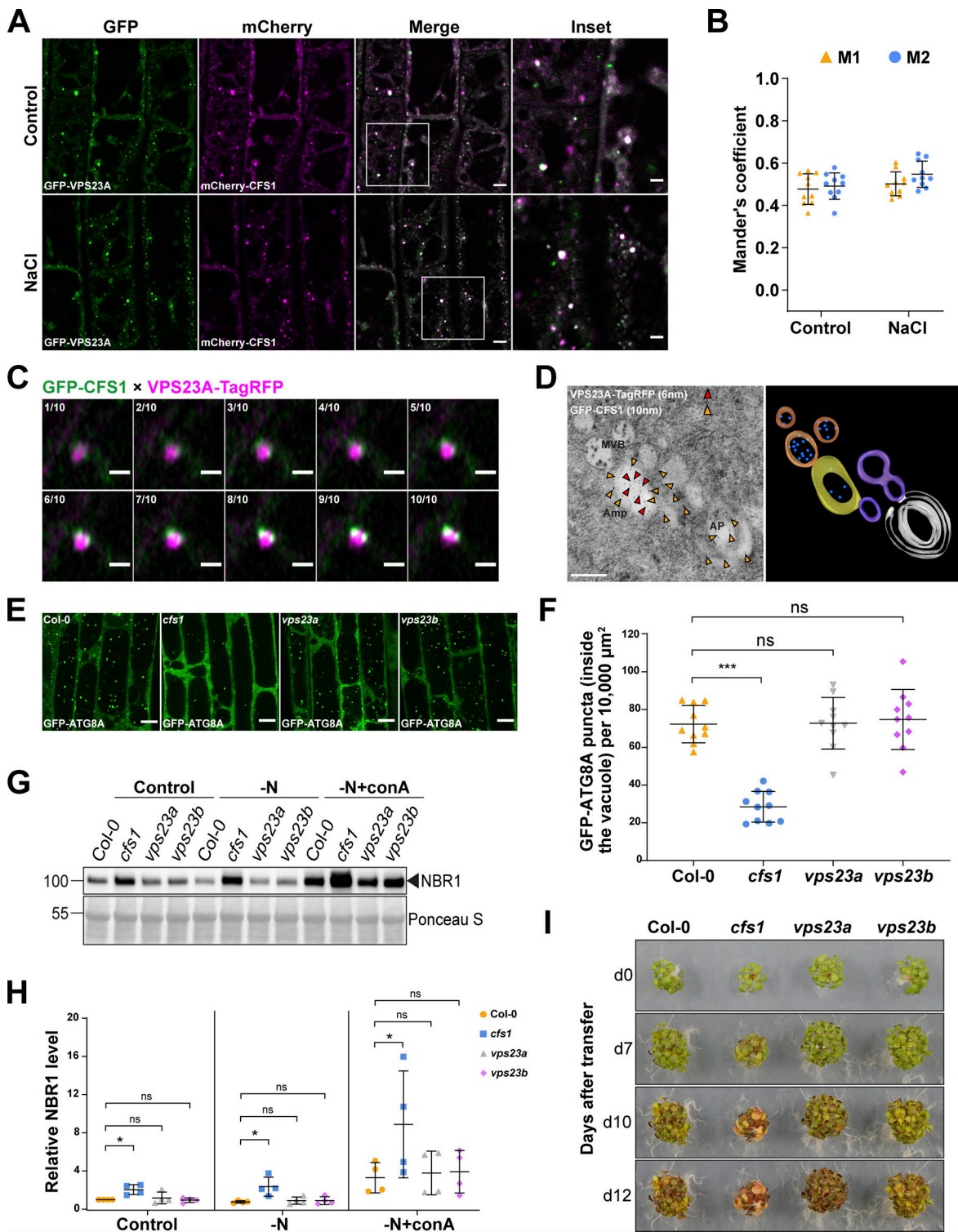

Figure 7. **Functional analysis of CFS1–VPS23 interaction revealed the amphisome in _A. thaliana_. (A)** Confocal microscopy images of _Arabidopsis_ root epidermal cells co-expressing pUBQ::mCherry-CFS1 with pUBQ::GFP-VPS23A. 5-d-old _Arabidopsis_ seedlings were incubated in either control or 150 mM NaCl-containing 1/2 MS media for 1 h before imaging. Representative images of 10 replicates are shown. Area highlighted in the white-boxed region in the merge panel was further enlarged and presented in the inset panel. Scale bars, 5 µm. Inset scale bars, 2 µm. **(B)** Quantification of confocal experiments in A showing the Mander's colocalization coefficients between mCherry-CFS1 and GFP-VPS23A. M1, fraction of GFP-VPS23A signal that overlaps with mCherry-CFS1 signal. M2, fraction of mCherry-CFS1 signal that overlaps with GFP-VPS23A signal. Bars indicate the mean ± SD of 10 replicates. **(C)** Airyscan time-lapse microscopy images of _Arabidopsis_ root epidermal cells showing the partial colocalization between GFP-CFS1 and VPS23A-TagRFP. 5-d-old _Arabidopsis_ seedlings co-expressing pVPS23A::VPS23A-TagRFP and pUBQ::GFP-CFS1 were incubated in 150 mM NaCl-containing 1/2 MS media for 1 h for autophagy induction before imaging. 10 continuous layers from one Z-stack image are shown separately in order. Interval of the Z-stack image, 0.15 µm. Scale bars, 1 µm. **(D)** Left panel, transmission electron microscopy micrographs showing the colocalization of VPS23A-TagRFP and GFP-CFS1 in _Arabidopsis_ root cells co-expressing

pVPS23A::VPS23A-TagRFP and pUBQ::GFP-CFS1. 5-d-old seedlings were incubated in 150 mM NaCl-containing 1/2 MS media for 2 h for autophagy induction before cryofixation. Sections were labeled with rabbit anti-RFP and chicken anti-GFP primary antibodies and secondary antibodies conjugated to 6- or 10-nm gold particles. Yellow and red arrowheads mark the representative RFP and GFP-specific gold particles, respectively. MVB, multivesicular bodies; Amp, amphisome; AP, autophagosome. Scale bars, 400 nm. Right panel, the three-dimensional model of the amphisome and its associated compartments shown in the left panel. The model was generated from serial sections flanking the section shown in the left panel. Autophagosome (white), autophagosome-associated vesicles (purple), amphisome (yellow), multivesicular bodies (orange), and internal vesicles (blue) were rendered into 3D surfaces. **(E)** Confocal microscopy images of root epidermal cells of Col-0, *cfs1*, *vps23a*, or *vps23b* seedlings expressing pUBQ::GFP-ATG8A under NaCl + concanamycin A (conA) treatment. 5-d-old *Arabidopsis* seedlings were incubated in 1/2 MS media containing 90 mM NaCl and 1 µM conA for 2 h before imaging. Representative images of 10 replicates are shown. Scale bars, 10 µm. **(F)** Quantification of the number of GFP-ATG8A puncta inside the vacuole per normalized area (10,000 µm²) of the cells imaged in E. Bars indicate the mean ± SD of 10 replicates. Two-tailed and unpaired Student *t* tests were performed to analyze the significance of GFP-ATG8A puncta density differences between Col-0 and *cfs1*, Col-0 and *vps23a*, or Col-0 and *vps23b*. ns, not significant. ***, P value <0.001. **(G)** Western blot showing the endogenous NBR1 level in Col-0, *cfs1*, *vps23a*, or *vps23b* under control or nitrogen-deficient ± conA conditions. *Arabidopsis* seeds were first grown in 1/2 MS media under continuous light for 1-wk and 7-d-old seedlings were subsequently transferred to 1/2 MS media (Control), nitrogen-deficient 1/2 MS media (−N) or nitrogen-deficient 1/2 MS media containing 1 µM conA (−N + conA) for 12 h. 10 µg of total protein extract was loaded and immunoblotted with anti-NBR1 antibodies. Representative images of four replicates are shown. Reference protein sizes are labeled as numbers at the left side of the blots (unit: kD). **(H)** Quantification of the relative NBR1 level in G compared to untreated (control) Col-0. Values were normalized to untreated (Control) Col-0 and calculated through normalization of protein bands to Ponceau S and shown as the mean ± SD of four replicates. One-tailed and paired Student *t* tests were performed to analyze the significance of the relative NBR1 level difference. ns, not significant. *, P < 0.05. **(I)** Phenotypic characterization of *Arabidopsis cfs1*, *vps23a*, and *vps23b* mutants during nitrogen starvation. 25 seeds per genotype were grown on 1/2 MS media plates (+1% plant agar) for 1-wk and 7-d-old seedlings were subsequently transferred to nitrogen-deficient 1/2 MS media plates (+0.8% plant agar) and grown for 2 wk. Plants were grown at 21°C under LEDs with 85 µM/m²/s and a 14 h light/10 h dark photoperiod. d0 depicts the day of transfer. Brightness of pictures was enhanced ≤30% with Adobe Photoshop (2020). Representative images of four replicates are shown. Source data are available for this figure: SourceData F7.

operations can be performed in the hubs rather than separately organized in each node (Oti, 2013). Organizing vacuolar trafficking through hubs (i) could allow the cell to intricately balance the anabolic and catabolic needs and seamlessly integrate various intrinsic and extrinsic signals, (ii) could facilitate the crosstalk between post-Golgi trafficking, endocytosis, and autophagy, and (iii) could provide a route for loading and secretion of extracellular vesicles.

One well-known drawback of "hub and spoke" distribution models is that the hub represents a single point of failure. Any congestion in the hub can severely impact the whole system. Therefore, building a robust hub is critical for a successful supply chain (Oti, 2013). This could be informative for translational approaches that aim to increase plant stress tolerance by engineering plant endomembrane trafficking.

### Speculation

One of our time-lapse videos analyzing CFS1-NBR1 co-trafficking showed an intriguing pattern: A CFS1-NBR1 positive punctum squeezed through a membrane, which looks like movement from one cell to another, and separated from each other after moving to the adjacent cell (Video 4). We would like to share it with the community and discuss the implications of cell-to-cell movement of autophagosomes: (i) In *Arabidopsis* roots, central vacuoles mature and acidify by fusion of small vacuoles in a developmentally regulated manner (Cui et al., 2019; Cui et al., 2020; Krüger and Schumacher, 2018). So, in meristematic cells, vacuoles may not be hydrolytically active. Autophagosomes in such cells may need to traffic to the neighboring cells to deliver autophagic cargo to an active lytic compartment. Intriguingly, we observed that mCherry-CFS1 highly accumulates in the quiescent center cells of *Arabidopsis*, where vacuoles have not matured yet (Fig. 10). (ii) Alternatively, they may be loaded with cargo that are involved in cell-to-cell communication. In both cases, the amphisomes formed through CFS1–VPS23A interaction may act as the "porter" role (Fig. 9). However, as the title of

the section implies, these scenarios need to be supported or refuted by further studies.

## Materials and methods

### Plant material and cloning procedure

All *Arabidopsis thaliana* lines used in this study originate from the Columbia (Col-0) ecotype and are listed below (Table 1). The primers used for genotyping are listed in Table 3. Coding sequences from gene of interest were amplified from Col-0 cDNA with primers listed in Table 3. Plasmids were assembled through the GreenGate cloning method (Lampropoulos et al., 2013). In short, CFS1 (At3g43230) and CFS2 (At1g29800) were cloned into two parts to remove internal *Bsa*I sites (see Table 3 for primers). For introducing point mutations in mCherry-CFS1$^{AIM}$ (WLNL-267-ALNA), mCherry-CFS1$^{FYVE}$ (RHHCR-195-AHACA), and mCherry-CFS1$^{PSAPP}$ (PSAPP-145-AAAAA), site-directed mutagenesis was applied using primers listed in Table 3. For mCherry-CFS1$^{SYLF}$ (K282A, R288A, K320A), the SYLF domain was mutagenized by ordering a synthetic DNA sequence carrying point mutations (Table 3) and replaced by restriction enzyme digestion with *Nco*I (NEB) and *Xba*I (NEB). All constructs used in this study are listed in Table 2. Transgenic plant lines were generated through the Agrobacterium-mediated floral-dip method (Clough and Bent, 1998) and are listed in Table 1.

For *M. polymorpha*, all lines are derived from the Takaragaike-1 (Tak-1) ecotype. Plants were maintained and cultivated asexually on half-strength (1/2) Gamborg B5 media (Gamborg B5 medium basal salt mixture [Duchefa] supplemented with 0.5 g/l MES and 1% sucrose, pH 5.5) plates (+1% plant agar [Duchefa]) under continuous white light with a light intensity of 50 µM/m²/s at 21°C. For cloning, MpATG8A (Mp1g21590) and MpATG8B (Mp5g05930) primers are listed in Table 3. For MpCFS1 (Mp3g11370), a synthetic DNA construct (Table 3) with Green-Gate compatible *Bsa*I overhangs was ordered. Plasmids were assembled through the Gateway (Ishizaki et al., 2015) or

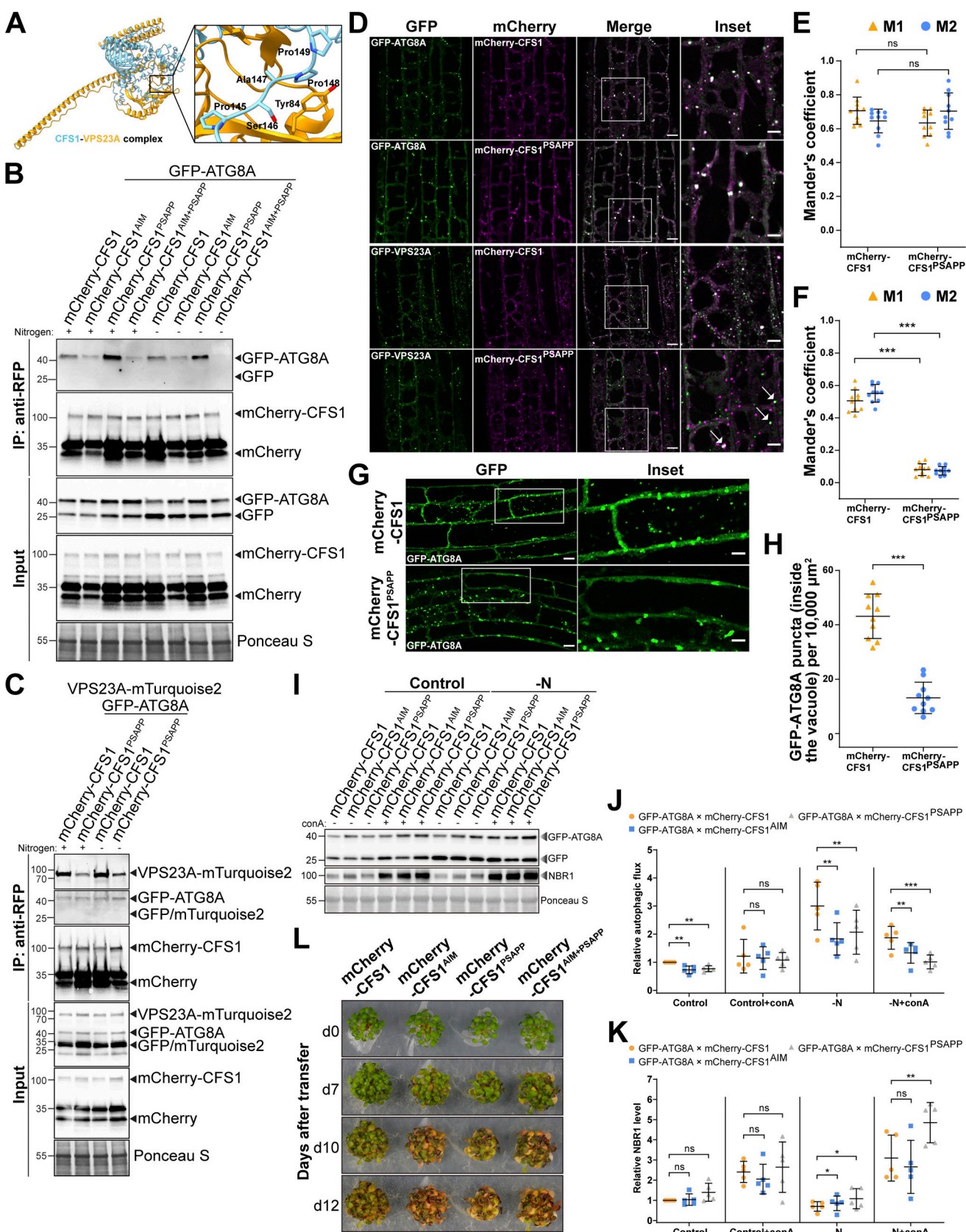

Figure 8. **PSAPP motif of CFS1 is crucial for CFS1-VPS23A interaction and autophagic flux. (A)** Homology modeling of CFS1/VPS23A complex. Prediction of CFS1-VPS23A heterocomplex formation generated by AlphaFold2 as implemented by ColabFold (Jumper et al., 2021; Mirdita et al., 2022). Structure of CFS1 and VPS23A is represented as ribbons and colored in light blue and orange, respectively. The predicted complex interaction interface involving AtCFS1 PSAPP motif is highlighted as a zoom in with the side chains of relevant residues represented as sticks. **(B)** RFP-Trap pull-down of *Arabidopsis* seedlings co-expressing

pUBQ::GFP-ATG8A with either pUBQ::mCherry-CFS1, pUBQ::mCherry-CFS1^AIM, pUBQ::mCherry-CFS^PSAPP or pUBQ::mCherry-CFS1^AIM+PSAPP. 7-d-old seedlings were incubated in either control (+) or nitrogen-deficient (−) 1/2 MS media for 12 h. Protein extracts were immunoblotted with anti-GFP and anti-RFP antibodies. Representative images of two replicates are shown. Reference protein sizes are labeled as numbers at the left side of the blots (unit: kD). **(C)** RFP-Trap pull-down of *Arabidopsis* seedlings co-expressing pUBQ::GFP-ATG8A and pUBQ::VPS23A-mTurquoise2 with either pUBQ::mCherry-CFS1 or pUBQ::mCherry-CFS1^PSAPP. 7-d-old seedlings were incubated in either control (+) or nitrogen-deficient (−) 1/2 MS media for 12 h. Protein extracts were immunoblotted with anti-GFP and anti-RFP antibodies. Representative images of two replicates are shown. Reference protein sizes are labeled as numbers at the left side of the blots (unit: kD). **(D)** Confocal microscopy images of *Arabidopsis* root epidermal cells co-expressing either pUBQ::GFP-ATG8A or pUBQ::GFP-VPS23A with either pUBQ::mCherry-CFS1 or pUBQ::mCherry-CFS1^PSAPP under salt stress. 5-d-old *Arabidopsis* seedlings were incubated in 150 mM NaCl-containing media for 1 h for autophagy induction before imaging. Representative images of 10 replicates are shown. Area highlighted in the white-boxed region in the merge panel was further enlarged and presented in the inset panel. Arrows point out the partial colocalization of GFP-VPS23A puncta and mCherry-CFS1^PSAPP puncta. Scale bars, 10 µm. Inset scale bars, 5 µm. **(E)** Quantification of confocal experiments in (D) showing the Mander's colocalization coefficients between GFP-ATG8A and either mCherry-CFS1 or mCherry-CFS1^PSAPP. M1, fraction of GFP-ATG8A signal that overlaps with mCherry-CFS1 or mCherry-CFS1^PSAPP signal. M2, fraction of mCherry-CFS1 or mCherry-CFS1^PSAPP signal that overlaps with GFP-ATG8A signal. Bars indicate the mean ± SD of 10 replicates. **(F)** Quantification of confocal experiments in D showing the Mander's colocalization coefficients between GFP-VPS23A and either mCherry-CFS1 or mCherry-CFS1^PSAPP. M1, fraction of GFP-VPS23A signal that overlaps with mCherry-CFS1 or mCherry-CFS1^PSAPP signal. M2, fraction of mCherry-CFS1 or mCherry-CFS1^PSAPP signal that overlaps with GFP-VPS23A signal. Bars indicate the mean ± SD of 10 replicates. **(G)** Confocal microscopy images of *Arabidopsis* root epidermal cells co-expressing pUBQ::GFP-ATG8A with either pUBQ::mCherry-CFS1 or pUBQ::mCherry-CFS1^PSAPP under NaCl + concanamycin A (conA) treatment. 5-d-old *Arabidopsis* seedlings were incubated in 1/2 MS media containing 90 mM NaCl and 1µM conA for 2 h before imaging. Scale bars, 10 µm. Inset scale bars, 5 µm. **(H)** Quantification of GFP-ATG8A puncta inside the vacuole per normalized area (10,000 µm²) of the cells imaged in G. Bars indicate the mean ± SD of 10 replicates. Two-tailed and unpaired Student's *t* test were performed to analyze the significance of difference between mCherry-CFS1 and mCherry-CFS1^PSAPP. ***, P value <0.001. **(I)** Western blots showing the GFP-ATG8A cleavage level and the endogenous NBR1 level in *Arabidopsis cfs1* mutants co-expressing pUBQ::GFP-ATG8A with either pUBQ::mCherry-CFS1, pUBQ::mCherry-CFS1^AIM, or pUBQ::mCherry-CFS1^PSAPP under control ± conA or nitrogen-deficient (−N) ± conA conditions. *Arabidopsis* seeds were first grown in 1/2 MS media under continuous light for 1-wk and 7-d-old seedlings were subsequently transferred to 1/2 MS media ±1 µM conA or nitrogen-deficient 1/2 MS media ±1 µM conA for 12 h. 10 µg of total protein extract was loaded and immunoblotted with anti-GFP and anti-NBR1 antibodies. Representative images of five replicates are shown. Reference protein sizes are labeled as numbers at the left side of the blots (unit: kD). **(J)** Quantification of I showing the relative autophagic flux. Values were calculated as the protein band intensities of GFP divided by the protein band intensity of GFP-ATG8A and were normalized to untreated (Control) Col-0. Results are shown as the mean ± SD of five replicates. One-tailed and paired Student's *t* tests were performed to analyze the significance of the relative autophagic flux differences. ns, not significant. **, P < 0.01. ***, P < 0.001. **(K)** Quantification of I showing the relative NBR1 level in respect to untreated (Control) Col-0. Values were calculated through normalization of protein bands to Ponceau S and shown as the mean ± SD of five replicates. One-tailed and paired Student *t* tests were performed to analyze the significance of the relative NBR1 level difference. ns, not significant. *, P < 0.05. **, P < 0.01. **(L)** Phenotypic characterization of *Arabidopsis cfs1* mutants complemented with either pUBQ::mCherry-CFS1, pUBQ::mCherry-CFS1^AIM, pUBQ::mCherry-CFS1^PSAPP, or pUBQ::mCherry-CFS1^AIM+PSAPP upon nitrogen starvation. 25 seeds per genotype were grown on 1/2 MS media plates (+1% plant agar) for 1-wk and 7-d-old seedlings were subsequently transferred to nitrogen-deficient 1/2 MS media plates (+0.8% plant agar) and grown for 2 wk. Plants were grown at 21°C under LEDs with 85 µM/m²/s and a 14 h light/10 h dark photoperiod. d0 depicts the day of transfer. Brightness of pictures was enhanced ≤19% with Adobe Photoshop (2020). Representative images of four replicates are shown. Source data are available for this figure: SourceData F8.

GreenGate (Lampropoulos et al., 2013) system and introduced through *Agrobacterium tumefaciens* transformation (Kubota et al., 2013). *cfs1* knockout mutants were achieved through CRISPR/Cas9 gene editing through 2 gRNAs. gRNAs were inserted at the *Bsa*I (NEB) sites of the pMpGE_En03 entry vector and subsequently inserted into the pMpGE010 destination vector through LR clonase II reaction (Invitrogen; Sugano et al., 2018). Successful transformants were verified through sequencing of PCR products from amplified genomic DNA. Stable plant lines and constructs used are listed in Tables 4 and 2, respectively.

### Plant phenotypic assays

*Arabidopsis* seedlings were grown as described in Jia et al. (2019). Briefly, 25 *Arabidopsis* seeds per bundle were vapor-phase sterilized (90% sodium hypochlorite 13 and 10% HCl 36%) and sown on plates which were layered with a thin nylon mesh on top of 1/2 MS media (Murashige and Skoog salt + Gamborg B5 vitamin mixture [Duchefa] supplemented with 0.5 g/liter MES and 1% sucrose, pH 5.7) plates (with 1% plant agar [Duchefa]) followed by 3 d of vernalization at 4°C in dark. Vernalized seeds were grown at 21°C under LEDs with 85 µM/m²/s with a 14 h light/10 h dark photoperiod for 7 d. 7-d-old seedlings were subsequently transferred to 1/2 MS media (CaissonLabs) or nitrogen-

deficient 1/2 MS media (Murashige and Skoog salt without nitrogen [CaissonLabs] + Gamborg B5 vitamin mixture [Duchefa] supplemented with 0.5 g/liter MES and 1% sucrose, pH 5.7) plates (+0.8% plant agar [Duchefa]) and grown for 2 wk.

### Sample preparation before protein extraction

Unless stated otherwise, 20–40 *A. thaliana* seeds were grown in 1/2 MS media in 12-well plates under continuous light and constant shaking at 80 rpm for 7 d. 7-d-old seedlings were subjected to indicated treatments. For nitrogen starvation, 1/2 MS media was replaced with nitrogen-deficient 1/2 MS media. For salt stress, 150 mM NaCl was added to 1/2 MS media. For drug-treatments, all drugs used were dissolved in DMSO and added to the desired concentration (3 µM Torin1 (CAS 1222998-36-8; Santa Cruz); 1 µM concanamycin A (conA; CAS 80890-47-7; Santa Cruz); and 50 µM 2,4-dinitrophenol (DNP; CAS 51-28-5; Sigma-Aldrich). Equal amount of pure DMSO was added to control samples. Seedlings were harvested in microcentrifugation tubes with different-sized glass beads (2.85–3.45, 1.7–2.1, and 0.75–1.00 mm; Lactan GmbH) and flash-frozen in liquid nitrogen. Plants were ground with a mixer mill MM400 (3 × 30 s, 30 Hz; Retsch).

For protein extraction of *M. polymorpha*, the propagules were grown for 10 d in 1/2 Gamborg B5 media under continuous light

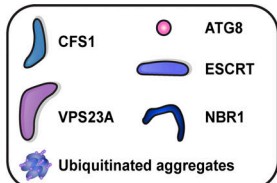

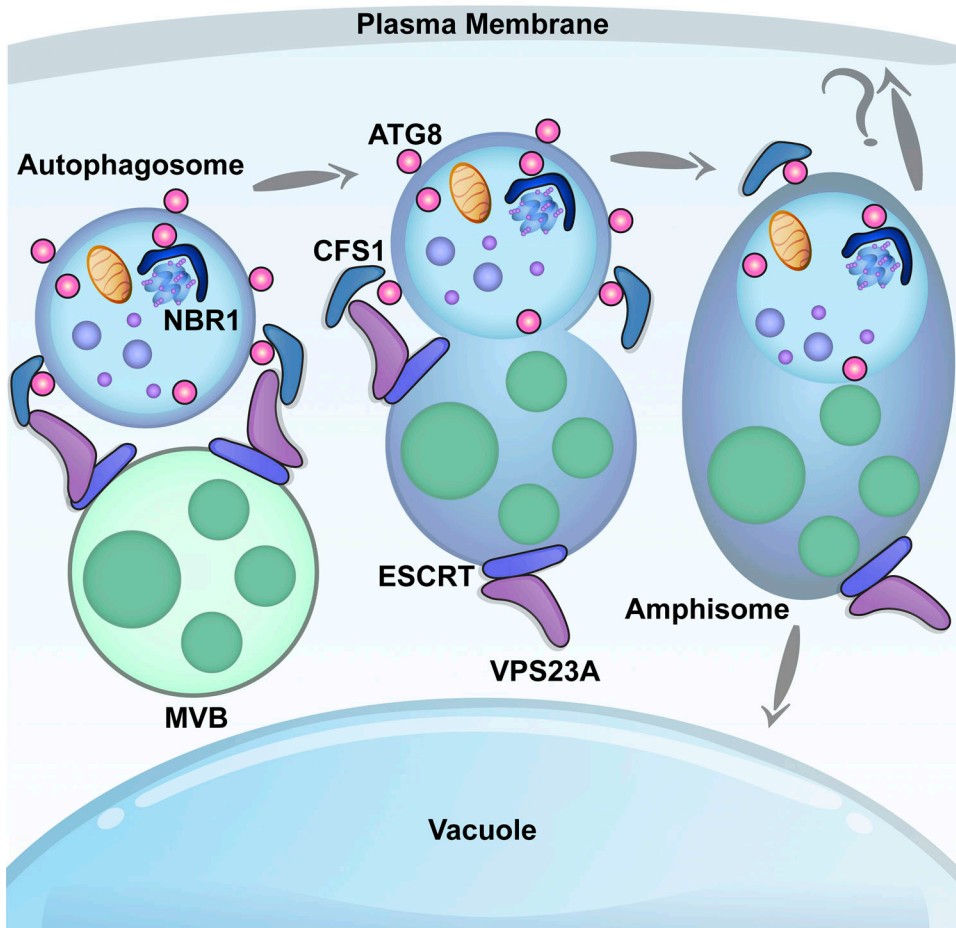

**Figure 9. Current working model: Vacuolar trafficking is organized as a "hub and spoke" type distribution system.** CFS1 interacts with both ATG8 and VPS23A and mediates the formation of amphisomes—hybrid prevacuolar compartments that are formed by the fusion of multivesicular bodies and autophagosomes.

with a light intensity of 50 µM/m²/s at 21°C and subjected to indicated treatments. Samples were flash-frozen in liquid nitrogen and ground using a mortar and pestle.

**Western blotting**

*Arabidopsis* protein extraction was achieved by adding 250 µl of protein extraction buffer (100 mM Tris [pH 7.5], 200 mM NaCl, 1 mM EDTA, 2% 2-Mercaptoethanol, 0.2% Triton X-100 and 1 tablet/50 ml Complete, EDTA-free Protease Inhibitor Cocktail [Sigma-Aldrich], pH 7.8) to grinded samples. Samples were subsequently well mixed and centrifuged at 15,000 rpm at 4°C for 10 min. The whole supernatant was transferred to a new tube. After another round of centrifugation at 15,000 rpm at 4°C for 10 min, resulting supernatant was diluted with 2× Laemmli buffer (4% SDS, 20% glycerol, 10% 2-mercaptoethanol, 0.004%

bromophenol blue and 0.125 M Tris HCl, pH 6.8) and boiled at 95°C for 10 min as the final protein extract sample.

For *M. polymorpha* protein extraction, grinded plant powder was weighed and added with 2× volume of buffer (10% glycerol, 25 mM Tris [pH 7.5], 150 mM NaCl, 1 mM EDTA, 2% PVPP, 1 mM DTT, 0.2% Nonidet P-40/Igepal and 1 tablet/50 ml cOmplete, EDTA-free Protease Inhibitor Cocktail [Sigma-Aldrich]) for lysing. Protein lysates were cleared through centrifuging samples at 12,000 rpm twice. Resulting supernatant was diluted in 4× Laemmli buffer and boiled at 70°C for 10 min as the final protein extract sample.

For *Arabidopsis*, protein concentration was measured using the Amido black method (Popov et al., 1975). 10 µl of protein sample was diluted in 190 µl water and added with 1 ml Amido black staining solution (90% methanol, 10% acetic acid, 0.005%

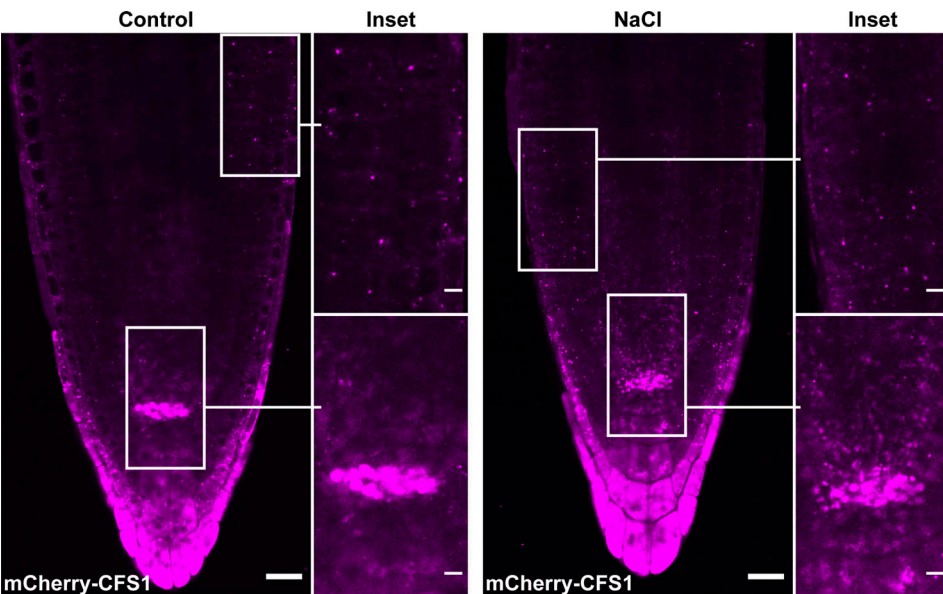

**Figure 10. CFS1 accumulates at the quiescent center cells of *A. thaliana*.** Confocal microscopy images showing that pUBQ::mCherry-CFS1 strongly accumulates at the quiescent center cells of *A. thaliana*. 5-d-old *Arabidopsis* seedlings were incubated in either control or 150 mM NaCl-containing 1/2 MS media for 1 h before imaging. Representative images of three replicates are shown. Area highlighted in the white-boxed region was further enlarged and presented in the inset panel. Scale bars, 20 μm. Inset scale bars, 5 μm.

[w/v] Amido black 10B [Sigma-Aldrich]). Samples were mixed thoroughly and centrifuged at 15,000 rpm for 10 min. After removal of supernatant, pellets were washed with 1-ml washing solution (90% ethanol and 10% acetic acid) and centrifuged at 15,000 rpm for 10 min. Resulting pellets were dissolved in 1 ml 0.2 N NaOH and the corresponding optical density (OD) at 630 nm was measured through a plate reader (Synergy HTX Multi-Mode Microplate Reader; BioTek). Protein concentration (C) was calculated through the formula $C = (OD–b)/10a$, where a and b were calibrated by the Bovine Serum Albumin (BSA) standard curve of the staining solution. For Marchantia, protein concentration was quantified using Bradford assay (Sigma-Aldrich). Chemiluminescence was acquired through iBright Imaging System (Invitrogen).

For Western blotting, indicated amount of protein was loaded on 4–20% Mini-PROTEAN TGX precast gel (Bio-Rad) and blotted on nitrocellulose using the semi-dry Trans-Blot Turbo Transfer System (Bio-Rad). Protein extract was immunoblotted with either rabbit anti-NBR1 (1:2,000, AS14 2805; Agrisera), rabbit anti-IDH (1:5,000, AS06 203A; Agrisera), rabbit anti-ACT (1:5,000; Agrisera, AS13 2,640), mouse anti-GFP (1:3,000, 11814460001; Roche), or mouse anti-RFP (1:3,000, AB_2631395; Chromotek) antibodies. Images were captured through Chem-iDoc Touch Imaging System (Bio-Rad) or iBright Imaging System (Invitrogen) by developing with SuperSignal West Pico PLUS Chemiluminescent Substrate (Thermo Fisher Scientific). Protein band intensities were quantified through Image Lab 6 (Bio-Rad) as previously described in Stephani et al. (2020).

### Protease protection assay
Roughly 5–10 mg of *Arabidopsis* seeds were grown in 6-well plates under continuous light and continuous shaking for 7 d.

7-d-old *Arabidopsis* seedlings were subjected to Torin1 treatment (3 μM for 90 min) for autophagy induction and immediately grinded in GTEN-based buffer (GTEN [10% glycerol, 30 mM Tris [pH 7.5], 150 mM NaCl, 1 mM EDTA [pH 8], 0.4 M sorbitol, 5 mM MgCl2, 1 mM Dithiothreitol [DTT], 100× liquid protease inhibitor cocktail [Sigma-Aldrich], and 1% Polyvinylpolypyrrolidon [PVPP]) in a 3:1 v/w ratio. Afterwards lysates underwent several differential centrifugation steps where each time the supernatant was transferred. Samples were spun for (i) 10 min at 1,000 $g$, to remove cell debris and nuclei; (ii) 10 min at 10,000 $g$, to remove bigger organelles like mitochondria and chloroplasts; (iii) 10 min at 15,000 $g$, to further remove organelles (S3 fraction); and finally (iv) 60 min at 100,000 $g$ (S4 and P4 fraction; LaMontagne et al., 2016). Protein concentration in S3 was normalized through Bradford (Sigma-Aldrich) to ensure that equal amount of protein was loaded before ultracentrifugation step. The P4 fraction was dissolved gently in GTEN-based buffer (without PVPP) and was further subjected to 30 ng/μl proteinase K (Sigma-Aldrich) and 1% Triton X-100 treatment for 30 min on ice. The reaction was stopped with 5 mM phenylmethylsulfonyl fluoride (PMSF). Proteins were then precipitated overnight with 0.1% sodium deoxycholate (NaDOC) and 11% trichloroacetic acid (TCA). Resulting pellets were washed twice with 100% acetone and dissolved in 2× Laemmli buffer. Proteins were quantified again through the Amido black method as described above and 5 μg were loaded on the gel.

### In vitro pull-down assays
Recombinant proteins were expressed using Rosetta2(DE3) pLysS *E. coli* strain. Bacteria were grown to an $OD_{600}$ of 0.6–0.7 followed by induction with 300 μM IPTG and overnight incubation at room temperature (RT). 50 ml of culture was pelleted

Table 1. **A. thaliana lines used in this study**

| Name | Accession Number | Source or reference |
|---|---|---|
| Col-0 | | |
| cfs1-2 | At3g43230 | SALK_024058; Sutipatanasomboon et al. (2017) Sci. Rep. |
| cfs2-1 | At1g29800 | SALK_02775; this study |
| atg5-1 | At5g17290 | SAIL_129B07L; Thompson et al. (2005) Plant Physiol. |
| vps23.1-3 | At3g12400 | SAIL_1233E07_2-4; Nagel et al. (2017) PNAS. |
| vps23.2-1 | At5g13860 | SAIL_237G05_3-4; Nagel et al. (2017) PNAS. |
| cfs1-2 × cfs2-1 | At3g43230 (CFS1); At1g29800 (CFS2) | This study |
| pUBQ::mCherry | | Geldner et al. (2009) Plant J. |
| pUBQ::mCherry | | This study |
| pUBQ::GFP-ATG8A | At4g21980 | Munch et al. (2015) The Plant Cell. |
| pUBQ::GFP-ATG8A × cfs1-2 | At4g21980 (ATG8A); At3g43230 (CFS1) | This study |
| pUBQ::mCherry-ATG8E | At2g45170 | Hu et al. (2020) J. Integr. Plant Biol. |
| pUBQ::mCherry-ATG8E × cfs1-2 | At2g45170 (ATG8E); At3g43230 (CFS1) | This study |
| pUBQ::mCherry-ATG8E × atg5-1 | At2g45170 (ATG8E); At5g17290 (ATG5) | This study |
| pCFS1::gCFS1 in cfs1-2 | At3g43230 | Sutipatanasomboon et al. (2017) Sci Rep. |
| pUBQ::mCherry-CFS1 in cfs1-2 | At3g43230 | This study |
| pUBQ::mCherry-CFS1$^{AIM}$ in cfs1-2 | At3g43230 | This study |
| pUBQ::GFP-CFS1 in cfs1-2 | At3g43230 | This study |
| pUBQ::mScarlett-CFS2 in pUBQ::GFP-CFS1 in cfs1-2 | At1g29800 (CFS2); At3g43230 (CFS1) | This study |
| pUBQ::mCherry-CFS1$^{FYVE}$ in pUBQ::GFP-ATG8A × cfs1-2 | At3g43230 (CFS1); At4g21980 (ATG8A) | This study |
| pUBQ::mCherry-CFS1$^{SYLF}$ in pUBQ::GFP-ATG8A × cfs1-2 | At3g43230 (CFS1); At4g21980 (ATG8A) | This study |
| pUBQ::mCherry-CFS1$^{tri}$ in pUBQ::GFP-ATG8A × cfs1-2 | At3g43230 (CFS1); At4g21980 (ATG8A) | This study |
| pUBQ::mCherry-CFS1$^{PSAPP}$ in cfs1-2 | At3g43230 | This study |
| pUBQ::mCherry-CFS1$^{PSAPP}$ in pUBQ::GFP-ATG8A × cfs1-2 | At3g43230 (CFS1); At4g21980 (ATG8A) | This study |
| pUBQ::mCherry-CFS1$^{AIM+PSAPP}$ in cfs1-2 | At3g43230 | This study |
| pUBQ::mCherry-CFS1$^{AIM+PSAPP}$ in pUBQ::GFP-ATG8A × cfs1-2 | At3g43230 (CFS1); At4g21980 (ATG8A) | This study |
| pUBQ::GFP-VPS23A in pUBQ::mCherry-CFS1 in cfs1-2 | At3g12400 (VPS23A); At3g43230 (CFS1) | This study |
| pUBQ::GFP-VPS23A in pUBQ::mCherry-CFS1$^{PSAPP}$ in cfs1-2 | At3g12400 (VPS23A); At3g43230 (CFS1) | This study |
| pUBQ::VPS23A-mTurquoise2 in pUBQ::GFP-ATG8A × pUBQ::mCherry-CFS1 in cfs1-2 | At3g12400 (VPS23A); At4g21980 (ATG8A); At3g43230 (CFS1) | This study |
| pUBQ::VPS23A-mTurquoise2 in pUBQ::mCherry-CFS1$^{PSAPP}$ in pUBQ::GFP-ATG8A × cfs1-2 | At3g12400 (VPS23A); At4g21980 (ATG8A); At3g43230 (CFS1) | This study |
| pUBQ::mCherry-MpCFS1 in pUBQ::GFP-ATG8A × cfs1-2 | Mp3g11370 (CFS1); At4g21980 (ATG8A); At3g43230 (CFS1) | This study |
| p35S::GFP-FREE1 in pUBQ::mCherry-CFS1 in cfs1-2 | At1g20110 (FREE1); At3g43230 (CFS1) | This study |
| atg5-1 × pUBQ::mCherry-CFS1 in cfs1-2 | At5g17290 (ATG5); At3g43230 (CFS1) | This study |
| pUBQ::GFP-ATG8A × pUBQ::mCherry-CFS1 in cfs1-2 | At4g21980 (ATG8A); At3g43230 (CFS1) | This study |
| pUBQ::GFP-ATG8A × pUBQ::mCherry-CFS1$^{AIM}$ in cfs1-2 | At4g21980 (ATG8A); At3g43230 (CFS1) | This study |
| pUBQ::mCherry-ATG8E × pUBQ::GFP-CFS1 in cfs1-2 | At2g45170 (ATG8E); At3g43230 (CFS1) | This study |
| pVPS3::mGFP-VPS3 | At1g22860 | Takemoto et al. (2018) PNAS. |
| pVPS3::mGFP-VPS3 × pUBQ::mCherry-CFS1 in cfs1-2 | At1g22860 (VPS3); At3g43230 (CFS1) | This study |
| pVPS39::VPS39-mGFP | At4g36630 | Takemoto et al. (2018) PNAS. |
| pVPS39::VPS39-mGFP × pUBQ::mCherry-CFS1 in cfs1-2 | At4g36630 (VPS39); At3g43230 (CFS1) | This study |
| pUBQ::YFP-VAMP711 | At4g32150 | Geldner et al. (2009) Plant J. |

| Name | Accession Number | Source or reference |
|------|------------------|---------------------|
| pUBQ::YFP-VAMP711 × pUBQ::mCherry-CFS1 in _cfs1-2_ | At4g32150 (VAMP711); At3g43230 (CFS1) | This study |
| p35S::NAG1-EGFP | At4g38240 | Grebe et al. (2003) Curr Biol. |
| p35S::NAG1-EGFP × pUBQ::mCherry-CFS1 in _cfs1-2_ | At4g38240(NAG1); At3g43230 (CFS1) | This study |
| pa1::VHA-a1-GFP | At2g28520 | Dettmer et al. (2006) The Plant Cell. |
| pa1::VHA-a1-GFP × pUBQ::mCherry-CFS1 in _cfs1-2_ | At2g28520 (VHA-a1); At3g43230 (CFS1) | This study |
| pRPS5a::GFP-ARA7 | At4g19640 | Richter et al. (2007) Nature. |
| pRPS5a::GFP-ARA7 × pUBQ::mCherry-CFS1 in _cfs1-2_ | At4g19640 (ARA7); At3g43230 (CFS1) | This study |
| pPIN::PIN2-GFP | At5g57090 | Xu and Scheres (2005) The Plant Cell. |
| pPIN2::PIN2-GFP × _cfs1-2_ | At5g57090 (PIN2); At3g43230 (CFS1) | This study |
| pNBR1::TagRFP-NBR1 | At4g24690 | Provided by Alyona Minina |
| pNBR1::NBR1-GFP | At4g24690 | Hafrén et al. (2017) PNAS. |
| pNBR1::NBR1-GFP × pUBQ::mCherry-CFS1 in _cfs1-2_ | At4g24690 (NBR1); At3g43230 (CFS1) | This study |
| pUBQ::mCherry-ATG8E × pNBR1::NBR1-GFP | At2g45170 (ATG8E); At4g24691 (NBR1) | This study |
| pUBQ::GFP-ATG11 | At4g30790 | Li et al. (2014) The Plant Cell. |
| pUBQ::GFP-ATG11 × pUBQ::mCherry-CFS1 in _cfs1-2_ | At4g30790 (ATG11); At3g43230 (CFS1) | This study |
| pALIX::GFP-3Gly-ALIX | At1g15130 | Cardona-López et al. (2015) The Plant Cell. |
| pALIX::GFP-3Gly-ALIX × pUBQ::mCherry-CFS1 in _cfs1-2_ | At1g15130 (ALIX); At3g43230 (CFS1) | This study |
| pVPS23A::VPS23A-TagRFP | At3g12400 | Nagel et al. (2017) PNAS. |
| pVPS23A::VPS23A-TagRFP × pUBQ::GFP-CFS1 in _cfs1-2_ | At3g12400 (VPS23A); At3g43230 (CFS1) | This study |
| pVPS23A::VPS23A-TagRFP × pUBQ::GFP-ATG8A | At3g12400 (VPS23A); At4g21980 (ATG8A) | This study |
| pUBQ::GFP-ATG8A × _vps23.1-3_ | At4g21980 (ATG8A); At3g12400 (VPS23A) | This study |
| pUBQ::GFP-ATG8A × _vps23.2-1_ | At4g21980 (ATG8A); At3g13860 (VPS23B) | This study |

and resuspended in 5 ml of EDTA-free buffer (10% Glycerol, 25 mM Tris/HCl pH 7.5, 150 mM NaCl, 0.1 mM TCEP, 0.1% Nonidet P-40/Igepal, 10 µM $ZnCl_2$ and 1 tablet/50 ml cOmplete, EDTA-free Protease Inhibitor Cocktail [Sigma-Aldrich]) containing 10× FastBreak Cell Lysis reagent (Promega) and Benzonase. Cells were broken open through incubation on a spinning wheel at room temperature for 10–20 min. Lysates were clarified through centrifugation at 15,000 rpm at 4°C for 10 min. For proteins expressed in planta, 40 seeds per genotype were grown. Frozen plant tissue was homogenized as described above and 700 µl EDTA-free buffer containing 2% PVPP was added. Lysates were centrifuged twice at 15,000 rpm at 4°C for 10 min. 100 µl of _E. coli_ lysate and 400 µl of plant lysate were mixed and incubated with 10 µl of equilibrated Glutathione High-Capacity Magnetic Agarose Beads (Sigma-Aldrich) for 1 h at 4°C on a spinning wheel. For peptide competition, peptides were added to a final concentration of 200 µM as described in Stephani et al. (2020). Beads were washed five times with EDTA-free buffer, without TCEP, eventually eluted in 50 µl 2× Laemmli buffer and boiled for 10 min at 95°C. Gels were loaded with 15 µl of sample per well. Anti-GST HRP conjugate (Sigma-Aldrich) was used in a 1:2,000 dilution.

### In vivo co-immunoprecipitation

For co-immunoprecipitation, 40 seeds per genotype were grown in 1/2 MS media for 7 d. Proteins were extracted by adding 800 µl of EDTA-free buffer with 2% PVPP. Lysates were cleared by centrifugation at 15,000 rpm at 4°C for 10 min twice. 500 µl supernatant was incubated with 20 µl RFP-Trap Magnetic Agarose beads (Chromotek) for 1 h. Beads were washed three and five times with EDTA-free buffer, without TCEP, before and after incubation with lysate, respectively. Beads were eluted in 50 µl 2× Laemmli buffer, boiled for 10 min at 95°C and subjected to Western blot analysis with indicated antibodies.

### Affinity purification-mass spectrometry (AP-MS)

For affinity purification, S4, P4, and P4 + proteinase K samples described in Fig. 1 A were prepared as same as the method described above for the protease protection sample preparation, except that 0.1% Nonidet P-40/Igepal was added and samples were incubated for 1 h with 40 µl RFP-Trap Magnetic Agarose beads (Chromotek) after proteinase K treatment. Mass spectrometry sample preparation and measurement were performed as previously described in Stephani and Picchianti et al., 2020 (Stephani et al., 2020).

### Mass spectrometry data processing

The total number of MS/MS fragmentation spectra was used to quantify each protein (Table S1). The data matrix of spectral count values (Table S2) was submitted to a negative-binomial test using the R package IPinquiry4 (https://github.com/hzuber67/IPinquiry4) that calculates fold change and P values

| Name | Accession number | Expression system | Backbone | Additional information | Source or reference |
|---|---|---|---|---|---|
| pUBQ::mCherry | | A. thaliana | pGGZ003 | | This study |
| pUBQ::mCherry-CFS1 | At3g43230 | A. thaliana | pGGZ003 | | This study |
| pUBQ::GFP-CFS1 | At3g43230 | A. thaliana | pGGZ003 | | This study |
| pUBQ::mCherry-CFS1$^{AIM}$ | At3g43230 | A. thaliana | pGGZ003 | WLNL267ALNA | This study |
| pUBQ::mCherry-CFS1$^{FYVE}$ | At3g43230 | A. thaliana | pGGZ003 | RHHCR195AHACA | This study |
| pUBQ::mCherry-CFS1$^{SYLF}$ | At3g43230 | A. thaliana | pGGZ003 | K282A, R288A, K320A | This study |
| pUBQ::mCherry-CFS1$^{tri}$ | At3g43230 | A. thaliana | pGGZ003 | RHHCR195AHACA (FYVE); K282A, R288A, K320A (SYLF); WLNL267ALNA (AIM) | This study |
| pUBQ::mCherry-CFS1$^{PSAPP}$ | At3g43230 | A. thaliana | pGGZ003 | PSAPP145AAAAA (PSAPP) | This study |
| pUBQ::mCherry-CFS1$^{AIM+PSAPP}$ | At3g43230 | A. thaliana | pGGZ003 | PSAPP145AAAAA (PSAPP); WLNL267ALNA (AIM) | This study |
| pUBQ::mScarlett-CFS2 | At1g29800 | A. thaliana | pGGZ003 | | This study |
| pUBQ::mCherry-MpCFS1 | Mp3g11370 | A. thaliana | pGGZ003 | | This study |
| p35S::GFP-FREE1 | At1g20110 | A. thaliana | pMDC43 | | Provided by Pedro L. Rodriguez |
| pUBQ::GFP-VPS23A | At3g12400 | A. thaliana | pGGZ003 | | |
| pUBQ::VPS23A-mTurquoise2 | At3g12400 | A. thaliana | pGGZ003 | | |
| GST | | E. coli | | | Stephani et al. (2020) |
| GST-ATG8A | At4g21980 | E. coli | | | Stephani et al. (2020) |
| GST-ATG8A$^{ADS}$ | At4g21980 | E. coli | | YL50AA | Stephani et al. (2020) |
| GST-ATG8B | At4g04620 | E. coli | | | Stephani et al. (2020) |
| GST-ATG8C | At1g62040 | E. coli | | | Stephani et al. (2020) |
| GST-ATG8D | At2g05630 | E. coli | | | Stephani et al. (2020) |
| GST-ATG8E | At2g45170 | E. coli | | | Stephani et al. (2020) |
| GST-ATG8F | At4g21980 | E. coli | | | Stephani et al. (2020) |
| GST-ATG8G | At3g60640 | E. coli | | | Stephani et al. (2020) |
| GST-ATG8H | At3g06420 | E. coli | | | Stephani et al. (2020) |
| GST-ATG8I | At3g15580 | E. coli | | | Stephani et al. (2020) |
| pEF::GFP-ATG8A | Mp1g21590 | M. polymorpha | pMpGWB303 | | |
| pEF::GFP-ATG8B | Mp5g05930 | M. polymorpha | pMpGWB303 | | This study |
| pEF::mScarlet-CFS1 | Mp3g11370 | M. polymorpha | pGGSUN | | This study |
| pMpGE010_CFS1g1g2 | Mp3g11370 | M. polymorpha | pMpGE010 | | This study |

using the quasi-likelihood negative binomial generalized log-linear model implemented in the edgeR package. The pairwise comparisons were the following: (i) mCherry P4 vs. mCherry-ATG8E P4, (ii) mCherry-ATG8E S4 vs. mCherry-ATG8E P4, and (iii) mCherry-ATG8E P4 + protease K vs. mCherry-ATG8E P4. In each case, comparisons were obtained from two independent biological replicates. Only proteins with log(FC) > 0 and P value <0.05 were considered to build the Venn diagram in Fig. 1 B. Venn diagram was built using the Venny 2.1.0 online tool (https://bioinfogp.cnb.csic.es/tools/venny/index.html) and then redrawn manually. Analysis results are shown in Table S3. The mass spectrometry proteomics data have been deposited to the ProteomeXchange Consortium through the PRIDE (Perez-Riverol et al., 2019) partner repository.

### Yeast two hybrid screening
The screening was performed by Hybrigenics against the Arabidopsis cDNA library. Results are shown in Table S4.

### Preparation of A. thaliana samples for confocal microscopy
For all experiments except PIN2 endocytosis imaging, Arabidopsis seeds were vapor-phase sterilized by chlorine (generated by a 10:1 mixture of 13% sodium hypochlorite and 36% HCl) for 15 min and were subsequently stored at 4°C for 2 d for vernalization. Vernalized seeds were spread on 1/2 MS media plates

**Table 3. Primer and synthetic sequences used in this study for genotyping, cloning, and mutagenesis**

| Name | Sequence | Additional information | Source or reference |
|---|---|---|---|
| LBb1.3 (SAIL) | 5'-ATTTTGCCGATTTCGGAAC-3' | For genotyping | SIGnAL. 2017. |
| cfs1-2 (LP) | 5'-GCTCGACTAAGAACAGCATGC-3' | For genotyping | SIGnAL. 2017. |
| cfs1-2 (RP) | 5'-TACATGGGTTTGGATGAGCG-3' | For genotyping | This study |
| cfs2-1 (LP) | 5'-TGGTGAGTTTTACGCTTACCG-3' | For genotyping | SIGnAL. 2017. |
| cfs2-1 (RP) | 5'-TTGGTCCATCAAATAAGGCTG-3' | For genotyping | SIGnAL. 2017. |
| LB3 (SALK) | 5'-AATTTCATAACCAATCTCGATACAC-3' | For genotyping | Nagel et al. (2017) PNAS. |
| vps23.1-3 (LP) | 5'-AACTTTGGACTTTGGAACATGTCCACTCCTTACA-3' | For genotyping | Nagel et al. (2017) PNAS. |
| vps23.1-3 (RP) | 5'-AGGAACCATCATCACACGCATAGCCACAG-3' | For genotyping | Nagel et al. (2017) PNAS. |
| vps23.2-1 (LP) | 5'-TCGTCATCTGTCGTCGTCTTCAAG-3' | For genotyping | Nagel et al. (2017) PNAS. |
| vps23.2-1 (RP) | 5'-GTATCACGAATGCAACCTAGCTGCAATGGAAG-3' | For genotyping | Nagel et al. (2017) PNAS. |
| CFS1_part1_fwd | 5'-AACAGGTCTCAGGCTCTATGGCTACTCTCAACGGAAA-3' | For cloning CDS, part 1 | This study |
| CFS1_part1_rev | 3'-AACAGGTCTCACGAGATATTACCAGACCAGT-5' | For cloning CDS, part 1 | This study |
| CFS1_part2_fwd | 5'-AACAGGTCTCTCTCGTAGACCAGACGGGTCATGG-3' | For cloning CDS, part 2 | This study |
| CFS1_part2_rev | 3'-AACAGGTCTCACTGACGGGCGCAAACGAGCATA-5' | For cloning CDS, part 2 | This study |
| CFS2_part1_fwd | 5'-ACCCTCTGCCATCTCTTCATTTGGT-3' | For cloning CDS, part 1 | This study |
| CFS2_part1_rev | 3'-AACAGGTCTCAGGGTGGTGACCAGGAGCCATCATCT-5' | For cloning CDS, part 1 | This study |
| CFS2_part2_fwd | 5'-TCCTGGTCTCCACCCTCTGC-3' | For cloning CDS, part 2 | This study |
| CFS2_part2_rev | 3'-AACAGGTCTCACTGAGTCTTCAGACAATGGAGAAATTGCCGGTGACCGAACAGTTTCA-5' | For cloning CDS, part 2 | This study |
| MpCFS1 | 5'-AACAGGTCTCAGGCTCTATGGAGGAGAATGAGTACAACGGACTCTGCCGGACAGATACCGTCGAAGAGGAGAAATATTCTGAGAAAAGTGGTCGATACAAGGGTGTCATTCGTGATGAGAAACCCACTTGTGCTGTCTGTATGGAAGTTCTAGGAACTCATGGTGGGCCTGCATCTCTCGCCTGCGGCCACAATGGATGTCTCGAGTGCTTACAACAGGTCCAGATGCACTCGAACATGCCTGTCTGTCCTCTTTGTCGTACACCCTTTGATGGAGATATCAATTTGGCCCCGAACTTAGACTTACGGGCTGCACTCGAGTGTGCCGAACAAGCCGCCGCAGCTGCGCGAGCTGCCGAGGATCGGAGGATGGTTGCTGTGTACAATTCTAAGCAGATTGTGAAGCCGAGCAGAGATAAAACCACCAGGTCACACCAACAGCGATTCATCTATGATTTGCCCTATCCTGAGGAAACAGGACCTTGGATTCCGGTGTCAATTCCGCCCAAGGGCTCGAAGGACTTTGATGGGCCTGACATGGGCAGTATATATGGAGATGTTTTTTCCGGAACCCTTGATGACAGTGGGTGTCCGACAAGGGGCTCCGGACATCAACCTTTGGAATGTTTTGAGTGGATTTCTTGCAATCATATCGGGAAGGGGTGGTACACAAAGCCAGAATTTCCACTCACAGTCCGGTTATTCTGATAGTAGAACATCTACGTACTACCGAGAAAGTTCTGGTCATGGAAGATCTAACAGCAATTCATGGAGTGGGAATTTACACAGGCCAAGTGCACCTCCACTGCTCCTTGGCACTAATGATTCCGATGTCAGCCTCATACGAGCTATTCTCGAAGCAGAGCCCCCGGACTGGATGCCTGACAGTTCGGCCCTTTCCTGTATGCTCTGTGGCGCTGCCTTTAGGGCAATAACTTGTGGAAGACATCATTGTCGGTTCTGCGGGGGAATTTTTTGTAGGAGATGCTCTTCCGGGAGATGTCTTTTACCTGTGAAGTTCCGGGACAGGGACCCGAAAAGAGTTTGCGATACCTGCTGGGAGAGACTGGAACCTGTACAGAAGAATCTTGCCGACCGAGTTAGTAATGCAGCACAAATCGCTCTACATGATGTGACTGACTTCAGCTGTATGACAGCATGGATAAACAGTCCACTTGGGCTTTCGATGGAGCAAGAAATTTTTAAGGCAACAAATGCCTTGCGTAGCTATTTGGAGATTGGAGTACTCAAACCAGAAAGATCCATCCCCGACGCTGTTCTGAAAGGAGCCTGTGGCTTGGCAATCATCACAGTGTTCAAGGCCGGGTTTATGATGACCTACAAATTTGGTACGGGATTGGTTGTTTCACGGAGGAGAGATGGGTCCTGGTCTGCACCATCAGCTATCGCATCGTGTGGTCTGGGATGGGGAGCTCAGGCTGGTGGGGAGCTGACTGATTTCATCATAGTCCTAAGGACAGAAGAAGCTGTGAAGGCCTTTAGTGGGAGAGTACATTTGTCAGTTGGTGCCGGCCTAAGCGTCGCTGCCGGACCTGTTGGAAGAGCTGCGGAAGCAGATCTTAGAGCTGGTGATGGGGGTACTGCAGCTTGTTACACTTACAGTCGAAGCAAAGGAGCGTTCGTCGGATGTTCAATCGAAGGCAATGTTGTGGCGACACGGTCCTCCGCCAACATGCGCTTTTATGGAGAATCTGGTGTCACTGCAGTCGACATTCTTCTTGGAGACGTTCCAAGACCTCGGGCTGCAGCTCCGCTTTATAGTGCTTTGGATGAACTTTTTGTAAAGGTGGACACATCAGTGAGACCTGTT-3' | For cloning CDS | This study |
| MpATG8A_fwd | 5'-AACAGGTCTCAGGCTCCATGGGAGAGAGCGAGTTTCAAT-3' | For cloning CDS | This study |

Table 3. **Primer and synthetic sequences used in this study for genotyping, cloning, and mutagenesis (*Continued*)**

| Name | Sequence | Additional information | Source or reference |
|---|---|---|---|
| MpATG8A_rev | 3'-AACAGGTCTCACTGATCAGCCAACGAGGGCGAC-5' | For cloning CDS | This study |
| MpATG8B_fwd | 5'-AACAGGTCTCAGGCTCCATGACGGGGAAGAGGAGTTC-3' | For cloning CDS | This study |
| MpATG8B_rev | 3'-AACAGGTCTCACTGACTAAACCAGAGGAAGGCAATTG-5' | For cloning CDS | This study |
| gRNA_MpCFS1_1_fwd | 5'-CTCGAAGTTCTAGGAACTCATGGT-3' | For CRISPR/Cas9 | This study |
| gRNA_MpCFS1_1_rev | 5'-AAACACCATGAGTTCCTAGAACTT-3' | For CRISPR/Cas9 | This study |
| gRNA_MpCFS1_2_fwd | 5'-CTCGACTTCAGCTGTATGACAGCA-3' | For CRISPR/Cas9 | This study |
| gRNA_MpCFS1_2_rev | 5'-AAACTGCTGTCATACAGCTGAAGT-3' | For CRISPR/Cas9 | This study |
| CFS1_mutAIM_fwd | 5'-TGGACTTGCTCAAGAGGAGCGTTGAATGCCCCTGTTGGTCTATCCAT-3' | For site-directed mutagenesis | PrimerX. 2017. |
| CFS1_mutAIM_rev | 3'-ATGGATAGACCAACAGGGGCATTCAACGCTCCTCTTGAGCAAGTCCA-5' | For site-directed mutagenesis | PrimerX. 2017. |
| CFS1_mutFYVE_fwd | 5'-GGTGCTCATGCTTGTGCTTTCTGCGGAGGGATATTTTGTAGA-3' | For site-directed mutagenesis | PrimerX. 2019. |
| CFS1_mutFYVE_rev | 3'-GAAAGCACAAGCATGAGCACCACACGTAATTGCAGTAAAAGG-5' | For site-directed mutagenesis | PrimerX. 2019. |
| CFS1_mutSYLF | 5'-AGTCTCCATGGAAGATGAGATATACGCAGCTGCTAATACGTTGGCAGGTTACTGCCAGGT AGCAAGACTAGACCCTGAGAAATCCATACCGCATGCTGTTCTTAGTCGAGCTAAAGGTTTGGC AATCATAACGGTTGCCGCTGCTGGGGCTTTACTGTCATACAAACTCGGGACTGGTCTGGTAAT ATCTCGTAGACCAGACGGGTCATGGTCTGCTCCATCTGCCATACTATCAGTAGGTCTTGGATG GGGTGCACAGATTGGAGGTGAGTTGATGGACTTCATAATAGTGCTTCATGATGTGAAAGC CGTGAAGACATTCTGCAGCAGAATGCACTTCTCTCTAGGCGCGGGATGCAGTGCAGCAGC AGGGCCTATCGGGAGAGTGTTGGAGGCTGACCTTCGAGCTGGTGACAAAGGCTCTGGTGT CTGCTATACTTACAGCCGCAGCAAAGGGGCATTTGTGGGAGTATCTCTAGAGAGACT-3' | For mutagenesis | This study |
| CFS1_mutPSAPP_fwd | 5'-CATTCGTCTGTTTACATAGCGGCTGCTGCGGCTCTACTTGAACCTAGTG-3' | For site-directed mutagenesis | PrimerX. 2018. |
| CFS1_mutPSAPP_rev | 3'-CACTAGGTTCAAGTAGAGCCGCAGCAGCCGCTATGTAAACAGACGAATG-5' | For site-directed mutagenesis | PrimerX. 2018. |
| VPS23A_fwd | 5'-AACAGGTCTCAGGCTCTATGGTTCCCCCGCCGTCTAATCC-3' | For cloning CDS, part 1 | This study |
| VPS23A-Bsa1_rev | 3'-AACAGGTCTCAGAAACGAGGCCAGAAGGAGTGACAT-5' | For cloning CDS, part 1 | This study |
| VPS23A-Bsa1_fwd | 5'-AACAGGTCTCATTTCTCTTCCGTACCTTCAGAATTGG-3' | For cloning CDS, part 2 | This study |
| VPS23A_rev | 3'-AACAGGTCTCACTGATGAATGTAACCTACCTGCGATGGCTG-5' | For cloning CDS, part 2 | This study |

(+1% plant agar [Duchefa]) and grown at 21°C at 60% humidity under LEDs with 50 mM/m²s a and a 16 h light/8 h dark photoperiod for 5 d. Plates were placed vertically to let the roots elongate along the media surface. 5-d-old seedlings were placed in 1/2 MS media and treated with salt or chemicals as indicated in each experiment before confocal imaging. For nitrogen starvation, 1/2 MS media was replaced by nitrogen-deficient 1/2 MS media. For PIN2 endocytosis imaging, *Arabidopsis* seeds were vapor-phase sterilized by chlorine (generated by a 25:1 mixture of 2.6% sodium hypochlorite and 36% HCl) for 3–4 h. The sterilized seeds were spread on 1/2 MS media plates (+1% plant agar [Duchefa]). The plated seeds were subsequently stored at 4°C for 2 d for vernalization. Vernalized seeds were then grown at 21°C at 60% humidity under LEDs with 50 mM/m²s a and a 16 h light/8 h dark photoperiod for 5 d. Plates were placed vertically to let the roots elongate along the media surface. 5-d-old *Arabidopsis*

seedlings were grown on 1/2 MS media plates under light or 6-h dark conditions before imaging.

For confocal microscopy, *Arabidopsis* seedlings were placed on a microscope slide with water or water with 0.002 mg/ml propidium iodide and covered with a coverslip. The epidermal cells of root meristem zone were used for CFS1 and FM 4-64 colocalization imaging, PIN2 endocytosis imaging, or vacuolar morphology imaging. The epidermal cells of root elongation zone were used for confocal imaging in Fig. 3 D and time-lapse imaging in Videos 1, 2, and 3. For all other experiments, the epidermal cells of root transition zone were used for image acquisition.

**Preparation of *M. polymorpha* samples for confocal microscopy**
The *M. polymorpha* asexual gemmae were incubated in 1/2 Gamborg B5 media for 2 d before imaging. 2-d-old *M. polymorpha* thalli were placed on a microscope slide with water and covered with a coverslip. The meristem region was used for image acquisition.

## Confocal microscopy

All images except for Airyscan imaging, PIN2 endocytosis imaging and time-lapse imaging were acquired by an upright point laser scanning confocal microscope ZEISS LSM800 Axio Imager.Z2 (Carl Zeiss) equipped with high-sensitive GaAsP detectors (Gallium Arsenide), a LD C-Apochromat 40X objective lens (numerical aperture 1.1, water immersion), and ZEN software (blue edition, Carl Zeiss). For Fig. 10, a Plan-Apochromat 20X objective lens (numerical aperture 0.8, dry) is used instead of the 40X one. GFP and BCECF-AM fluorescence were excited at 488 nm and detected between 488 and 545 nm. TagRFP, mCherry, and mScarlet fluorescence were excited at 561 nm and detected between 570 and 617 nm. FM 4-64 fluorescence was excited at 561 nm and detected between 656 and 700 nm (Fig. S4 A) or 576 and 700 nm. For Z-stack imaging, interval between the layers was set as 1 µm. For each experiment, all replicate images were acquired using identical confocal microscopic parameters. Confocal images were processed with Fiji (version 1.52, Fiji) and Imaris (version 9.0.1, BITPLANE). For Airyscan imaging, Z-stack images were acquired by an inverted point laser scanning confocal microscope ZEISS LSM980 Axio Observer.Z1/7 (Carl Zeiss) equipped with a hexagonal GaAsP detector Airyscan II (Gallium Arsenide), a plan-Apochromat 63x objective lens (numerical aperture 1.40, oil immersion), and ZEN software (blue edition, Carl Zeiss) under room temperature. GFP fluorescence was excited at 488 nm and detected between 495 and 550 nm. TagRFP fluorescence was excited at 561 nm and detected between 573 and 627 nm. Interval between layers was set as 0.15 µm. Original images were deconvoluted by ZEN software using the default mode (blue edition, Carl Zeiss). Deconvoluted images were processed with Fiji (version 1.52, Fiji).

For PIN2 endocytosis imaging, images were acquired by an inverted Zeiss microscope AxioObserver Z1 (Carl Zeiss) equipped with a spinning disk module CSU-W1-T3 (Yokogawa), a Prime 95B camera (http://www.photometrics.com/), a Plan-Apochromat 63× objective lens (numerical aperture 1.4, oil immersion), and the Metamorph acquisition software (Molecular Devices). GFP was excited with a 488 nm laser (150 mW)

and was detected by a 525/50 nm BrightLine single-band bandpass filter (Semrock). Confocal images were processed with Fiji (version 1.52, Fiji).

For time-lapse imaging, images were acquired by an Andor Dragonfly confocal platform (Andor) equipped with a spinning disc confocal microscope (Nikon Ti2E inverted microscope). A 40X objective lens (numerical aperture 1.15, water immersion) was installed for acquiring images. The pinhole disk pattern was set as 40 µm. Green fluorescence signals were excited by 488 nm laser and detected through an Andor Zyla sCMOS camera. Red fluorescence signals were excited by 561 nm laser and detected through an Andor iXon 888 EMCCD camera. The two cameras acquired the image at the same time. For time-lapse mode imaging, the total imaging time is 60 s, with an interval of 1 s. Confocal images were processed with Fiji (version 1.52, Fiji).

## Image processing and quantification

Mander's colocalization analyses were performed by Fiji (version 1.52, Fiji). Confocal images were background-subtracted with 25 pixels of rolling ball radius. Mander's coefficients M1 and M2 were calculated through JACoP plugin (Bolte and Cordelières, 2006). Thresholds in JACoP plugin settings were adjusted according to the puncta signals in original confocal images.

Puncta quantification was performed by Fiji (version 1.52, Fiji). Z-stack images (at least five layers) were background-subtracted with 25 pixels of rolling ball radius. Each Z-stack image was subsequently thresholded using the MaxEntropy method and was converted to an 8-bit grayscale image. Threshold values were adjusted according to the puncta signals in original confocal images. The number of puncta in thresholded images was counted by the Analyze Particles function in Fiji. For all puncta quantification, puncta with the size between 0.10 and –4.00 µm² were counted.

Imaris (version 9.0.1, BITPLANE) was used for 3D construction of the vacuole structure. Default settings were used for 3D construction while the surface signals were smoothed with a surface detail of 0.25 µm.

## TEM

For TEM samples preparation, high-pressure freezing, freeze substitution, resin embedding, and ultramicrotomy were performed as described before (Ma et al., 2021; Kang, 2010; Wang et al., 2017). Briefly, 7-d-old *Arabidopsis* seedlings were incubated in 150 mM NaCl or 50 µM DNP for 1 h and were then rapidly frozen with an EM ICE high-pressure freezer (Leica Microsystems).

For Epon resin-embedded samples, the samples were freeze-substituted in 2% OsO₄ with acetone at –80°C for 24 h and were then slowly warmed up to room temperature over 48 h. Excess OsO₄ was removed by rinsing with acetone at room temperature. Root samples were separated from planchettes and embedded in Embed-812 resin (cat. no. 14120; Electron Microscopy Sciences), and the resin was polymerized at 65°C. Ultrathin (100 nm) sections from the samples were collected on copper slot grids coated with formvar.

Table 4.   **M. polymorpha lines used in this study**

| Name | Accession Number | Source or reference |
|---|---|---|
| Tak-1 | | |
| pEF::GFP-ATG8A | Mp1g21590 | This study |
| cfs1 in pEF::GFP-ATG8A | Mp3g11370 (CFS1); Mp1g21590 (ATG8A) | This study |
| pEF::mScarlet-CFS1 in pEF::GFP-ATG8A | Mp3g11370 (CFS1); Mp1g21590 (ATG8A) | This study |
| pEF::GFP-ATG8B | Mp5g05930 | This study |
| cfs1 in pEF::GFP-ATG8B | Mp3g11370 (CFS1); Mp5g05930 (ATG8B) | This study |
| pEF::mScarlet-CFS1 in pEF::GFP-ATG8B | Mp3g11370 (CFS1); Mp5g05930 (ATG8B) | This study |

For immuno-gold labeling samples, frozen specimens were freeze-substituted in anhydrous acetone containing 0.25% glutaraldehyde and 0.1% uranyl acetate at –80°C for 24 h and were slowly warmed up to –45°C. After rinsing with precooled acetone, the cells were embedded in Lowicryl HM20 resin at –45°C and the resin was polymerized by ultraviolet illumination. Ultrathin (100 nm) sections from the samples were collected on nickel slot grids coated with formvar. The sections were probed with rabbit anti-ATG8 (Agrisera), rabbit anti-RFP (Abcam), or chicken anti-GFP polyclonal primary antibodies (Abcam) and gold particles (6, 10 nm) conjugated secondary antibodies (Ted Pella). Sections were post-stained and examined with a Hitachi 7400 TEM (Hitachi-High Technologies) operated at 80 kV.

For the quantification of immuno-gold labeling samples, ~900 gold particles in 50 TEM images from 8 TEM sections (5–10 micrographs per section) were grouped into autophagosomes, cytosol, or other organelles according to their locations. The sections were collected from five individual plastic-embedded *Arabidopsis* roots with three times of replicates for cryo-fixation and sample embedding.

### Statistical analyses

All quantification analyses and statistical tests were performed with GraphPad Prism 8 software. F-test was used to check whether the variances were significantly different ($P < 0.05$). For comparing the significance of differences between two experimental groups, Student's *t* tests were performed as indicated in each experiment. The significance level of differences between two experimental groups was marked as *, $P < 0.05$; **, $P < 0.01$; ***, $P < 0.001$; ns, not significant. For comparing the significance of differences between multiple experimental groups, one-way ANOVA was performed as indicated in each experiment.

For PIN2 endocytosis qualitative quantification, the *Arabidopsis* seedling with at least five root epidermal cells that contain visible PIN2-GFP in the vacuole is considered as the one with high PIN2 endocytic activities.

### Phylogenic tree analysis

Coding sequences of CFS1 and CFS2 homologs were obtained using the BLAST tool against representative species of different plant lineages in Phytozome (Goodstein et al., 2012). The tree was inferred from a 2283-nt-long alignment using the maximum likelihood method and Tamura-Nei model as implemented by MEGA X (Tamura and Nei, 1993; Kumar et al., 2018). 100 bootstrap method and a discrete Gamma distribution were used to model evolutionary rate differences among sites (5 categories [+G, parameter = 0.8072]; Felsenstein, 1985). Best-scoring ML tree (–98686.19) is shown with purple circles indicating bootstrap values above 80 on their respective clades. The tree is represented using Interactive Tree Of Life (iTOL) v4 (Letunic and Bork, 2019).

### Homology modeling

Cartoon representations of AtCFS1 structure and prediction of AtCFS1-VPS23A heterocomplex formation were generated by homology modeling using AlphaFold2 (Jumper et al., 2021) as implemented by ColabFold (Mirdita et al., 2022).

### Online supplemental material

Fig. S1 shows differential centrifugation coupled to AP-MS revealed CFS1 as an autophagosome-associated protein in *A. thaliana*. Fig. S2 shows that CFS2 does not play a role in autophagic flux. Fig. S3 shows functional characterization of CFS1-ATG8 interaction. Fig. S4 shows AIM between the FYVE and SYLF domains of CFS1 is conserved in plants. Fig. S5 shows CFS1 does not colocalize with VPS3, VPS39, or VAMP711, but partially colocalizes with FREE1 and ALIX. Source Data files include the uncropped blots used in the corresponding figures. Table S1 shows MS dataset that is used for analysis in Fig. 1. Table S2 shows spectral count of the mass spec data used in Fig. 1. Table S3 shows the final list of proteins identified in analysis described in Fig. 1. Table S4 shows yeast two hybrid dataset. Video 1 shows ATG8-CFS1 co-movement. Video 2 shows NBR1-CFS1 co-movement. Video 3 shows CFS1-VPS23 co-movement. Video 4 shows autophagosome movement from one cell to another.

## Data availability

All the raw images, blots, and replicates associated with figures are uploaded to the Zenodo server and can be accessed under the DOI: doi.org/10.5281/zenodo.7139130 and doi.org/10.5281/zenodo.7139412. Proteomics data are available via ProteomeXchange with identifier PXD031787.

## Acknowledgments

We thank Suayip Üstün, Karin Schumacher, Erika Isono, Gerd Juergens, Takashi Ueda, Daniel Hofius, and Liwen Jiang for sharing published materials.

We acknowledge funding from Austrian Academy of Sciences, Austrian Science Fund (FWF, P 32355, P 34944), Austrian Science Fund (FWF-SFB F79), Vienna Science and Technology Fund (WWTF, LS17-047) to Y. Dagdas; Austrian Academy of Sciences DOC Fellowship to J. Zhao, Marie Curie VIP2 Fellowship to J.C. De La Concepcion and M. Clavel; Hong Kong Research Grant Council (GRF14121019, 14113921, AoE/M-05/12, C4002-17G) to B.-H. Kang. We thank Vienna Biocenter Core Facilities (VBCF) Protein Chemistry, Biooptics, Plant Sciences, Molecular Biology, and Protein Technologies. We thank J. Matthew Watson and members of the Dagdas lab for the critical reading and editing of the manuscript.

The authors declare no competing financial interests.

Author contributions: J. Zhao performed microscopy experiments, flux measurements, phenotypic plate assays, and in vivo pull-down assays, prepared figures, and wrote the draft. M.T. Bui generated *Arabidopsis* lines, performed flux measurements, protease protection assays, phenotypic plate assays, and AP-MS experiments, and wrote the draft. J. Ma performed electron microscopy experiments and mitophagy flux assays, prepared figures, and wrote the draft. F. Künzl generated *Arabidopsis* lines. L. Picchianti performed in vitro pull-down assays and drew the model. J.C. De La Concepcion performed

phylogenetic analysis and Alphafold models, and wrote the draft. Y. Chen contributed to protease protection assays. S. Petsangouraki contributed to phenotypic characterization experiments. A. Mohseni, M. García-Leon, and M.S. Gomez performed the Marchantia-related experiments. C. Giannini performed the time-lapse imaging experiments. D. Gwennogan performed the PIN2 endocytosis experiments. R. Kobylinska performed the AP-MS experiments. M. Clavel performed the IP-MS experiments, analyzed data, prepared figures, and wrote the draft. S. Schellmann generated transgenic *Arabidopsis* lines. Y. Jaillais and J. Friml supervised students and contributed to the draft. B.-H. Kang supervised students and wrote the draft. Y. Dagdas supervised students and postdocs and wrote the draft.

Submitted: 31 March 2022

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

# Supplemental material

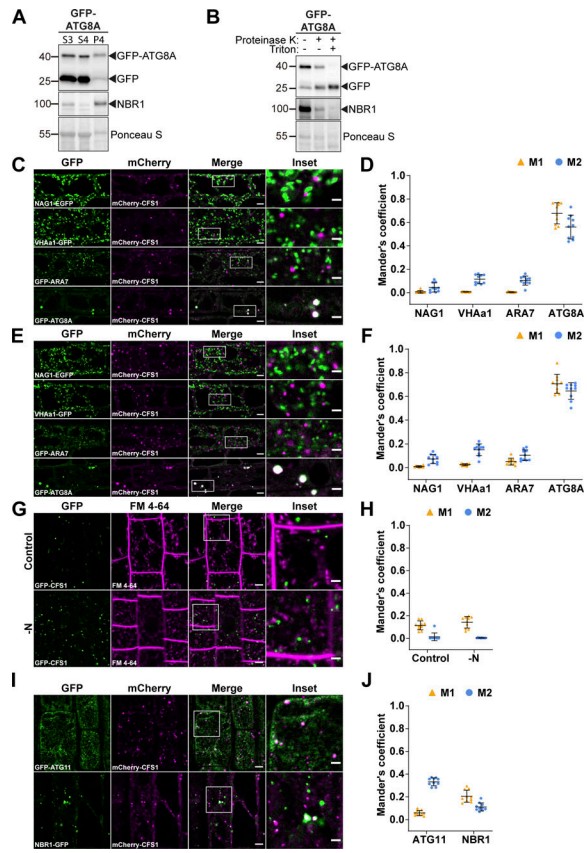

**Figure S1.  Differential centrifugation coupled to affinity purification-mass spectrometry (AP-MS) revealed CFS1 as an autophagosome-associated protein in *A. thaliana*. (A and B)** Ultracentrifugation enriches for intact autophagosomes. **(A)** Western blot analysis of 7-d-old Col-0 seedlings expressing pUBQ::GFP-ATG8A. *Arabidopsis* seedlings were treated with 3 µM Torin 1 for 90 min prior to differential centrifugation described in Fig. 1 A. A total of 5 µg of protein was loaded in each lane. Protein extracts were immunoblotted with anti-GFP and anti-NBR1 antibodies. Representative images of four replicates are shown. Reference protein sizes are labeled as numbers at the left side of the blots (unit: kD). **(B)** Protease protection assay of enriched autophagosomes in A. Autophagosomes were treated with 30 ng/µl proteinase K in the absence or presence of 1% Triton X-100. A total of 5 µg of protein was loaded in each lane. Protein extracts were immunoblotted with anti-GFP and anti-NBR1 antibodies. Representative images of 4 replicates are shown. Reference protein sizes are labeled as numbers at the left side of the blots (unit: kD). **(C and D)** CFS1 localizes to the autophagosomes under control conditions. **(C)** Confocal microscopy images of *Arabidopsis* root epidermal cells co-expressing pUBQ::mCherry-CFS1 with either Golgi body marker p35S::NAG1-EGFP, trans-Golgi network marker pa1::VHAa1-GFP, late endosome marker pRPS5a::GFP-ARA7 or autophagosome marker pUBQ::GFP-ATG8A under control conditions. 5-d-old *Arabidopsis* seedlings were incubated in control 1/2 MS media before imaging. Representative images of 10 replicates are shown. Area highlighted in the white-boxed region in the merge panel was further enlarged and presented in the inset panel. Scale bars, 5 µm. Inset scale bars, 2 µm. **(D)** Quantification of confocal experiments in C showing the Mander's colocalization coefficients between mCherry-CFS1 and the GFP-fused marker NAG1, VHAa1, ARA7, or ATG8A. M1, fraction of GFP-fused marker signal that overlaps with mCherry-CFS1 signal. M2, fraction of mCherry-CFS1 signal that overlaps with GFP-fused marker signal. Bars indicate the mean ± SD of 10 replicates. **(E and F)** CFS1 localizes to the autophagosomes under salt stress. **(E)** Confocal microscopy images of *Arabidopsis* root epidermal cells co-expressing pUBQ::mCherry-CFS1 with either Golgi body marker p35S::NAG1-EGFP, trans-Golgi network marker pa1::VHAa1-GFP, MVB marker pRPS5a::GFP-ARA7 or autophagosome marker pUBQ::GFP-ATG8A under salt stress. 5-d-old *Arabidopsis* seedlings were incubated in 150 mM NaCl-containing 1/2 MS media for 1 h for autophagy induction before imaging. Representative images of 10 replicates are shown. Area highlighted in the white-boxed region in the merge panel was further enlarged and presented in the inset panel. Scale bars, 5 µm. Inset scale bars, 2 µm. **(F)** Quantification of confocal experiments in E showing the Mander's colocalization coefficients between mCherry-CFS1 and the GFP-fused marker NAG1, VHAa1, ARA7, or ATG8A. M1, fraction of GFP-fused marker signal that overlaps with mCherry-CFS1 signal. M2, fraction of mCherry-CFS1 signal that overlaps with GFP-fused marker signal. Bars indicate the mean ± SD of 10 replicates. **(G and H)** CFS1 does not colocalize with the endocytic marker dye FM 4-64. **(G)** Confocal microscopy images of *Arabidopsis* root epidermal cells expressing pUBQ::GFP-CFS1 and stained with FM 4-64. 5-d-old *Arabidopsis* seedlings were first incubated in either control or nitrogen-deficient (−N) 1/2 MS media for 4 h and were then incubated in either control or nitrogen-deficient 1/2 MS media containing 4 µM FM 4-64 for 30 min before imaging. Representative images of 10 replicates are shown. Area highlighted in the white-boxed region in the merge panel was further enlarged and presented in the inset panel. Scale bars, 5 µm. Inset scale bars, 2 µm. **(H)** Quantification of confocal experiments in G showing the Mander's colocalization coefficients between GFP-CFS1 and FM 4-64. M1, fraction of GFP-CFS1 signal that overlaps with FM 4-64 signal. M2, fraction of FM 4-64 signal that overlaps with GFP-CFS1 signal. Bars indicate the mean ± SD of 10 replicates. **(I and J)** CFS1 colocalizes with the autophagosome marker proteins ATG11 and NBR1 during salt stress. **(I)** Confocal microscopy images of *Arabidopsis* root epidermal cells co-expressing pUBQ::mCherry-CFS1 with either pUBQ::GFP-ATG11 or pNBR1::NBR1-GFP. 5-d-old *Arabidopsis* seedlings were incubated in 150 mM NaCl-containing 1/2 MS media for 1 h for autophagy induction before imaging. Representative images of 10 replicates are shown. Area highlighted in the white-boxed region in the merge panel was further enlarged and presented in the inset panel. Scale bars, 5 µm. Inset scale bars, 2 µm. **(J)** Quantification of confocal experiments in I showing the Mander's colocalization coefficients between mCherry-CFS1 and the GFP-fused marker ATG11 or NBR1. M1, fraction of GFP-fused marker signal that overlaps with mCherry-CFS1 signal. M2, fraction of mCherry-CFS1 signal that overlaps with GFP-fused marker signal. Bars indicate the mean ± SD of 10 biological replicates. Source data are available for this figure: SourceData FS1.

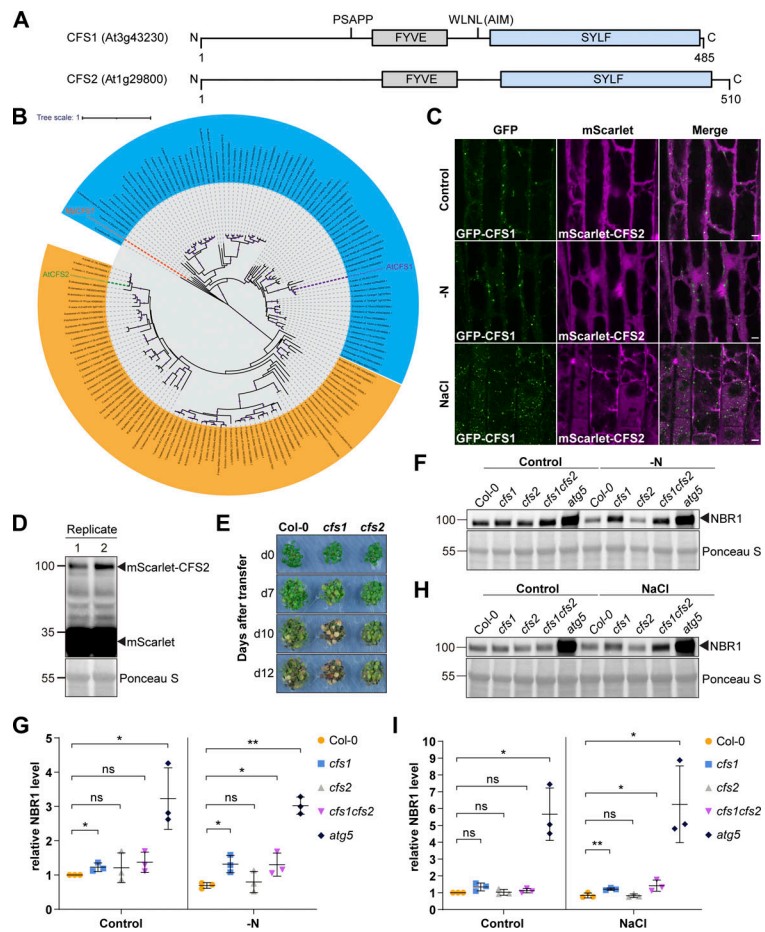

Figure S2. **CFS2 does not play a role in autophagic flux. (A)** Schematic diagrams showing the domain structures of *Arabidopsis* CFS1 and CFS2. **(B)** Maximum likelihood (ML) phylogenetic tree showing that across the plant kingdom, no homology could be detected between CFS1 homologs and CFS2 homologs. Coding sequences of CFS1 and CFS2 homologs were obtained using the BLAST tool against representative species of different plant lineages in Phytozome (Goodstein et al., 2012). The tree was inferred from a 2283-nt-long alignment using the ML method and Tamura-Nei model as implemented by MEGA X (Tamura and Nei, 1993; Kumar et al., 2018). 100 bootstrap method and a discrete Gamma distribution was used to model evolutionary rate differences among sites (5 categories [+G, parameter = 0.8072]) (Felsenstein, 1985). The tree is represented using Interactive Tree Of Life (iTOL) v4 (Letunic and Bork, 2019). Best-scoring ML tree (−98686.19) is shown with purple circles indicating bootstrap values above 80 on their respective clades. The scale bar indicates the evolutionary distance based on the nucleotide substitution rate. All CFS1 homologs are grouped in the blue region while all CFS2 homologs are grouped in the orange region. Genes that encode *A. thaliana* CFS1 (AtCFS1), *A. thaliana* CFS2 (AtCFS2), and *M. polymorpha* CFS1 (MpCFS1) are highlighted with purple, green, and orange colors, respectively. **(C)** Confocal microscopy images of *Arabidopsis* root epidermal cells co-expressing pUBQ::GFP-CFS1 and pUBQ:: mScarlet-CFS2. 5-d-old *Arabidopsis* seedlings were incubated in either control, nitrogen-deficient (−N) or 150 mM NaCl-containing 1/2 MS media before imaging. Representative images of 10 replicates are shown. Note that no CFS2 puncta signals could be detected. Scale bars, 5 μm. **(D)** Western blot analysis of *Arabidopsis* seedlings expressing pUBQ::mScarlet-CFS2 used in C. Total lysates were immunoblotted with anti-RFP antibodies. Images of two replicates are shown. Reference protein sizes are labeled as numbers at the left side of the blots (unit: kD). **(E)** Phenotypic characterization of *Arabidopsis* cfs1 and cfs2 mutants upon nitrogen starvation. 25 *Arabidopsis* seeds per genotype were first grown on 1/2 MS media plates (+1% plant agar) for 1-wk and 7-d-old seedlings were subsequently transferred to nitrogen-deficient (−N) 1/2 MS media plates (+0.8% plant agar) and grown for 2 wk. Plants were grown at 21°C under LEDs with 85 μM/m²/s and a 14 h light/10 h dark photoperiod. d0 depicts the day of transfer. Brightness of pictures was enhanced ≤19% with Adobe Photoshop (2020). Representative images of four replicates are shown. **(F)** Western blots showing the endogenous NBR1 level in Col-0, *cfs1*, *cfs2*, *cfs1cfs2*, or *atg5* under control or nitrogen-starved (−N) conditions. *Arabidopsis* seeds were first grown in 1/2 MS media under continuous light for 1 wk and 7-d-old seedlings were subsequently transferred to control or nitrogen-deficient 1/2 MS media for 12 h. 10 μl of total seedling extract was loaded and immunoblotted with anti-NBR1 antibodies. Representative images of three replicates are shown. Reference protein sizes are labeled as numbers at the left side of the blots (unit: kD). **(G)** Quantification of F showing the relative NBR1 level of Col-0, *cfs1*, *cfs2*, *cfs1cfs2*, or *atg5* under control or nitrogen-starved conditions. Values were calculated through normalization of protein bands to Ponceau S and to untreated (Control) Col-0 and shown as the mean ± SD of three replicates. One-tailed and paired Student's *t* tests were performed to analyze the significance of the relative NBR1 level differences between Col-0 and each mutant. ns, not significant. *, P value <0.05. **, P value <0.01. **(H)** Western blot showing the endogenous NBR1 level in Col-0, *cfs1*, *cfs2*, *cfs1cfs2*, or *atg5* under control or salt-stressed (NaCl) conditions. *Arabidopsis* seeds were first grown in 1/2 MS media under continuous light for 1-wk and 7-d-old seedlings were subsequently transferred to control or 150 mM NaCl-containing 1/2 MS media for 16 h. 10 μl of total seedling extract was loaded and immunoblotted with anti-NBR1 antibodies. Representative images of three replicates are shown. Reference protein sizes are labeled as numbers at the left side of the blots (unit: kD). **(I)** Quantification of H showing the relative NBR1 level of Col-0, *cfs1*, *cfs2*, *cfs1cfs2*, or *atg5* under control or salt stress (NaCl) conditions. Values were calculated through normalization of protein bands to Ponceau S and to untreated (Control) Col-0 and shown as the mean ± SD of three replicates. One-tailed and paired Student's *t* tests were performed to analyze the significance of the relative NBR1 level difference between Col-0 and each mutant. ns, not significant. *, P value <0.05. **, P value <0.01. Source data are available for this figure: SourceData FS2.

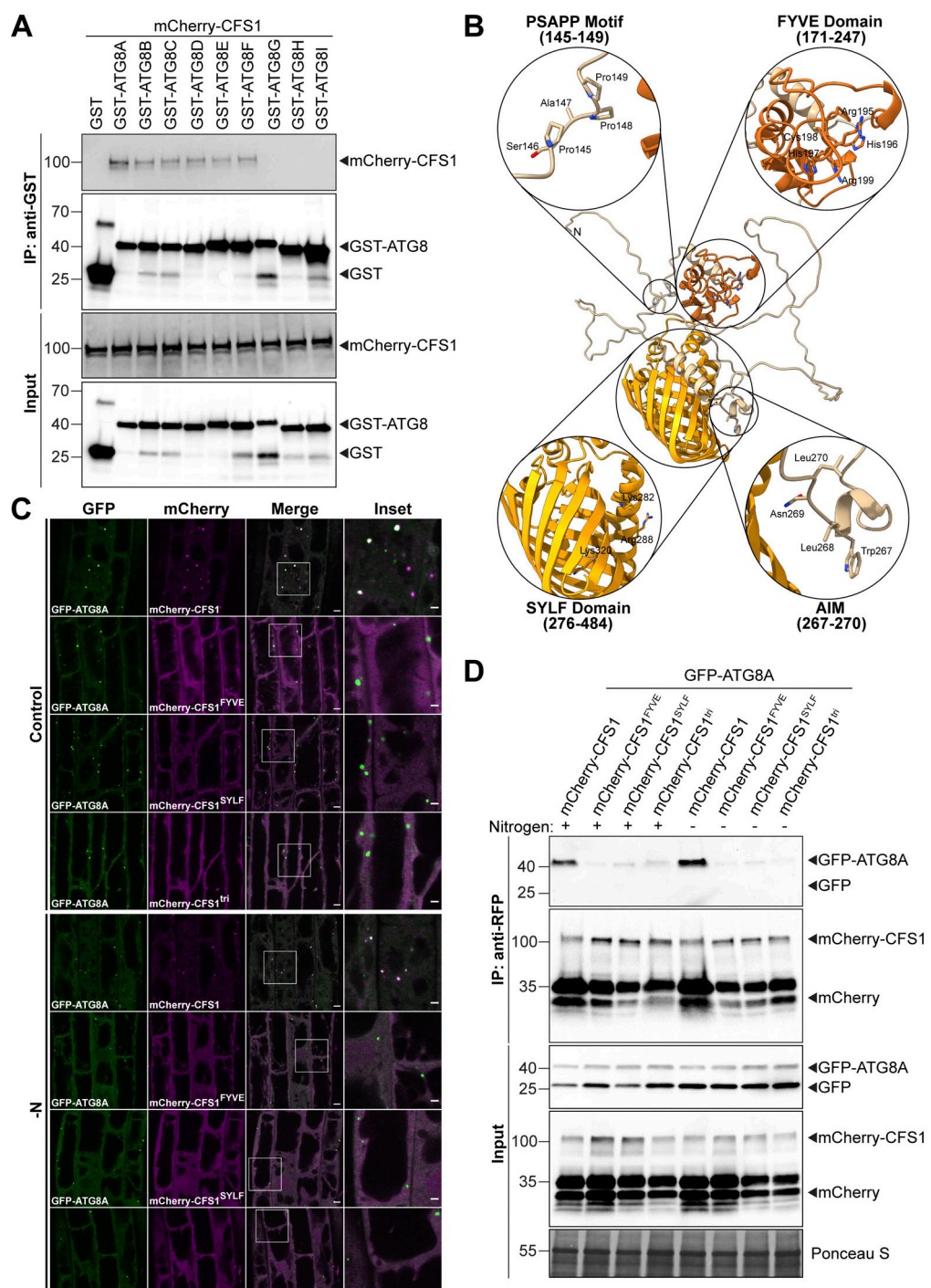

Figure S3. **Functional characterization of CFS1-ATG8 interaction. (A)** GST pull-downs of *E. coli* lysates expressing either GST, GST-ATG8A, GST-ATG8B, GST-ATG8C, GST-ATG8D, GST-ATG8E, GST-ATG8F, GST-ATG8G, GST-ATG8H, or GST-ATG8I and *A. thaliana* whole-seedling lysates expressing mCherry-CFS1. Proteins were visualized by immunoblotting with anti-GST and anti-RFP antibodies. Representative images of two replicates are shown. Reference protein sizes are labeled as numbers at the left side of the blots (unit: kD). **(B)** Homology modeling and domain representation of CFS1. CFS1 structure is shown as ribbons, and relevant motifs and domains are highlighted as zoom-in, with the side chains of relevant residues represented as stick. For clarity, the FYVE and SYLF domains of CFS1 are colored in brick red and orange, respectively. **(C)** Confocal microscopy images of *cfs1* mutants co-expressing pUBQ::GFP-ATG8A with either pUBQ::mCherry-CFS1, pUBQ::mCherry-CFS1[FYVE], pUBQ::mCherry-CFS1[SYLF] or pUBQ::mCherry-CFS1[tri]. 5-d-old *Arabidopsis* seedlings were incubated in either control or nitrogen-deficient (−N) 1/2 MS media for 4 h before imaging. Representative images of 10 replicates are shown. Area highlighted in the white-boxed region in the merge panel was further enlarged and presented in the inset panel. Scale bars, 5 µm. Inset scale bars, 2 µm. **(D)** RFP-Trap pull-down of *Arabidopsis* seedlings co-expressing pUBQ::GFP-ATG8A with either pUBQ::mCherry-CFS1, pUBQ::mCherry-CFS1[FYVE], pUBQ::mCherry-CFS1[SYLF] or pUBQ::mCherry-CFS1[tri]. 7-d-old seedlings were incubated in either control (+) or nitrogen-deficient (−) 1/2 MS media for 12 h. Protein extracts were immunoblotted with anti-GFP and anti-RFP antibodies. Representative images of two replicates are shown. Reference protein sizes are labeled as numbers at the left side of the blots (unit: kD). Source data are available for this figure: SourceData FS3.

Figure S4. **The AIM between the FYVE and SYLF domains of CFS1 is conserved in plants.** Multiple sequencing alignments showing the conserved AIM (WLNL) between the FYVE and SYLF domains of CFS1. The regions that belong to FYVE or SYLF domains are labeled with black boxes.

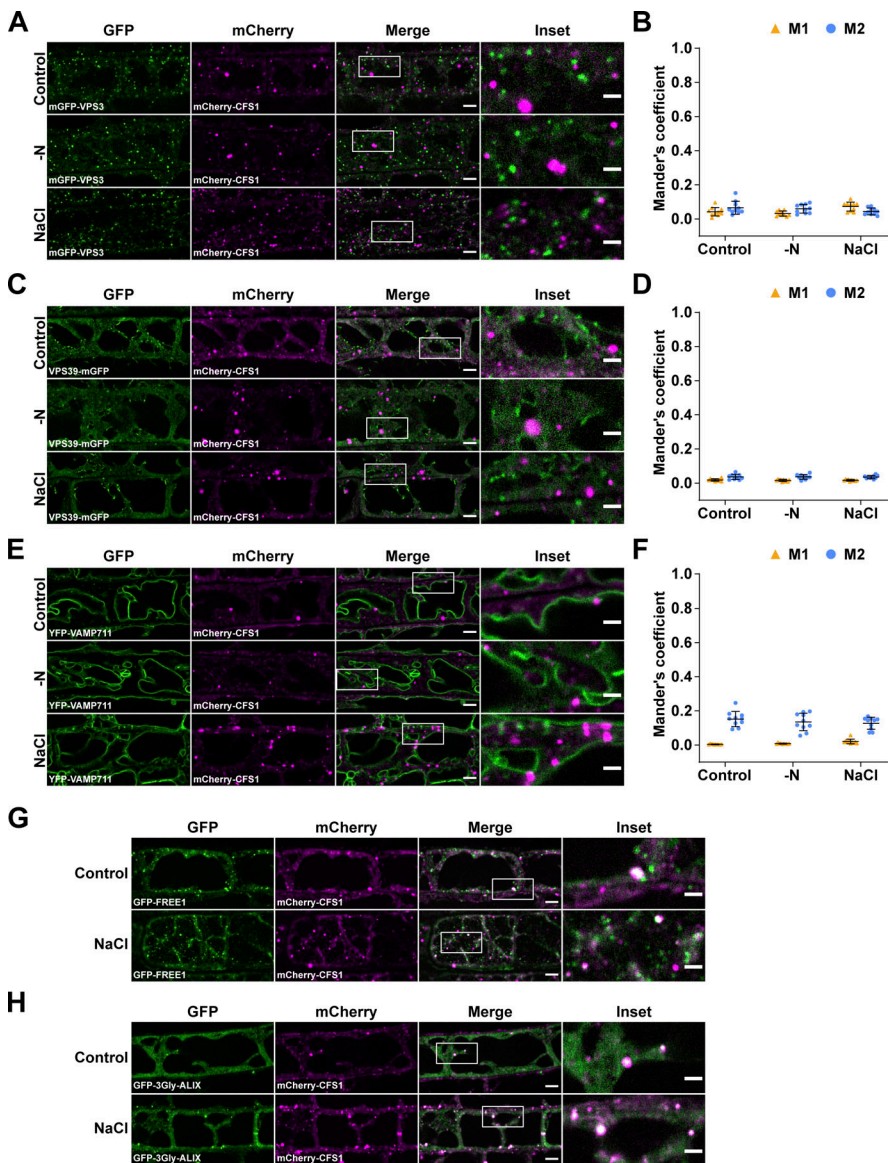

Figure S5.  **CFS1 does not colocalize with VPS3, VPS39, or VAMP711, but partially colocalizes with FREE1 and ALIX. (A)** Confocal microscopy images of *Arabidopsis* root epidermal cells co-expressing pVPS3::mGFP-VPS3 with pUBQ::mCherry-CFS1. 5-d-old *Arabidopsis* seedlings were incubated in either control, nitrogen-deficient (−N) or 150 mM NaCl-containing 1/2 MS media before imaging. Representative images of 10 replicates are shown. Area highlighted in the white-boxed region in the merge panel was further enlarged and presented in the inset panel. Scale bars, 5 μm. Inset scale bars, 2 μm. **(B)** Quantification of confocal experiments in A showing the Mander's colocalization coefficients between mCherry-CFS1 and mGFP-VPS3. M1, fraction of mGFP-VPS3 signal that overlapped with the mCherry-CFS1 signal. M2, fraction of mCherry-CFS1 signal that overlapped with the mGFP-Vps3 signal. Bars indicate the mean ± SD of 10 replicates. **(C)** Confocal microscopy images of *Arabidopsis* root epidermal cells co-expressing pVPS39::VPS39-mGFP with pUBQ::mCherry-CFS1. 5-d-old *Arabidopsis* seedlings were incubated in either control, nitrogen-deficient (−N) or 150 mM NaCl-containing 1/2 MS media before imaging. Representative images of 10 replicates are shown. Area highlighted in the white-boxed region in the merge panel was further enlarged and presented in the inset panel. Scale bars, 5 μm. Inset scale bars, 2 μm. **(D)** Quantification of confocal experiments in C showing the Mander's colocalization coefficients between mCherry-CFS1 and VPS39-mGFP. M1, fraction of VPS39-mGFP signal that overlapped with the mCherry-CFS1 signal. M2, fraction of mCherry-CFS1 signal that overlapped with the VPS39-mGFP signal. Bars indicate the mean ± SD of 10 replicates. **(E)** Confocal microscopy images of *Arabidopsis* root epidermal cells co-expressing pUBQ::YFP-VAMP711 with pUBQ::mCherry-CFS1. 5-d-old *Arabidopsis* seedlings were incubated in either control, nitrogen-deficient (−N) or 150 mM NaCl-containing 1/2 MS media before imaging. Representative images of 10 replicates are shown. Area highlighted in the white-boxed region in the merge panel was further enlarged and presented in the inset panel. Scale bars, 5 μm. Inset scale bars, 2 μm. **(F)** Quantification of confocal experiments in E showing the Mander's colocalization coefficients between mCherry-CFS1 and YFP-VAMP711. M1, fraction of YFP-VAMP711 signal that overlapped with the mCherry-CFS1 signal. M2, fraction of mCherry-CFS1 signal that overlapped with the YFP-VAMP711 signal. Bars indicate the mean ± SD of 10 replicates. **(G)** Representative confocal microscopy images showing the colocalization of p35S::GFP-FREE1 and pUBQ::mCherry-CFS1 in *Arabidopsis* root epidermal cells. 5-d-old *Arabidopsis* seedlings were incubated in either control or 150 mM NaCl-containing 1/2 MS media for 1 h before imaging. Area highlighted in the white-boxed region in the merge panel was further enlarged and presented in the inset panel. Scale bars, 5 μm. Inset scale bars, 2 μm. **(H)** Representative confocal microscopy images showing the colocalization of pALIX::GFP-3Gly-ALIX and pUBQ::mCherry-CFS1 in *Arabidopsis* root epidermal cells. 5-d-old *Arabidopsis* seedlings were incubated in either control or 150 mM NaCl-containing 1/2 MS media for 1 h before imaging. Area highlighted in the white-boxed region in the merge panel was further enlarged and presented in the inset panel. Scale bars, 5 μm. Inset scale bars, 2 μm.

Video 1.   **Time-lapse video showing that mCherry-CFS1 (magenta) moves together with GFP-ATG8A (green) in *Arabidopsis* root epidermal cells.** 5-d-old *Arabidopsis* seedlings co-expressing pUBQ::GFP-ATG8A and pUBQ::mCherry-CFS1 were incubated in 150 mM NaCl-containing 1/2 MS media for 1 h for autophagy induction before imaging. Total imaging time, 60 s. Interval, 1 s. Scale bar, 10 µm.

Video 2.   **Time-lapse video showing that mCherry-CFS1 (magenta) moves together with NBR1-GFP (green) in *Arabidopsis* root epidermal cells.** 5-d-old *Arabidopsis* seedlings co-expressing pNBR1::NBR1-GFP and pUBQ::mCherry-CFS1 were incubated in 150 mM NaCl-containing 1/2 MS media for 1 h for autophagy induction before imaging. Total imaging time, 60 s. Interval, 1 s. Scale bar, 10 µm.

Video 3.   **Time-lapse video showing the partial colocalization and the associated movement between GFP-CFS1 (green) and VPS23A-TagRFP (magenta) in *Arabidopsis* root epidermal cells.** 5-d-old *Arabidopsis* seedlings co-expressing pUBQ::GFP-CFS1 and pVPS23A::VPS23A-TagRFP were incubated in 150 mM NaCl-containing 1/2 MS media for 1 h for autophagy induction before imaging. Total imaging time, 60 s. Interval, 1 s. Scale bar, 10 µm.

Video 4.   **Time-lapse video showing cell-to-cell movement of NBR1-GFP (green)/mCherry-CFS1 (magenta) punctum in *Arabidopsis* root cells.** 5-d-old *Arabidopsis* seedlings co-expressing pNBR1::NBR1-GFP and pUBQ::mCherry-CFS1 were incubated in 150 mM NaCl-containing 1/2 MS media for 1 h for autophagy induction before imaging. Total imaging time, 60 s. Interval, 1 s. Scale bar, 5 µm.

**Provided online are Table S1, Table S2, Table S3, and Table S4. Table S1 shows original mass spectromertry dataset used for analysis. Table S2 shows spectral count of the mass spectromertry data. Table S3 shows final list of proteins identified in analysis. Table S4 shows yeast two hybrid dataset.**

