## [Peer Review File · The Journal of Cell Biology]

Plant autophagosomes mature into amphisomes prior to their delivery to the central vacuole

Jierui Zhao, Mai Bui, Juncai Ma, Fabian Kuenzl, Lorenzo Picchianti, Juan De La Concepcion, Yixuan Chen, Sofia Petsangouraki, Azadeh Mohseni, Marta García-León, Marta Salas-Gomez, Caterina Giannini, Dubois Gwennogan, Roksolana Kobylinska, Marion Clavel, Swen Schellmann, Yvon Jaillais, Jiří Friml, Byung-Ho Kang, and Yasin Dagdas

Corresponding Author(s): Yasin Dagdas, Gregor Mendel Institute-Vienna Biocenter

Review Timeline:

Submission Date:	2022-03-31
Editorial Decision:	2022-04-04
Revision Received:	2022-08-12
Editorial Decision:	2022-09-08
Revision Received:	2022-09-13

Monitoring Editor: Dominique Bergmann

Scientific Editor: Andrea Marat

Transaction Report:

DOI: <https://doi.org/10.1083/jcb.202203139>

Revision 0

Review #1

1. Evidence, reproducibility and clarity:

Evidence, reproducibility and clarity (Required)

In the current work, the authors performed an elegant screen aiming to discover proteins associated with ATG8 on the outer membrane of the autophagosome, i.e., autophagy modulators. They identified CSF1 as a novel autophagy adaptor, connecting ATG8 with the ESCRT protein VPS23A. CSF1 activity is necessary for autophagic flux, binding autophagosomes, and endosomes to produce amphisomes. The authors also add an "evolutionary touch", showing that CSF1 function is conserved in non-vascular plants.

I liked reading this manuscript very much. It is well-written, the methodology elegant and well-executed, and the conclusions are exciting. I have a few minor comments/questions:

- The authors use both ATG8A and ATG8E interchangeably throughout the work. It would be good to add a few words about these isoforms as it might seem strange, especially for the non-expert reader.
- Figure 3F - TEM images showed CSF1 localization on the inner membrane of the autophagosome. They suggest that CSF1 is recruited to the growing phagophore. It would be interesting if the authors discussed the timing of this recruitment a bit? Why would CSF1 be recruited to the autophagosome so soon? Is that so that no autophagosome is missed?
- Figure 4C and Figure 5I - I found it a bit strange that under ConA treatment, the amounts of NBR1 changed considerably (as expected), but the ratio between free GFP and GFP-ATG8 almost didn't change in Col-0 samples. I would expect to see less free GFP. The authors might like to clarify that.
- I was missing the number of biological replicates used in TEM imaging. I could not find it in either the figure legend or the methods section. The authors should add some information about this.
- In the acknowledgments section, there is a typo in Suayib Üstün's name.

2. Significance:

Significance (Required)

Autophagy is an important degradation mechanism in eukaryotic organisms, including plants. However, though the core autophagy-related genes are highly conserved between kingdoms, our knowledge regarding autophagy regulation in plants is lagging behind what is known in yeast and animal systems. ATG8 is a protein localized on the inner and outer membrane of the autophagosome. It functions in recruiting various proteins to interact with autophagosomes, either as cargo, in the inner membrane, or as modulators, on the outer membrane. Identifying

novel ATG8-interacting proteins has been of great interest in recent years. However, most of the work focused on cargo receptors, rather than autophagy regulators. In that sense, the approach taken by the authors is extremely novel. The connection between autophagy and ESCT/MVB-related proteins has been previously implied in plants (the authors might want to also discuss doi:10.1105/tpc.113.113399 and doi: 10.1105/tpc.114.135939), this is the first time the direct connection between autophagosomes and endosomes has been demonstrated, and the connecting protein between autophagy and MVB identified. The manuscript will appeal to autophagy researchers in general, and specifically to plant autophagy researchers.

3. How much time do you estimate the authors will need to complete the suggested revisions:

Estimated time to Complete Revisions (Required)

(Decision Recommendation)

Less than 1 month

4. Review Commons values the work of reviewers and encourages them to get credit for their work. Select 'Yes' below to register your reviewing activity at Publons; note that the content of your review will not be visible on Publons.

Reviewer Publons

Yes

Review #2

1. Evidence, reproducibility and clarity:

Evidence, reproducibility and clarity (Required)

Zhao, Bui, Ma et al., describe a novel autophagy adaptor that mediates the maturation of autophagosomes into amphisomes before they are transported into the lytic vacuole. The authors develop a new method to isolate intact autophagosomes which was used in the manuscript to perform AP-MS in order to identify proteins interacting with autophagosomes. This approach resulted in the identification of CFS1, which is further characterized as a novel autophagy adaptor. CFS1 specifically interacts with ATG8 via an ATG8-interacting motif and is required to

maintain autophagic flux during nitrogen starvation or salt stress. The authors also identify that CFS1 is evolutionarily conserved highlighting its importance within the autophagy pathway in plants. Finally, the authors reveal via a Y2H approach that CFS1 interacts with ESCRT-1 component VPS23, which is required to form amphisomes. Altering the possible interaction between both proteins negatively influences autophagic flux and the authors conclude that amphisomes may serve as a sorting hub for multivesicular bodies and autophagosomes prior delivery to the vacuole.

While the first part characterizing the role of CFS1 as an adaptor is performed in an excellent manner, the VPS23 part may need some small experimental adjustment to strengthen the amphisome-sorting hub idea.

****Specific points:****

- The authors perform nitrogen starvation and salt stress experiments to induce autophagy. I was wondering how *cfs1* mutant behaves under salt stress or even drought stress? Given the role of NBR1 in salt/drought stress (according to Zhou et al., 2013) and impaired flux of NBR1 in the *cfs1* mutant it would be interesting to see how *cfs1* behaves under different stresses (e.g., salt, drought, oxidative stress etc.). This will strengthen the idea that *cfs1* is a "general" autophagy adaptor during stress.
- Fig 2C: The interaction of CFS1AIM with ATG8 is also further strengthened upon autophagy induction. If the authors really want to claim this, I would add some quantification. Otherwise, I would tone down the conclusion. Why is CFS1AIM still interacting with ATG8? Is there another motif within the protein?
- Figure S9B-C: Connecting to my previous question: Do FYVE/SYLF mutants still interact with ATG8 and how would a double mutant of AIM/FYVE or AIM/SYLF behave?
- I am not an expert of TEM but what exactly makes the authors conclude that this is an amphisome? Is it the presence of VPS23? How often do the authors see these structures and can they even observe fusion events of MVBs and autophagosomes? I understand that quantification of TEM data might be very different, but this is an important point to address (either experimentally or a more detailed presentation of the TEM data). Theoretically, those structures should be absent in the *cfs1* mutant or in the CFS1PSAPP complementation line. Did the authors perform TEM in them?
- To my knowledge the authors do not provide any additional interaction data of VPS23/CFS1. Considering their efforts to show the association of CFS1-ATG8 I would expect similar efforts with VPS23-CFS-ATG8 (e.g., Co-IP, including autophagy inducing conditions).
- Is the interaction of CFS1PSAPP and VPS23 really affected? Performing interaction studies might be necessary to provide further evidence.
- Amphisome-like structures were also reported by Zhuang et al., 2013, characterizing the role of SH3P2 in autophagy. Given the interaction of SH3P2 and VPS23 (Isono lab) and the presence of SH3P2 in their AP-MS, the authors could elaborate more on these interactions. Is SH3P2 interacting with CFS1?
- Regarding the speculation: I think this is a very interesting idea. However, as far as I can see from the image the authors do not really image MZ cells, they are already in the TZ or EZ where the autophagic activity is already higher.

2. Significance:

Significance (Required)

The authors impressively demonstrate that CFS1 is a novel autophagy adaptor which is extremely understudied in plants. Thus, their findings are very relevant to the entire plant community, as autophagy and trafficking connects so many different processes. On another note, the authors identify a component that is required to form amphisomes in plants and propose that they serve as a sorting hub, which I personally like very much as an idea/hypothesis. This is a novel idea and these findings are very timely and may change the field entirely. Overall, the quality of the manuscript is outstanding (especially the amount of data and controls) and the discussion is very concise.

3. How much time do you estimate the authors will need to complete the suggested revisions:

Estimated time to Complete Revisions (Required)

(Decision Recommendation)

Between 1 and 3 months

4. Review Commons values the work of reviewers and encourages them to get credit for their work. Select 'Yes' below to register your reviewing activity at Publons; note that the content of your review will not be visible on Publons.

Reviewer Publons

No

Review #3

1. Evidence, reproducibility and clarity:

Evidence, reproducibility and clarity (Required)

In this ms, Zhao et al. show that in plant cells as in mammalian cells, autophagosomes (some?) can fuse with MVBs to form amphisomes which subsequently fuse with the vacuole. Intracellular trafficking of autophagosomes to MVB and formation of amphisomes requires the CFS protein, and the authors thoroughly characterized the CFS1 isoform. They show that CFS1 interacts with ATG8 via an AIM motif, and with VPS23A, an ESCRT/MVB protein. The authors conclude that CFS1 may function as an adapter protein that regulates autophagosome maturation and autophagic flux by making the connection between autophagosomes and the endocytic pathway. The quality of the experiments is outstanding. The data is very interesting even if some of the conclusions should be nuanced or qualified.

****Major comments:****

1. I'm not sure I fully understood all the steps of the subcellular fractionation that led to a fraction enriched in microsomes including autophagosomes and the subsequent biochemical treatments. fig.1A suggests that the authors performed an affinity purification step prior to mass spectrometry, after proteinase K treatment. Could the authors clarify this point? What type of affinity matrix and directed against what type of protein(s)?
2. The authors state in particular in the discussion (L298) that: "Our differential centrifugation-based autophagosomes enrichment procedure selects for proteins that associate with closed, mature autophagosomes." The fraction of which they speak is enriched in microsomes and certainly also in autophagosomes. Some of the microsomes can be closed (inside-out or not), or open. Proteinase K digestion does not necessarily prove that all the non-digested microsomes/autophagosomes were "closed, intact and mature". Furthermore, it is not clear what the authors mean by ATG8E-associated proteins (Fig. 1B, for example). Some cytosolic proteins are known to stick to microsomes or any type of purified organelle.
3. Would CFS1 interact with all Arabidopsis ATG8? What would be the nature of organelles labeled with GFP-CFS1 or mCherry-CFS1 (Fig. 3G) but which would not be autophagosomes or contain ATG8?
4. Consistently, FP-CFS1 detected by western blot is in the form of a doublet (example fig. 2). Could the authors comment on this observation?
5. Would we observe even partial colocalization of ATG8A and VPS23A (with or without NaCl treatment) in the absence of CFS1?
6. Colocalization of CFS1 and VPS23A increases after salt treatment; is this also true for other stresses inducing the autophagy pathway or would this observation be specific to salt stress? And in general, can the authors exclude the fact that the formation of amphisomes would be limited to autophagosomes induced by certain stress, knowing that during salt stress in this case the endocytic pathway is also activated and some proteins of the plasma membrane are actively sent to the vacuole (cf. their hub theory)?
7. I have a fundamental problem with the authors' model shown in fig.6 : According to this scheme, the autophagosome, a double-membrane vesicle, fuses with the MVB and releases its content/cargo into the MVB to form the amphisome. Subsequently, the amphisome would fuse with the vacuole, the cargo of the autophagosome no longer being surrounded by any biological membrane apart the tonoplast. How to explain that the authors manage to count ATG8-labeled puncta in the lumen of the vacuole (fig. 5G-H for example) which should be membrane-bound structures (autophagic bodies)?

****Minor comment:****

I fully agree with the authors that video 4 apparently showing trans-cellular vesicular transport is intriguing and deserves some clarification. I would add to the intrigue by specifying that it seems to me (possibly an artifact?) that the vesicle splits into 2 parts after passing through what appears to be a plasmodesmata.

2. Significance:

Significance (Required)

This work is very interesting and demonstrates experimentally a phenomenon theoretically speculated so far. The only difficulty I have with the conclusions is a certain lack of nuance in the interpretation of the results of very well designed experiments.

3. How much time do you estimate the authors will need to complete the suggested revisions:

Estimated time to Complete Revisions (Required)

(Decision Recommendation)

Between 3 and 6 months

4. Review Commons values the work of reviewers and encourages them to get credit for their work. Select 'Yes' below to register your reviewing activity at Publons; note that the content of your review will not be visible on Publons.

Reviewer Publons

Yes

Manuscript number: RC-2022-01327

Corresponding author: Yasin Dagdas

1. General Statements [optional]

We are forwarding our manuscript "Plant autophagosomes mature into amphisomes prior to their delivery to the central vacuole", which was reviewed by 3 reviewers via the ReviewCommons. As you can see below, all three reviewers are highly supportive of our work and its significance. They are asking some additional evidence to further support our claims. These experiments would certainly improve our manuscript but would not change the overall conclusions. We would need 8 weeks to perform these experiments and modify the manuscript. Please let us know if you want us to do all the experiments that we have listed and agree with our action plan.

Looking forward to hearing from you.

Yours sincerely,

Yasin Dagdas

Byung-Ho Kang

2. Description of the planned revisions

Dear reviewers,

Thank you for reviewing our manuscript "Plant autophagosomes mature into amphisomes prior to their delivery to the central vacuole". We appreciate the time and effort that you dedicated to providing feedback on our manuscript. We are grateful for your insightful comments and constructive suggestions to our work. We have incorporated most of the suggestions and responded to every question that you asked. Please see below, in blue, for a point-by-point response to your comments and concerns. The experiments that we propose to address your concerns are in *italic*.

Reviewer #1 (Evidence, reproducibility and clarity (Required)):

In the current work, the authors performed an elegant screen aiming to discover proteins associated with ATG8 on the outer membrane of the autophagosome, i.e., autophagy modulators. They identified CFS1 as a novel autophagy adaptor, connecting ATG8 with the ESCRT protein VPS23A. CFS1 activity is necessary for autophagic flux, binding autophagosomes, and endosomes to produce amphisomes. The authors also add an "evolutionary touch", showing that CFS1 function is conserved in non-vascular plants. I liked reading this manuscript very much. It is well-written, the methodology elegant and well-executed, and the conclusions are exciting.

Our response: Thank you!

I have a few minor comments/questions:

- The authors use both ATG8A and ATG8E interchangeably throughout the work. It would be good to add a few words about these isoforms as it might seem strange, especially for the non-expert reader.

Our response: Thank you for the advice. We will add the information about Arabidopsis ATG8 isoforms in the text. *We will also perform the in vitro pull downs with other ATG8 isoforms to show that CFS1 can bind different Arabidopsis isoforms.*

- Figure 3F - TEM images showed CFS1 localization on the inner membrane of the autophagosome. They suggest that CFS1 is recruited to the growing phagophore. It would be interesting if the authors discussed the timing of this recruitment a bit? Why would CFS1 be recruited to the autophagosome so soon? Is that so that no autophagosome is missed?

Our response: We will modify the discussion accordingly.

ATG8 is already present in early phagophores and CFS1's ability to bind Pi3P possibly recruits it to the phagophores early on. That's why we think it also localizes partially to the inner autophagosome membranes and undergoes degradation. Therefore, the biogenesis system does not seem to differentiate inner and outer autophagosome membranes, but specific proteins such as ATG4 shaves of the ATG8s that are not bound by adaptor proteins. This is also consistent with a recent story where the authors have studied local fatty acid channeling that fuels phospholipid synthesis required for phagophore growth. The Acyl CoA synthetase, FAA that localizes at the growing tip of the phagophore also undergoes autophagic degradation (Schuetter et al., 2020, Cell).

Revision Plan

- Figure 4C and Figure 5I - I found it a bit strange that under ConA treatment, the amounts of NBR1 changed considerably (as expected), but the ratio between free GFP and GFP-ATG8 almost didn't change in Col-0 samples. I would expect to see less free GFP. The authors might like to clarify that.

Our response: Thank you for pointing out this discrepancy. GFP-ATG8A expression is driven by pUBQ10 constitutive promoter, whereas for NBR1 we use an antibody against the native protein. Overexpressing ATG8 could lead to this difference. That's why we always used multiple experimental read outs to reach to our conclusions. As the reviewer would agree, in two different autophagy inducing conditions, we could still see a clear difference, as we have quantified multiple independent replicates in Fig. 4D or Fig5J.

- I was missing the number of biological replicates used in TEM imaging. I could not find it in either the figure legend or the methods section. The authors should add some information about this.

Our response: Thanks for pointing out this important information that was indeed missing. We have done high-pressure freezing and fixation experiments for more than three independent biological replicates. Specifically, for the quantification of immuno-gold labeling samples, approximately 900 gold particles in 50 TEM images from 8 TEM sections (5-10 micrographs per section) were grouped into autophagosomes, cytosol or other organelles according to their locations. The sections were collected from 5 individual plastic embedded *Arabidopsis* roots with three times of replicates for cryo-fixation and sample embedding. We have now included this information in the methods section.

- In the acknowledgments section, there is a typo in Suayib Üstün's name.

Our response: We have changed it.

Reviewer #1 (Significance (Required)):

Autophagy is an important degradation mechanism in eukaryotic organisms, including plants. However, though the core autophagy-related genes are highly conserved between kingdoms, our knowledge regarding autophagy regulation in plants is lagging behind what is known in yeast and animal systems. ATG8 is a protein localized on the inner and outer membrane of the autophagosome. It functions in recruiting various proteins to interact with autophagosomes, either as cargo, in the inner membrane, or as modulators, on the outer membrane. Identifying novel ATG8-interacting proteins has been of great interest in recent years. However, most of the work focused on cargo receptors, rather than autophagy regulators. In that sense, the approach

taken by the authors is extremely novel. The connection between autophagy and ESCRT/MVB-related proteins has been previously implied in plants (the authors might want to also discuss doi:10.1105/tpc.113.113399 and doi: 10.1105/tpc.114.135939), this is the first time the direct connection between autophagosomes and endosomes has been demonstrated, and the connecting protein between autophagy and MVB identified. The manuscript will appeal to autophagy researchers in general, and specifically to plant autophagy researchers.

Our response: Thank you for your insightful comments. We appreciate it!

Reviewer #2 (Evidence, reproducibility and clarity (Required)):

Zhao, Bui, Ma et al., describe a novel autophagy adaptor that mediates the maturation of autophagosomes into amphisomes before they are transported into the lytic vacuole. The authors develop a new method to isolate intact autophagosomes which was used in the manuscript to perform AP-MS in order to identify proteins interacting with autophagosomes. This approach resulted in the identification of CFS1, which is further characterized as a novel autophagy adaptor. CFS1 specifically interacts with ATG8 via an ATG8-interacting motif and is required to maintain autophagic flux during nitrogen starvation or salt stress. The authors also identify that CFS1 is evolutionary conserved highlighting its importance within the autophagy pathway in plants. Finally, the authors reveal via a Y2H approach that CFS1 interacts with ESCRT-1 component VPS23, which is required to form amphisomes. Altering the possible interaction between both proteins negatively influences autophagic flux and the authors conclude that amphisomes may serve as a sorting hub for multivesicular bodies and autophagosomes prior delivery to the vacuole.

While the first part characterizing the role of CFS1 as an adaptor is performed in an excellent manner, the VPS23 part may need some small experimental adjustment to strengthen the amphisome-sorting hub idea.

Our response: Thank you! We appreciate your constructive feedback.

Specific points:

- The authors perform nitrogen starvation and salt stress experiments to induce autophagy. I was wondering how cfs1 mutant behaves under salt stress or even drought stress? Given the role of NBR1 in salt/drought stress (according to Zhou et al., 2013) and impaired flux of NBR1 in the cfs1 mutant it would be interesting to see how cfs1 behaves under different stresses (e.g.,

salt, drought, oxidative stress etc.). This will strengthen the idea that *cfs1* is a "general" autophagy adaptor during stress.

Our response: Thank you for the advice. We have performed phenotypic assays on *cfs1* mutants using salt stress, however it is hard to find an experimental condition under which the difference between Col-0, *cfs1* and *atg5* is clear (most of the time they all looked similar or all showed etiolation). Since the phenotypic assays take a long time to optimize, we chose the nitrogen starvation for phenotypic assays. But for microscopy and biochemical characterization experiments, we kept using both nitrogen starvation and salt stress to show *cfs1* is a "general" autophagy adaptor. If the reviewer thinks we need another set of phenotypic assay to support CFS1's role in autophagy, we could provide carbon starvation or ER stress phenotypic analysis, which are readily established in our lab. But we are afraid that this would dilute the focus of the story.

- Fig 2C: The interaction of CFS1^{AIM} with ATG8 is also further strengthened upon autophagy induction. If the authors really want to claim this, I would add some quantification. Otherwise, I would tone down the conclusion. Why is CFS1^{AIM} still interacting with ATG8? Is there another motif within the protein?

Our response: *We will quantify the bands and modify the figure as suggested by the reviewer.* We think the SYLF and the FYVE domains are also contributing to ATG8 association. Since autophagosome biogenesis is driven by Pi3P enriched microcompartments, we think CFS1's ability to bind Pi3P contributes to ATG8 association. Consistently, as the reviewer could see in Fig. S9, mutating these domains prevents ATG8 colocalization.

- Figure S9B-C: Connecting to my previous question: Do FYVE/SYLF mutants still interact with ATG8 and how would a double mutant of AIM/FYVE or AIM/SYLF behave?

Our response: *We only performed microscopy experiments with these mutants. But we will now perform the colP experiments to further test the role of each domain/motif in ATG8 interaction.*

- I am not an expert of TEM but what exactly makes the authors conclude that this is an amphisome? Is it the presence of VPS23?

Our response: Amphisomes are defined as intermediate organelles formed by fusion of autophagosomes and endosomes (Sanchez-Wandelmer and Reggiori, EMBO J., 2013). Our serial sections demonstrate the fusion process and the interaction of autophagosome and MVBs through the intermediate amphisome structure in our serial sections. The double-immunogold

labeling is also consistent with a hybrid compartment. So, we think our TEM and the microscopy results are consistent with the amphisome definition.

How often do the authors see these structures and can they even observe fusion events of MVBs and autophagosomes? I understand that quantification of TEM data might be very different, but this is an important point to address (either experimentally or a more detailed presentation of the TEM data). Theoretically, those structures should be absent in the *cfs1* mutant or in the CFS1PSAPP complementation line. Did the authors perform TEM in them?

Our response: As the reviewer pointed out, it is difficult to perform TEM on several different lines. TEM analysis is not an appropriate approach for proving the absence of a particular structure in a cell. However, we have performed live-cell imaging experiments that also showed colocalization of CFS1 with Vps23 (Fig.5).

- To my knowledge the authors do not provide any additional interaction data of VPS23/CFS1. Considering their efforts to show the association of CFS1-ATG8 I would expect similar efforts with VPS23-CFS-ATG8 (e.g., Co-IP, including autophagy inducing conditions).

Our response: The reviewer is right that we don't have the same depth for CFS1-Vps23 interaction. *We will now perform the in vitro and in vivo colIP experiments, suggested by the reviewer.*

- Is the interaction of CFS1PSAPP and VPS23 really affected? Performing interaction studies might be necessary to provide further evidence.

Our response: As we mentioned above, we will perform these experiments under control and autophagy inducing conditions in the WT and PSAPP mutant.

- Amphisome-like structures were also reported by Zhuang et al., 2013, characterizing the role of SH3P2 in autophagy. Given the interaction of SH3P2 and VPS23 (Isono lab) and the presence of SH3P2 in their AP-MS, the authors could elaborate more on these interactions. Is SH3P2 interacting with CFS1?

Our response: We think this is beyond the scope of this manuscript. But *we indeed checked the colocalization of SH3P2 with CFS1 and did see partial colocalization (See the figure below).*

- Regarding the speculation: I think this is a very interesting idea. However, as far as I can see from the image the authors do not really image MZ cells, they are already in the TZ or EZ where the autophagic activity is already higher.

Our response: Thank you for the encouraging comments. *We will now modify that section to include the expression and localization of CFS1 in the roots, which shows very high level of CFS1 localization in the meristematic zone with both UBQ10 and native promoter driven constructs.* The reviewer is right that we have imaged the TZ or EZ but considered together with the localization data that we will now present, it may help picturing this speculative idea.

Reviewer #2 (Significance (Required)):

The authors impressively demonstrate that CFS1 is a novel autophagy adaptor which is extremely understudied in plants. Thus, their findings are very relevant to the entire plant community, as autophagy and trafficking connects so many different processes. On another note, the authors identify a component that is required to form amphisomes in plants and propose that they serve as a sorting hub, which I personally like very much as an idea/hypothesis. This is a novel idea and these findings are very timely and may change the field entirely. Overall, the quality of the manuscript is outstanding (especially the amount of data and controls) and the discussion is very concise.

Our response: Thank you for the positive feedback.

Reviewer #3 (Evidence, reproducibility and clarity (Required)):

In this ms, Zhao et al. show that in plant cells as in mammalian cells, autophagosomes (some?) can fuse with MVBs to form amphisomes which subsequently fuse with the vacuole. Intracellular trafficking of autophagosomes to MVB and formation of amphisomes requires the CFS protein, and the authors thoroughly characterized the CFS1 isoform. They show that CFS1 interacts with ATG8 via an AIM motif, and with VPS23A, an ESCRT/MVB protein. The authors conclude that CFS1 may function as an adaptor protein that regulates autophagosome maturation and autophagic flux by making the connection between autophagosomes and the endocytic pathway. The quality of the experiments is outstanding. The data is very interesting even if some of the conclusions should be nuanced or qualified.

Our response: Thank you for the constructive feedback.

Major comments:

1. I'm not sure I fully understood all the steps of the subcellular fractionation that led to a fraction enriched in microsomes including autophagosomes and the subsequent biochemical treatments. fig.1A suggests that the authors performed an affinity purification step prior to mass

spectrometry, after proteinase K treatment. Could the authors clarify this point? What type of affinity matrix and directed against what type of protein(s)?

Our response: We had this information in the methods part, but now we will add this info in the legends as well. We performed the immunoprecipitation experiments using RFP-trap magnetic agarose beads.

2. The authors state in particular in the discussion (L298) that: "Our differential centrifugation-based autophagosomes enrichment procedure selects for proteins that associate with closed, mature autophagosomes." The fraction of which they speak is enriched in microsomes and certainly also in autophagosomes. Some of the microsomes can be closed (inside-out or not), or open. Proteinase K digestion does not necessarily prove that all the non-digested microsomes/autophagosomes were "closed, intact and mature". Furthermore, it is not clear what the authors mean by ATG8E-associated proteins (Fig. 1B, for example). Some cytosolic proteins are known to stick to microsomes or any type of purified organelle.

Our response: We agree with the reviewer, and we will modify the discussion and the results parts to include these nuances.

3. Would CFS1 interact with all Arabidopsis ATG8? What would be the nature of organelles labeled with GFP-CFS1 or mCherry-CFS1 (Fig. 3G) but which would not be autophagosomes or contain ATG8?

Our response: We will now perform *in vitro* pull downs with all Arabidopsis ATG8 isoforms to test if CFS1 has any specificity. We expect it will interact with most ATG8 isoforms if not all, because we can already detect several ATG8 isoforms in our IP-MS data. CFS1 also interacts with PI3P enriched compartments, since it has a FYVE domain that is known to interact with PI3P.

4. Consistently, GFP-CFS1 detected by western blot is in the form of a doublet (example fig. 2). Could the authors comment on this observation?

Our response: Thank you for the question. This might be due to post translational modifications at the N terminus since we can detect phosphorylation of the serine residues (S9 and S12) in our mass spectrometry datasets. We will mention this possibility by modifying the results section. However, we think in depth investigation of a potential phosphorylation and its role in CFS1 function is beyond the scope of this manuscript.

5. Would we observe even partial colocalization of ATG8A and VPS23A (with or without NaCl treatment) in the absence of CFS1?

Our response: To address reviewer's question, *we will now provide data on colocalization of CFS1-PSAPP mutant with Vps23A under control and autophagy inducing conditions.*

6. Colocalization of CFS1 and VPS23A increases after salt treatment; is this also true for other stresses inducing the autophagy pathway or would this observation be specific to salt stress? And in general, can the authors exclude the fact that the formation of amphisomes would be limited to autophagosomes induced by certain stress, knowing that during salt stress in this case the endocytic pathway is also activated and some proteins of the plasma membrane are actively sent to the vacuole (cf. their hub theory)?

Our response: To address this point, *we will now provide live cell imaging data looking at the colocalization of CFS1 and VPS23A under nitrogen starvation conditions.*

7. I have a fundamental problem with the authors' model shown in fig.6: According to this scheme, the autophagosome, a double-membrane vesicle, fuses with the MVB and releases its content/cargo into the MVB to form the amphisome. Subsequently, the amphisome would fuse with the vacuole, the cargo of the autophagosome no longer being surrounded by any biological membrane apart the tonoplast. How to explain that the authors manage to count ATG8-labeled puncta in the lumen of the vacuole (fig. 5G-H for example) which should be membrane-bound structures (autophagic bodies)?

Our response: In our model we propose that autophagosomes fuse with MVBs in a similar manner to the fusion event that happens at the tonoplast. These MVBs would then go and fuse with the tonoplast. So, we are not proposing that the cargo is released into the MVBs, we propose that the single membraned autophagic bodies are released into the MVBs. Therefore, when the amphisomes fuse with the tonoplast eventually, we will still have single membraned autophagic bodies. We will modify the discussion/model to clarify these nuances.

Minor comment:

I fully agree with the authors that video 4 apparently showing trans-cellular vesicular transport is intriguing and deserves some clarification. I would add to the intrigue by specifying that it seems to me (possibly an artifact?) that the vesicle splits into 2 parts after passing through what appears to be a plasmodesmata.

Our response: We agree with the reviewer and working on substantiating this potentially exciting observation. But we will add this comment on the splitting to accurately describe the events that we observe.

Reviewer #3 (Significance (Required)):

Revision Plan

This work is very interesting and demonstrates experimentally a phenomenon theoretically speculated so far. The only difficulty I have with the conclusions is a certain lack of nuance in the interpretation of the results of very well-designed experiments.

Our response: As we have described above, we will substantially modify the discussion part to include alternative interpretations of our findings. We hope these will reflect the nuances better.

The list of new experiments that we can do to address the reviewer comments:

1. In vitro pull-down experiments to test if CFS1 can interact with all 9 Arabidopsis ATG8 isoforms
2. *In vivo* Co-IP experiments to test the role of CFS1-FYVE/SYLF domains for ATG8 interaction
3. In vivo and in vitro Co-IP experiments to further support VPS23-CFS1 interaction and the role of PSAPP motif in this interaction.
4. Confocal imaging ATG8-VPS23 and ATG8-VPS23-PSAPP mutant under control and autophagy inducing conditions.

April 4, 2022

Re: JCB manuscript #202203139T

Dr. Yasin Dagdas
Gregor Mendel Institute-Vienna Biocenter
Gregor Mendel Institute
Doctor Bohr Gasse 3
Vienna 1030
Austria [AT]

Dear Dr. Dagdas,

Thank you for submitting your transfer manuscript entitled "Plant autophagosomes mature into amphisomes prior to their delivery to the central vacuole" from Review Commons. We have assessed the reviewers' comments and your proposed revision plan. We agree that your study provides interesting new general insight into autophagy that will likely be of high interest to the readership of JCB. Therefore, we invite you to submit a revised manuscript. Overall, we find your proposed revisions reasonable, and agree that testing *csf1* behavior under different stresses (Reviewer #2 first specific point), is not essential for the current study.

GENERAL GUIDELINES:

Text limits: Character count for an Article is < 40,000, not including spaces. Count includes title page, abstract, introduction, results, discussion, and acknowledgments. Count does not include materials and methods, figure legends, references, tables, or supplemental legends.

Figures: Articles may have up to 10 main text figures. Figures must be prepared according to the policies outlined in our Instructions to Authors, under Data Presentation, <https://jcb.rupress.org/site/misc/ifora.xhtml>. All figures in accepted manuscripts will be screened prior to publication.

Supplemental information: There are strict limits on the allowable amount of supplemental data. Articles may have up to 5 supplemental figures. Up to 10 supplemental videos or flash animations are allowed. A summary of all supplemental material should appear at the end of the Materials and methods section.

Please note that JCB now requires authors to submit Source Data used to generate figures containing gels and Western blots with all revised manuscripts. This Source Data consists of fully uncropped and unprocessed images for each gel/blot displayed in the main and supplemental figures. Since your paper includes cropped gel and/or blot images, please be sure to provide one Source Data file for each figure that contains gels and/or blots along with your revised manuscript files. File names for Source Data figures should be alphanumeric without any spaces or special characters (i.e., SourceDataF#, where F# refers to the associated main figure number or SourceDataFS# for those associated with Supplementary figures). The lanes of the gels/blots should be labeled as they are in the associated figure, the place where cropping was applied should be marked (with a box), and molecular weight/size standards should be labeled wherever possible.

The typical timeframe for revisions is three to four months. While most universities and institutes have reopened labs and allowed researchers to begin working at nearly pre-pandemic levels, we at JCB realize that the lingering effects of the COVID-19 pandemic may still be impacting some aspects of your work, including the acquisition of equipment and reagents. Therefore, if you anticipate any difficulties in meeting this aforementioned revision time limit, please contact us and we can work with you to find an appropriate time frame for resubmission. Please note that papers are generally considered through only one revision

cycle, so any revised manuscript will likely be either accepted or rejected.

Thank you for this interesting contribution to Journal of Cell Biology. You can contact us at the journal office with any questions, cellbio@rockefeller.edu or call (212) 327-8588.

Sincerely,

Dominique Bergmann, PhD
Monitoring Editor

Andrea L. Marat, PhD
Senior Scientific Editor

Journal of Cell Biology

1st Revision - Authors' Response to Reviewers: August 12, 2022

Dear Editors,

We are submitting the revised version of our manuscript “Plant autophagosomes mature into amphisomes prior to their delivery to the central vacuole”. Our manuscript was initially reviewed by 3 reviewers via the Review Commons. All three reviewers were highly supportive of our work and its significance, but suggested some experiments to improve the manuscript. To address reviewer’s concerns, we performed several additional experiments and revised the text significantly. We have also re-organized the figures based on JCB’s guidelines. We hope the revised manuscript meets your expectations.

Looking forward to hearing from you.

Yours sincerely,

Yasin Dagdas

Byung-Ho Kang

Response to reviewers

Dear reviewers,

Thank you for reviewing our manuscript "Plant autophagosomes mature into amphisomes prior to their delivery to the central vacuole". We appreciate the time and effort that you dedicated to providing feedback on our manuscript. We are grateful for your insightful comments and constructive suggestions to our work. We have incorporated your suggestions and responded to every question that you asked. Please see below, in blue, for a point-by-point response to your comments and concerns.

Reviewer #1 (Evidence, reproducibility and clarity (Required)):

In the current work, the authors performed an elegant screen aiming to discover proteins associated with ATG8 on the outer membrane of the autophagosome, i.e., autophagy modulators. They identified CFS1 as a novel autophagy adaptor, connecting ATG8 with the ESCRT protein VPS23A. CFS1 activity is necessary for autophagic flux, binding autophagosomes, and endosomes to produce amphisomes. The authors also add an "evolutionary touch", showing that CFS1 function is conserved in non-vascular plants. I liked reading this manuscript very much. It is well-written, the methodology elegant and well-executed, and the conclusions are exciting.

Our response: Thank you!

I have a few minor comments/questions:

1) The authors use both ATG8A and ATG8E interchangeably throughout the work. It would be good to add a few words about these isoforms as it might seem strange, especially for the non-expert reader.

Our response: Thank you for the suggestion. We have performed in vitro pull-down experiments with all the 9 ATG8 isoforms of Arabidopsis. Interestingly, CFS1 interacts with ATG8A-F, but not with ATG8G, H, and I. These results are presented in Fig. S3 A. This suggests, there are other autophagy adaptors that mediates trafficking of ATG8G-I labelled autophagosomes, which we discussed in the discussion part (Lines 372-373).

2) Figure 3F - TEM images showed CFS1 localization on the inner membrane of the autophagosome. They suggest that CFS1 is recruited to the growing phagophore. It would be interesting if the authors discussed the timing of this recruitment a bit? Why would CFS1 be recruited to the autophagosome so soon? Is that so that no autophagosome is missed?

Our response: ATG8 is already present in early phagophores and CFS1's ability to bind Pi3P possibly recruits it to the phagophores early on. That's why we think it also localizes partially to the inner autophagosome membranes and undergoes degradation. Therefore, the biogenesis system does not seem to differentiate inner and outer autophagosome membranes, but specific proteins such as ATG4 shaves of the ATG8s that are not bound by adaptor proteins. This is also consistent with a recent story where the authors have studied local fatty acid channeling that fuels phospholipid synthesis required for phagophore growth. The Acyl CoA synthetase, FAA that localizes at the growing tip of the phagophore also undergoes autophagic degradation (Schuetter et al., 2020, Cell).

We have now included new experiments with the FYVE, SYLF and the triple mutant, where we mutated FYVE, SYLF and the AIM regions, and looked at autophagosome localization and ATG8 association. These experiments are presented in Fig. S3 C and D, and show that each of these domains are important for ATG8 colocalization and interaction.

3) Figure 4C and Figure 5I - I found it a bit strange that under ConA treatment, the amounts of NBR1 changed considerably (as expected), but the ratio between free GFP and GFP-ATG8 almost didn't change in Col-0 samples. I would expect to see less free GFP. The authors might like to clarify that.

Our response: Thank you for pointing out this discrepancy. GFP-ATG8A expression is driven by pUBQ10 constitutive promoter, whereas for NBR1 we use an antibody against the native protein. Overexpressing ATG8 could lead to this difference. That's why we always used multiple experimental read outs to reach to our conclusions. As the reviewer would agree, in two different autophagy inducing conditions, we could still see a clear difference, as we have quantified multiple independent replicates in Fig. 4 D or Fig.8 J.

4) I was missing the number of biological replicates used in TEM imaging. I could not find it in either the figure legend or the methods section. The authors should add some information about this.

Our response: Thanks for pointing out this important information that was indeed missing. We have done high-pressure freezing and fixation experiments for more than three independent biological replicates. Specifically, for the quantification of immuno-gold labeling samples, approximately 900 gold particles in 50 TEM images from 8 TEM sections (5-10 micrographs per section) were grouped into autophagosomes, cytosol or other organelles according to their locations. The sections were collected from 5 individual plastic embedded *Arabidopsis* roots with three times of replicates for cryo-fixation and sample embedding. We have now included this information in the methods section.

5) In the acknowledgments section, there is a typo in Suayib Üstün's name.

Our response: We have changed it.

Reviewer #1 (Significance (Required)):

Autophagy is an important degradation mechanism in eukaryotic organisms, including plants. However, though the core autophagy-related genes are highly conserved between kingdoms, our knowledge regarding autophagy regulation in plants is lagging behind what is known in yeast and animal systems. ATG8 is a protein localized on the inner and outer membrane of the autophagosome. It functions in recruiting various proteins to interact with autophagosomes, either as cargo, in the inner membrane, or as modulators, on the outer membrane. Identifying novel ATG8-interacting proteins has been of great interest in recent years. However, most of the work focused on cargo receptors, rather than autophagy regulators. In that sense, the approach taken by the authors is extremely novel. The connection between autophagy and ESCT/MVB-related proteins has been previously implied in plants (the authors might want to also discuss doi:10.1105/tpc.113.113399 and doi: 10.1105/tpc.114.135939), this is the first time the direct connection between autophagosomes and endosomes has been demonstrated, and the connecting protein between autophagy and MVB identified. The manuscript will appeal to autophagy researchers in general, and specifically to plant autophagy researchers.

Our response: Thank you for your insightful comments. We appreciate it! We have modified the discussion to include those papers (Lines 359-363)

Reviewer #2 (Evidence, reproducibility and clarity (Required)):

Zhao, Bui, Ma et al., describe a novel autophagy adaptor that mediates the maturation of autophagosomes into amphisomes before they are transported into the lytic vacuole. The authors develop a new method to isolate intact autophagosomes which was used in the manuscript to perform AP-MS in order to identify proteins interacting with autophagosomes. This approach resulted in the identification of CFS1, which is further characterized as a novel autophagy adaptor. CFS1 specifically interacts with ATG8 via an ATG8-interacting motif and is required to maintain autophagic flux during nitrogen starvation or salt stress. The authors also identify that CFS1 is evolutionary conserved highlighting its importance within the autophagy pathway in plants. Finally, the authors reveal via a Y2H approach that CFS1 interacts with ESCRT-1 component VPS23, which is required to form amphisomes. Altering the possible interaction between both proteins negatively influences autophagic flux and the authors conclude that amphisomes may serve as a sorting hub for multivesicular bodies and autophagosomes prior delivery to the vacuole.

While the first part characterizing the role of CFS1 as an adaptor is performed in an excellent manner, the VPS23 part may need some small experimental adjustment to strengthen the amphisome-sorting hub idea.

Our response: Thank you! We appreciate your constructive feedback.

Specific points:

1) The authors perform nitrogen starvation and salt stress experiments to induce autophagy. I was wondering how *cfs1* mutant behaves under salt stress or even drought stress? Given the role of NBR1 in salt/drought stress (according to Zhou et al., 2013) and impaired flux of NBR1 in the *cfs1* mutant it would be interesting to see how *cfs1* behaves under different stresses (e.g., salt, drought, oxidative stress etc.). This will strengthen the idea that *cfs1* is a "general" autophagy adaptor during stress.

Our response: Thank you for the advice. We have performed phenotypic assays on *cfs1* mutants using salt stress, however it is hard to find an experimental condition under which the difference between Col-0, *cfs1* and *atg5* is clear (most of the time they all looked similar or all showed etiolation). Since the phenotypic assays take a long time to optimize, we chose the nitrogen starvation for phenotypic assays. But for microscopy and biochemical characterization

experiments, we kept using both nitrogen starvation and salt stress to show *cfs1* is a “general” autophagy adaptor.

2) Fig 2C: The interaction of CFS1^{AIM} with ATG8 is also further strengthened upon autophagy induction. If the authors really want to claim this, I would add some quantification. Otherwise, I would tone down the conclusion. Why is CFS1^{AIM} still interacting with ATG8? Is there another motif within the protein?

Our response: Thank you for the suggestion. To address this concern, we generated new lines where we complemented GFP-ATG8A expressing *cfs1* mutant line with FYVE, SYLF, and the FYVE+SYLF+AIM mutant version of CFS1 and performed microscopy and coIP experiments. These experiments, presented in Fig. S3 C and D, showed that each domain is crucial for colocalization. However, in the coIPs, we could still detect some weak association. We think this is due to VPS23 interaction. When we mutate both the PSAPP (required for VPS23 interaction) and AIM (required for ATG8 interaction), as shown in Fig. 8B, we lost the interaction completely.

3) Figure S9B-C: Connecting to my previous question: Do FYVE/SYLF mutants still interact with ATG8 and how would a double mutant of AIM/FYVE or AIM/SYLF behave?

Our response: As we present in Fig. S3D, even the triple mutant has some residual interaction in the coIP. We think this is due to the VPS23 interaction.

4) I am not an expert of TEM but what exactly makes the authors conclude that this is an amphisome? Is it the presence of VPS23?

Our response: Amphisomes are defined as intermediate organelles formed by fusion of autophagosomes and endosomes (Sanchez-Wandelmer and Reggiori, EMBO J., 2013). Our serial sections demonstrate the fusion process and the interaction of autophagosome and MVBs through the intermediate amphisome structure in our serial sections. The double-immunogold labeling and the confocal microscopy of CFS1-VPS23 colocalization analyses are also consistent with a hybrid compartment that by definition corresponds to amphisomes.

5) How often do the authors see these structures and can they even observe fusion events of MVBs and autophagosomes? I understand that quantification of TEM data might be very different, but this is an important point to address (either experimentally or a more detailed presentation of the TEM data). Theoretically, those structures should be absent in the *cfs1* mutant or in the CFS1PSAPP complementation line. Did the authors perform TEM in them?

Our response: As the reviewer pointed out, it is difficult to perform TEM on several different lines. Also, TEM analysis is not an appropriate approach for proving the absence of a particular structure in a cell. However, to address reviewer's concerns, we performed live-cell imaging experiments that also showed colocalization of CFS1 with VPS23 (Fig.7). Furthermore, we performed coIP, confocal microscopy, autophagic flux, and plate assays to show the importance of VPS23-CFS1 interaction, where we mutated the PSAPP motif that is crucial for CFS1's interaction with VPS23. All of these experiments, presented in Fig. 8, show that similar to ATG8 interaction, VPS23 interaction is also crucial for autophagic flux.

6) To my knowledge the authors do not provide any additional interaction data of VPS23/CFS1. Considering their efforts to show the association of CFS1-ATG8 I would expect similar efforts with VPS23-CFS-ATG8 (e.g., Co-IP, including autophagy inducing conditions).

Our response: The reviewer is right that we don't have the same depth for CFS1-Vps23 interaction. We now performed all the experiments suggested by the reviewer, which points to a PSAPP dependent interaction and its importance for autophagic flux. These experiments are presented in Fig. 8.

7) Is the interaction of CFS1PSAPP and VPS23 really affected? Performing interaction studies might be necessary to provide further evidence.

Our response: As we mentioned above, we performed these experiments and present them in Fig. 8.

8) Amphisome-like structures were also reported by Zhuang et al., 2013, characterizing the role of SH3P2 in autophagy. Given the interaction of SH3P2 and VPS23 (Isono lab) and the presence of SH3P2 in their AP-MS, the authors could elaborate more on these interactions. Is SH3P2 interacting with CFS1?

Our response: SH3P2 is involved in both endocytosis and autophagy, and the literature on this protein is rather confusing. Therefore, we think this is beyond the scope of this manuscript. But

we indeed checked the colocalization of SH3P2 with CFS1 and did see partial colocalization (See the figure below).

9) Regarding the speculation: I think this is a very interesting idea. However, as far as I can see from the image the authors do not really image MZ cells, they are already in the TZ or EZ where the autophagic activity is already higher.

Our response: Thank you for the encouraging comments. We have now included the localization pattern on CFS1 that shows very high accumulation levels in the QC of Arabidopsis roots. This data is presented in Fig. 10.

Reviewer #2 (Significance (Required)):

The authors impressively demonstrate that CFS1 is a novel autophagy adaptor which is extremely understudied in plants. Thus, their findings are very relevant to the entire plant community, as autophagy and trafficking connects so many different processes. On another note, the authors identify a component that is required to form amphisomes in plants and propose that they serve as a sorting hub, which I personally like very much as an idea/hypothesis. This is a novel idea and these findings are very timely and may change the field entirely. Overall, the quality of the manuscript is outstanding (especially the amount of data and controls) and the discussion is very concise.

Our response: Thank you for the positive feedback.

Reviewer #3 (Evidence, reproducibility and clarity (Required)):

In this ms, Zhao et al. show that in plant cells as in mammalian cells, autophagosomes (some?) can fuse with MVBs to form amphisomes which subsequently fuse with the vacuole. Intracellular trafficking of autophagosomes to MVB and formation of amphisomes requires the CFS protein, and the authors thoroughly characterized the CFS1 isoform. They show that CFS1 interacts with ATG8 via an AIM motif, and with VPS23A, an ESCRT/MVB protein. The authors conclude that

CFS1 may function as an adaptor protein that regulates autophagosome maturation and autophagic flux by making the connection between autophagosomes and the endocytic pathway. The quality of the experiments is outstanding. The data is very interesting even if some of the conclusions should be nuanced or qualified.

Our response: Thank you for the constructive feedback.

Major comments:

1) I'm not sure I fully understood all the steps of the subcellular fractionation that led to a fraction enriched in microsomes including autophagosomes and the subsequent biochemical treatments. fig.1A suggests that the authors performed an affinity purification step prior to mass spectrometry, after proteinase K treatment. Could the authors clarify this point? What type of affinity matrix and directed against what type of protein(s)?

Our response: We had this information in the methods part, but now we also included it in the legends as well. We performed the immunoprecipitation experiments using RFP-trap magnetic agarose beads.

2) The authors state in particular in the discussion (L298) that: "Our differential centrifugation-based autophagosomes enrichment procedure selects for proteins that associate with closed, mature autophagosomes." The fraction of which they speak is enriched in microsomes and certainly also in autophagosomes. Some of the microsomes can be closed (inside-out or not), or open. Proteinase K digestion does not necessarily prove that all the non-digested microsomes/autophagosomes were "closed, intact and mature". Furthermore, it is not clear what the authors mean by ATG8E-associated proteins (Fig. 1B, for example). Some cytosolic proteins are known to stick to microsomes or any type of purified organelle.

Our response: Just to clarify, we do an RFP pull down after obtaining the microsome enriched P4 fraction. So, we think it is ok to say that we are enriching for ATG8E-associated proteins. However, we agree with the reviewer that for each candidate further experiments are needed to connect them to autophagy.

3) Would CFS1 interact with all Arabidopsis ATG8? What would be the nature of organelles labeled with GFP-CFS1 or mCherry-CFS1 (Fig. 3G) but which would not be autophagosomes or contain ATG8?

Our response: Thank you for the question. We now performed pull down experiments with all Arabidopsis ATG8 isoforms and showed that CFS1 interacts with ATG8A-F, but not with ATG8G-I. This experiment is presented in Fig. S3 A. We think CFS1 positive ATG8A/E negative puncta are likely to be Pi3P enriched endomembrane compartments.

4) Consistently, GFP-CFS1 detected by western blot is in the form of a doublet (example fig. 2). Could the authors comment on this observation?

Our response: Thank you for the question. This might be due to post translational modifications at the N terminus, since we can detect phosphorylation of the serine residues (S9 and S12) in our mass spectrometry datasets. However, we think in depth investigation of a potential phosphorylation and its role in CFS1 function is beyond the scope of this manuscript.

5) Would we observe even partial colocalization of ATG8A and VPS23A (with or without NaCl treatment) in the absence of CFS1?

Our response: We have addressed this comment with several experiments where we looked at the role of PSAPP motif in autophagic flux. These experiments are presented in Fig. 8.

6) Colocalization of CFS1 and VPS23A increases after salt treatment; is this also true for other stresses inducing the autophagy pathway or would this observation be specific to salt stress? And in general, can the authors exclude the fact that the formation of amphisomes would be limited to autophagosomes induced by certain stress, knowing that during salt stress in this case the endocytic pathway is also activated and some proteins of the plasma membrane are actively sent to the vacuole (cf. their hub theory)?

Our response: We thank the reviewer for this suggestion. We think this is not a salt specific response, since we also see a nitrogen starvation response in the figures presented in Fig. 8.

7) I have a fundamental problem with the authors' model shown in fig.6: According to this scheme, the autophagosome, a double-membrane vesicle, fuses with the MVB and releases its content/cargo into the MVB to form the amphisome. Subsequently, the amphisome would fuse with the vacuole, the cargo of the autophagosome no longer being surrounded by any biological membrane apart the tonoplast. How to explain that the authors manage to count ATG8-labeled puncta in the lumen of the vacuole (fig. 5G-H for example) which should be membrane-bound structures (autophagic bodies)?

Our response: In our model we propose that autophagosomes fuse with MVBs in a similar manner to the fusion event that happens at the tonoplast. These MVBs would then go and fuse with the tonoplast. So, we are not proposing that the cargo is released into the MVBs, we propose that the single membraned autophagic bodies are released into the MVBs. Therefore, when the amphisomes fuse with the tonoplast eventually, we will still have single membraned autophagic bodies. We modified the model, presented in Fig. 9, accordingly.

Minor comment:

I fully agree with the authors that video 4 apparently showing trans-cellular vesicular transport is intriguing and deserves some clarification. I would add to the intrigue by specifying that it seems to me (possibly an artifact?) that the vesicle splits into 2 parts after passing through what appears to be a plasmodesmata.

Our response: We agree with the reviewer and working on substantiating this potentially exciting observation. Nevertheless, we modified the text to accurately describe the events that we observe.

Reviewer #3 (Significance (Required)):

This work is very interesting and demonstrates experimentally a phenomenon theoretically speculated so far. The only difficulty I have with the conclusions is a certain lack of nuance in the interpretation of the results of very well-designed experiments.

Our response: As we have described above, we modified the result and discussion part to include alternative interpretations of our findings. We hope these will reflect the nuances better.

September 8, 2022

RE: JCB Manuscript #202203139R

Dr. Yasin Dagdas
Gregor Mendel Institute-Vienna Biocenter
Gregor Mendel Institute
Doctor Bohr Gasse 3
Vienna 1030
Austria [AT]

Dear Dr. Dagdas:

Thank you for submitting your revised manuscript entitled "Plant autophagosomes mature into amphisomes prior to their delivery to the central vacuole". As you will see, the reviewers are all very positive about your revised study therefore we would be happy to publish your paper in JCB pending final revisions necessary to meet our formatting guidelines (see details below).

A. MANUSCRIPT ORGANIZATION AND FORMATTING:

- 1) Text limits: Character count for Articles is < 40,000, not including spaces. Count includes abstract, introduction, results, discussion, and acknowledgments. Count does not include title page, figure legends, materials and methods, references, tables, or supplemental legends.
- 2) Figures limits: Articles may have up to 10 main text figures.
- 3) Figure formatting: Scale bars must be present on all microscopy images, including inset magnifications. Molecular weight or nucleic acid size markers must be included on all gel electrophoresis.
- 4) Statistical analysis: Error bars on graphic representations of numerical data must be clearly described in the figure legend. The number of independent data points (n) represented in a graph must be indicated in the legend. Statistical methods should be explained in full in the materials and methods. For figures presenting pooled data the statistical measure should be defined in the figure legends. Please also be sure to indicate the statistical tests used in each of your experiments (either in the figure legend itself or in a separate methods section) as well as the parameters of the test (for example, if you ran a t-test, please indicate if it was one- or two-sided, etc.). Also, if you used parametric tests, please indicate if the data distribution was tested for normality (and if so, how). If not, you must state something to the effect that "Data distribution was assumed to be normal but this was not formally tested."
- 5) Abstract and title: The abstract should be no longer than 160 words and should communicate the significance of the paper for a general audience. The title should be less than 100 characters including spaces. Make the title concise but accessible to a general readership.
- 6) Materials and methods: Should be comprehensive and not simply reference a previous publication for details on how an experiment was performed. Please provide full descriptions in the text for readers who may not have access to referenced manuscripts. For example, please briefly describe the mass spec preparation as described in Stephani and Picchianti 2020.
- 7) Please be sure to provide the sequences for all of your primers/oligos and RNAi constructs in the materials and methods. You must also indicate in the methods the source, species, and catalog numbers (where appropriate) for all of your antibodies. Please also indicate the acquisition and quantification methods for immunoblotting/western blots.
- 8) Microscope image acquisition: The following information must be provided about the acquisition and processing of images:
 - a. Make and model of microscope
 - b. Type, magnification, and numerical aperture of the objective lenses
 - c. Temperature
 - d. Imaging medium
 - e. Fluorochromes
 - f. Camera make and model

g. Acquisition software

h. Any software used for image processing subsequent to data acquisition. Please include details and types of operations involved (e.g., type of deconvolution, 3D reconstitutions, surface or volume rendering, gamma adjustments, etc.).

10) Supplemental materials: There are strict limits on the allowable amount of supplemental data. Articles may have up to 5 supplemental figures. Please also note that tables, like figures, should be provided as individual, editable files. A summary of all supplemental material should appear at the end of the Materials and methods section.

13) ORCID IDs: ORCID IDs are unique identifiers allowing researchers to create a record of their various scholarly contributions in a single place. At resubmission of your final files, please consider providing an ORCID ID for as many contributing authors as possible.

Please note that JCB now requires authors to submit Source Data used to generate figures containing gels and Western blots with all revised manuscripts. This Source Data consists of fully uncropped and unprocessed images for each gel/blot displayed in the main and supplemental figures. Since your paper includes cropped gel and/or blot images, please be sure to provide one Source Data file for each figure that contains gels and/or blots along with your revised manuscript files. File names for Source Data figures should be alphanumeric without any spaces or special characters (i.e., SourceDataF#, where F# refers to the associated main figure number or SourceDataFS# for those associated with Supplementary figures). The lanes of the gels/blots should be labeled as they are in the associated figure, the place where cropping was applied should be marked (with a box), and molecular weight/size standards should be labeled wherever possible.

B. FINAL FILES:

Thank you for this interesting contribution, we look forward to publishing your paper in Journal of Cell Biology.

Sincerely,

Dominique Bergmann, PhD
Monitoring Editor

Andrea L. Marat, PhD
Senior Scientific Editor

Journal of Cell Biology

Reviewer #1 (Comments to the Authors (Required)):

I am very happy that the authors addressed all of my suggestions. The additional interaction data makes the conclusions even more solid. Fantastic work, congrats to the authors.

Reviewer #2 (Comments to the Authors (Required)):

In this version of their manuscript, the authors have taken into account all my previous remarks in a satisfactory way. The additional experiments and the results obtained clarify relatively minor points raised by the different reviewers in the first version of the manuscript.

One minor point that may deserve the attention of the authors: in the model presented in figure 9, I am not sure that the outer membrane of the amphisome would be decorated with ATG8 molecules because of their recycling.

Reviewer #3 (Comments to the Authors (Required)):

The authors have addressed most of my concerns as well as the other reviewers. The new experiments are well-executed and contribute to the overall manuscript.